# FedREP: A Byzantine-Robust, Communication-Efficient and Privacy-Preserving Framework for Federated Learning

## Abstract

Federated learning (FL) has recently become a hot research topic, in which Byzantine robustness, communication efficiency and privacy preservation are three important aspects. However, the tension among these three aspects makes it hard to simultaneously take all of them into account. In view of this challenge, we theoretically analyze the conditions that a communication compression method should satisfy to be compatible with existing Byzantine-robust methods and privacy-preserving methods. Motivated by the analysis results, we propose a novel communication compression method called *consensus sparsification* (ConSpar). To the best of our knowledge, ConSpar is the first communication compression method that is designed to be compatible with both Byzantine-robust methods and privacy-preserving methods. Based on ConSpar, we further propose a novel FL framework called *FedREP*, which is Byzantine-robust, communication-efficient and privacy-preserving. We theoretically prove the Byzantine robustness and the convergence of FedREP. Empirical results show that FedREP can significantly outperform communication-efficient privacy-preserving baselines. Furthermore, compared with Byzantine-robust communication-efficient baselines, FedREP can achieve comparable accuracy with an extra advantage of privacy preservation.

## 1 Introduction

Federated learning (FL), in which participants (also called clients) collaborate to train a learning model while keeping data privately-owned, has recently become a hot research topic (Konevcnỳ et al., 2016; McMahan & Ramage, 2017). Compared to traditional data-center based distributed learning (Haddadpour et al., 2019; Jaggi et al., 2014; Lee et al., 2017; Lian et al., 2017; Shamir et al., 2014; Sun et al., 2018; Yu et al., 2019a; Zhang & Kwok, 2014; Zhao et al., 2017; 2018; Zhou et al., 2018; Zinkevich et al., 2010), service providers have less control over clients and the network is usually less stable with smaller bandwidth in FL applications. Furthermore, participants will also take the risk of privacy leakage in FL if privacy-preserving methods are not used. Consequently, Byzantine robustness, communication efficiency and privacy preservation have become three important aspects of FL methods (Kairouz et al., 2021) and have attracted much attention in recent years.

**Byzantine robustness.** In FL applications, failure in clients or network transmission may not get discovered and resolved in time (Kairouz et al., 2021). Moreover, some clients may get attacked by an adversarial party, sending incorrect or even harmful information purposely. The clients in failure or under attack are also called Byzantine clients. To obtain robustness against Byzantine clients, there are mainly three different ways, which are known as redundant computation, server validation and robust aggregation, respectively. Redundant computation methods (Chen et al., 2018; Konstantinidis & Ramamoorthy, 2021; Rajput et al., 2019) require different clients to compute gradients for the same training instances. These methods are mostly for traditional data-center based distributed learning, but unavailable in FL due to the privacy principle. In server validation methods (Xie et al., 2019b; 2020b), server validates clients' updates based on a public dataset. However, the performance of server validation methods depends on the quantity and quality of training instances. In many scenarios, it is hard to obtain a large-scale high-quality public dataset. The third way is to replace the mean aggregation on server with robust aggregation (Alistarh et al., 2018; Bernstein et al., 2019; Blanchard et al., 2017; Chen et al., 2017; Ghosh et al., 2020; Karimireddy et al., 2021; Li et al.,

Table 1: Comparison among different methods in terms of the three aspects of federated learning

| Method | Byzt.-robust | Comm.-efficient | Privacy-preserving |
|---|---|---|---|
| RCGD (Ghosh et al., 2021) | ✓ | ✓ | - |
| F$^2$ed-Learning (Wang et al., 2020) | ✓ | - | ✓ |
| SHARE (Velicheti et al., 2021) | ✓ | - | ✓ |
| SparseSecAgg (Ergun et al., 2021) | - | ✓ | ✓ |
| FedREP (Ours) | ✓ | ✓ | ✓ |

2019; Sohn et al., 2020; Yin et al., 2018; 2019). Compared to redundant computation and server validation, robust aggregation usually has a wider scope of application. Many Byzantine-robust FL methods (Wang et al., 2020; Xie et al., 2019a) take this way.

**Communication efficiency.** In many FL applications, server and clients are connected by wide area network (WAN), which is usually less stable and has smaller bandwidth than the network in traditional data-center based distributed machine learning. Therefore, communication cost should also be taken into consideration. Local updating technique (Konevcnỳ et al., 2016; McMahan et al., 2017; Yu et al., 2019b; Zhao et al., 2017; 2018), where clients locally update models for several iterations before global aggregation, is widely used in FL methods. Communication cost can also be reduced by communication compression techniques, which mainly include quantization (Alistarh et al., 2017; Faghri et al., 2020; Gandikota et al., 2021; Safaryan & Richtárik, 2021; Seide et al., 2014; Wen et al., 2017), sparsification (Aji & Heafield, 2017; Chen et al., 2020; Stich et al., 2018; Wangni et al., 2018) and sketching[1] (Rothchild et al., 2020). Error compensation (also known as error feedback) technique (Gorbunov et al., 2020; Wu et al., 2018; Xie et al., 2020c) is proposed to alleviate the accuracy decrease for communication compression methods. Moreover, different techniques can be combined to further reduce communication cost (Basu et al., 2020; Lin et al., 2018).

**Privacy preservation.** Most of the existing FL methods send gradients or model parameters during training process while keeping data decentralized due to the privacy principle. However, sending gradients or model parameters may also cause privacy leakage problems (Kairouz et al., 2021; Zhu et al., 2019). Random noise is used to hide the true input values in some privacy-preserving techniques such as differential privacy (DP) (Abadi et al., 2016; Jayaraman et al., 2018; McMahan et al., 2018) and sketching (Liu et al., 2019; Zhang & Wang, 2021). Secure aggregation (SecAgg) (Bonawitz et al., 2017; Choi et al., 2020) is proposed to ensure the privacy of computation. Based on secure multiparty computation (MPC) and Shamir's $t$-out-of-$n$ secret sharing (Shamir, 1979), SecAgg allows server to obtain only the average value for global model updating without knowing each client's local model parameters (or gradients). Since noises can be simply added to stochastic gradients in most of the existing FL methods to provide input privacy, we mainly focus on how to combine SecAgg with Byzantine-robust and communication-efficient methods in this work.

There are also some methods that consider two of the three aspects (Byzantine robustness, communication efficiency and privacy preservation), including RCGD (Ghosh et al., 2021), F$^2$ed-Learning (Wang et al., 2020), SHARE (Velicheti et al., 2021) and SparseSecAgg (Ergun et al., 2021), which we summarize in Table 1. However, the tension among these three aspects makes it hard to simultaneously take all of the three aspects into account. In view of this challenge, we theoretically analyze the tension among Byzantine robustness, communication efficiency and privacy preservation, and propose a novel FL framework called FedREP. The main contributions are listed as follows:

- We theoretically analyze the conditions that a communication compression method should satisfy to be compatible with Byzantine-robust methods and privacy-preserving methods. Motivated by the analysis results, we propose a novel communication compression method called *consensus sparsification* (ConSpar). To the best of our knowledge, ConSpar is the first communication compression method that is designed to be compatible with both Byzantine-robust methods and privacy-preserving methods.

- Based on ConSpar, we further propose a novel FL framework called *FedREP*, which is Byzantine-robust, communication-efficient and privacy-preserving.

---

[1]Sketching technique can be used in different ways for reducing communication cost or protecting privacy. Thus, sketching appears in both communication-efficient methods and privacy-preserving methods.

- We theoretically prove the Byzantine robustness and the convergence of FedREP.
- We empirically show that FedREP can significantly outperform existing communication-efficient privacy-preserving baselines. Furthermore, compared with Byzantine-robust communication-efficient baselines, FedREP can achieve comparable accuracy with an extra advantage of privacy preservation.

## 2 PRELIMINARY

In this work, we mainly focus on the conventional federated learning setup with $m$ clients and a single server (Kairouz et al., 2021), which collaboratively to solve the finite-sum optimization problem:

$$\min_{\mathbf{w} \in \mathbb{R}^d} F(\mathbf{w}) = \sum_{k=1}^{m} p_k F_k(\mathbf{w}) \qquad \text{s.t.} \qquad F_k(\mathbf{w}) = \frac{1}{|\mathcal{D}_k|} \sum_{i \in \mathcal{D}_k} f_i(\mathbf{w}), \ k = 1, 2, \ldots, m, \quad (1)$$

where $\mathbf{w}$ is the model parameter and $d$ is the dimension of parameter. $f_i(\mathbf{w})$ is the empirical loss of parameter $\mathbf{w}$ on the $i$-th training instance. $\mathcal{D}_k$ denotes the index set of instances stored on the $k$-th client and $F_k(\mathbf{w})$ is the local loss function of the $k$-th client. We assume that $\mathcal{D}_k \cap \mathcal{D}_{k'} = \emptyset$ when $k \neq k'$, and consider the instances on different clients with the same value as several distinct instances. $p_k$ is the weight of the $k$-th client satisfying that $p_k > 0$ and $\sum_{k=1}^{m} p_k = 1$. A common setting of $p_k$ is that $p_k = |\mathcal{D}_k|/(\sum_{k=1}^{m} |\mathcal{D}_k|)$. For simplicity, we assume $|\mathcal{D}_k| = |\mathcal{D}_{k'}|$ for all $k, k' \in [m]$ and thus $p_k = 1/m$. The analysis in this work can be extended to general cases in a similar way.

Most federated learning methods (Karimireddy et al., 2020; McMahan et al., 2017; 2018) to solve problem (1) are based on distributed stochastic gradient descent and its variants, where clients locally update model parameters according to its own training instances and then communicate with server for model aggregation in each iteration. However, the size of many widely-used models (Devlin et al., 2018; He et al., 2016) is very large, leading to heavy communication cost. Thus, techniques to reduce communication cost are required in FL. Moreover, FL methods should also be robust to potential Byzantine attack and privacy attack in real-world applications.

**Byzantine attack.** Let $[m] = \{1, 2, \ldots, m\}$ denote the set of clients. $\mathcal{G} \subseteq [m]$ denotes the set of good (non-Byzantine) clients, which will execute the algorithm faithfully. The rest clients $[m] \setminus \mathcal{G}$ are Byzantine, which may act maliciously and send arbitrary values. The server, which is usually under service provider's control, will faithfully execute the algorithm as well. This Byzantine attack model is consistent to that in many previous works (Karimireddy et al., 2021). Although there are some works (Burkhalter et al., 2021) focusing on another types of attacks called backdoor attacks (Kairouz et al., 2021), in this paper we mainly focus on Byzantine attacks. The purpose of Byzantine attacks is to degrade the model performance. One typical technique to defend against Byzantine attacks is robust aggregation (Kairouz et al., 2021), which guarantees bounded aggregation error even if Byzantine clients send incorrect values.

**Privacy attack.** In a typical FL method, server is responsible for using the average of clients' local updating values for global model updating. However, local updating information may be used to recover client's training instances (Zhu et al., 2019), which will increase the risk of privacy leakage. Thus, server is prohibited to directly receive individual client's updating information by the requirement of privacy preservation (Kairouz et al., 2021). Secure aggregation (Bonawitz et al., 2017) is a typical privacy-preserving method, which only allows server to have access to the average value for global model updating.

There are mainly two different types of FL settings, which are also called cross-silo FL and cross-device FL (Kairouz et al., 2021). We mainly focus on the cross-silo FL setting in this paper, where the number of clients $m$ is usually not too large and all clients can participate in each training iteration. Meanwhile, in this paper we mainly focus on synchronous FL methods.

## 3 METHODOLOGY

In this section, we analyze the conditions that a communication compression method should satisfy to be compatible with Byzantine-robust methods and privacy-preserving methods. Based on the analysis, we propose a novel communication compression method called *consensus sparsification* and a novel

federated learning framework called *FedREP* that is Byzantine-robust, communication-efficient and privacy-preserving. In FedREP, we adopt robust aggregation technique to obtain Byzantine robustness due to its wider scope of application than redundant computation and server validation. For privacy preservation, we mainly focus on secure aggregation, which is a widely used technique in FL to make server only have access to the average of clients' local updating values.

## 3.1 MOTIVATION

We first analyze the compatibility of existing communication compression methods with secure aggregation (SecAgg) (Bonawitz et al., 2017). SecAgg is usually adopted together with quantization since it requires to operate on a finite field to guarantee the privacy preservation. Traditional quantization methods that represent each coordinate in lower bits can compress gradients in floating point number (32 bits) only up to $1/32$ of the original size. Even with quantization, SecAgg still suffers from heavy communication cost. Thus, sparsification is required to further reduce the communication cost (Ergun et al., 2021). However if we simply combine traditional sparsification methods (e.g., random-$K$ and top-$K$ sparsification) with SecAgg, the random mask in SecAgg will damage the sparsity. Thus, non-Byzantine clients should agree on the non-zero coordinates in order to keep the sparsity in SecAgg.

Then we analyze the compatibility of sparsification with robust aggregation. As previous works (Karimireddy et al., 2021) have shown, to obtain Byzantine robustness, it requires the distances between compressed updates from different clients (a.k.a. dissimilarity between clients) to be small. Specifically, we present the definition of $(\delta, c)$-robust aggregator in Definition 1.

**Definition 1** ($(\delta, c)$-robust aggregator (Karimireddy et al., 2021; 2022)). *Assume constant $\delta \in [0, \frac{1}{2})$ and index set $\mathcal{G} \subseteq [m]$ satisfies $|\mathcal{G}| \geq (1 - \delta)m$. Suppose that we are given $m$ random vectors $\mathbf{v}_1, \ldots, \mathbf{v}_m \in \mathbb{R}^d$ such that $\mathbb{E}\|\mathbf{v}_k - \mathbf{v}_{k'}\|^2 \leq \rho^2$ for any fixed $k, k' \in \mathcal{G}$. $\mathbf{v}_k$ can be arbitrary value if $k \in [m] \setminus \mathcal{G}$. Aggregator $\mathbf{Agg}(\cdot)$ is said to be $(\delta, c)$-robust if the aggregation error $\mathbf{e} = \mathbf{Agg}(\{\mathbf{v}_k\}_{k=1}^m) - \frac{1}{|\mathcal{G}|} \sum_{k \in \mathcal{G}} \mathbf{v}_k$ satisfies that*

$$\mathbb{E}\|\mathbf{e}\|^2 \leq c\delta\rho^2. \tag{2}$$

As shown in previous works (Karimireddy et al., 2022), many widely-used aggregators, such as Krum (Blanchard et al., 2017), geoMed (Chen et al., 2017) and coordinate-wise median (Yin et al., 2018), combined with averaging in buffers (please refer to Section 3.3), satisfy Definition 1. Moreover, $O(\delta\rho^2)$ is the tightest order (Karimireddy et al., 2021). Thus, a compression method which is compatible with robust aggregation should satisfy the condition that the expectation of dissimilarity between clients' updates is kept small after compression. Therefore, we theoretically analyze the expectation of dissimilarity after sparsification. For space saving, we only present the results here. Proof details can be found in Appendix B.

**Theorem 1.** *Let $\{\mathbf{v}_k\}_{k=1}^m$ denote random vectors that satisfy $\mathbb{E}\|\mathbf{v}_k - \mathbf{v}_{k'}\|^2 = (\rho_{k,k'})^2$ and $\mathbb{E}\|\mathbf{v}_k\|^2 = (\mu_k)^2$ for any fixed $k, k' \in \mathcal{G}$. More specifically, $\mathbb{E}[(\mathbf{v}_k)_j - (\mathbf{v}_{k'})_j]^2 = \xi_{k,k',j}(\rho_{k,k'})^2$ and $\mathbb{E}[(\mathbf{v}_k)_j^2] = \zeta_{k,j}(\mu_k)^2$, where $\xi_{k,k',j} > 0$, $\mu_{k,j} > 0$, $\sum_{j \in [d]} \xi_{k,k',j} = 1$ and $\sum_{j \in [d]} \zeta_{k,j} = 1$ for any fixed $k, k' \in \mathcal{G}$. Let $\mathcal{C}(\cdot)$ denote any sparsification operator and $\mathcal{N}_k$ denote the set of non-zero coordinates in $\mathcal{C}(\mathbf{v}_k)$. For any fixed $k, k' \in \mathcal{G}$, we have:*

$$\mathbb{E}\|\mathcal{C}(\mathbf{v}_k) - \mathcal{C}(\mathbf{v}_{k'})\|^2 = (\rho_{k,k'})^2 \cdot \sum_{j \in [d]} \left( \xi_{k,k',j} \Pr[j \in \mathcal{N}_k \cap \mathcal{N}_{k'}] \right)$$
$$+ (\mu_k)^2 \cdot \sum_{j \in [d]} \left( \zeta_{k,j} \Pr[j \in \mathcal{N}_k \setminus \mathcal{N}_{k'}] \right) + (\mu_{k'})^2 \cdot \sum_{j \in [d]} \left( \zeta_{k',j} \Pr[j \in \mathcal{N}_{k'} \setminus \mathcal{N}_k] \right). \tag{3}$$

Please note that when dissimilarity between the $k$-th and the $k'$-th clients is not too large, $(\mu_k)^2$ and $(\mu_{k'})^2$ are usually much larger than $(\rho_{k,k'})^2$. In Equation (3), terms $(\mu_k)^2$ and $(\mu_{k'})^2$ vanish if and only if $\mathcal{N}_k \setminus \mathcal{N}_{k'} = \mathcal{N}_{k'} \setminus \mathcal{N}_k = \emptyset$ with probability 1, which is equivalent to that $\mathcal{N}_k = \mathcal{N}_{k'}$ with probability 1. Furthermore, in order to lower the dissimilarity bewteen any pair of non-Byzantine clients, all non-Byzantine clients should agree on the non-zero coordinates of sparsified vectors. Motivated by the analysis results in these two aspects, we propose the *consensus sparsification*.

## 3.2 CONSENSUS SPARSIFICATION

We introduce the consensus sparsification (ConSpar) method in this section. For simplicity, we assume the hyper-parameter $K$ is a multiple of client number $m$. We use $\mathbf{u}_k^t$ to denote the local memory for error compensation (Stich et al., 2018) on client_$k$ at the $t$-th iteration. Initially, $\mathbf{u}_k^0 = \mathbf{0}$.

Let $\mathbf{g}_k^t$ denote the updates vector to be sent from client_$k$ at the $t$-th iteration. Client_$k$ first generates a coordinate set $\mathcal{T}_k^t$ by top-$\frac{K}{m}$ sparsification criterion. More specifically, $\mathcal{T}_k^t$ contains the coordinates according to the largest $\frac{K}{m}$ absolute values in $\mathbf{g}_k^t$. Then, set $\tilde{\mathcal{T}}_k^t$ is generated by randomly selecting $(\frac{K}{m} - r_k^t)$ elements from $\mathcal{T}_k^t$, where random variable $r_k^t$ follows the binomial distribution $\mathrm{B}(\frac{K}{m}, \alpha)$ $(0 \leq \alpha \leq 1)$. Thus, $|\tilde{\mathcal{T}}_k^t| = \frac{K}{m} - r_k^t$. Then, set $\mathcal{R}_k^t$ is generated by randomly selecting $r_k^t$ elements from $[d] \setminus \tilde{\mathcal{T}}_k^t$. Finally, client_$k$ computes $\mathcal{I}_k^t = \tilde{\mathcal{T}}_k^t \cup \mathcal{R}_k^t$ and sends $\mathcal{I}_k^t$ to server. The main purpose of the operations above is to obfuscate the top-$\frac{K}{m}$ dimension for privacy preservation. Larger $\alpha$ could provide stronger privacy preservation on the coordinates, but may degrade the model accuracy, as we will empirically show in Section 5. The sent coordinates are totally random when $\alpha = 1$.

When server has received $\{\mathcal{I}_k^t\}_{k=1}^m$ from all clients, it decides the coordinate set of sparsified gradients $\mathcal{I}^t = \cup_{k=1}^m \mathcal{I}_k^t$, and broadcasts $\mathcal{I}^t$ to all clients. Finally, each client receives $\mathcal{I}^t$, and computes the sparsified $\tilde{\mathbf{g}}_k^t$ according to it.[2] More specifically, $(\tilde{\mathbf{g}}_k^t)_j = (\mathbf{g}_k^t)_j$ if $j \in \mathcal{I}^t$ and $(\tilde{\mathbf{g}}_k^t)_j = 0$ otherwise, where $(\tilde{\mathbf{g}}_k^t)_j$ denotes the value in the $j$-th coordinate of $\tilde{\mathbf{g}}_k^t$. Since $|\mathcal{I}_k^t| = \frac{K}{m}$, we have $|\mathcal{I}^t| = |\cup_{k=1}^m \mathcal{I}_k^t| \leq \sum_{k=1}^m |\mathcal{I}_k^t| = K$. Please note that clients only need to send $(\tilde{\mathbf{g}}_k^t)_{\mathcal{I}^t}$ to the server since clients and the server have reached an agreement on the non-zero coordinates $\mathcal{I}^t$ of sparsified gradients. More specifically, $(\tilde{\mathbf{g}}_k^t)_{\mathcal{I}^t} = ((\tilde{\mathbf{g}}_k^t)_{j_1}, \ldots, (\tilde{\mathbf{g}}_k^t)_{j_{|\mathcal{I}^t|}})$, where $j_s \in \mathcal{I}^t$ $(s = 1, \ldots, |\mathcal{I}^t|)$ are in ascending order. When server has received $\{(\tilde{\mathbf{g}}_k^t)_{\mathcal{I}^t}\}_{k=1}^m$, it uses robust aggregation to obtain $(\tilde{\mathbf{G}}^t)_{\mathcal{I}^t} = \mathbf{Agg}(\{(\tilde{\mathbf{g}}_k^t)_{\mathcal{I}^t}\}_{k=1}^m)$. Please note that $(\tilde{\mathbf{G}}^t)_{\mathcal{I}^t}$ is still a sparsified vector. Moreover, server is only required to broadcast $(\tilde{\mathbf{G}}^t)_{\mathcal{I}^t}$ since $\mathcal{I}^t$ has already been sent to clients before. Thus, ConSpar is naturally a two-way sparsification method without the need to adopt DoubleSqueeze technique (Tang et al., 2019). Then we analyze the dissimilarity between clients after consensus sparsification. Please note that we do not assume the behaviour of Byzantine clients, which may send arbitrary $\mathcal{I}_k^t$.

**Proposition 1.** *Let $\{\tilde{\mathbf{g}}_k^t\}_{k=1}^m$ denote the consensus sparsification results of vectors $\{\mathbf{g}_k^t\}_{k=1}^m$ and then we have $\mathbb{E}\|\tilde{\mathbf{g}}_k^t - \tilde{\mathbf{g}}_{k'}^t\|^2 \leq \mathbb{E}\|\mathbf{g}_k^t - \mathbf{g}_{k'}^t\|^2$ for any fixed $k, k' \in \mathcal{G}$.*

Proposition 1 indicates that ConSpar will not enlarge the dissimilarity between clients, which is consistent with Theorem 1. Meanwhile, SecAgg can be used in the second communication round and random masks are needed to add on the consensus non-zero coordinates only. Then we analyze the privacy preservation of the mechanism that we use to generate $\mathcal{I}_k^t$ in ConSpar. To begin with, we present the definition of $\epsilon$-differential privacy ($\epsilon$-DP) in Definition 2.

**Definition 2.** *Let $\epsilon > 0$ be a real number. A random mechanism $\mathcal{M}$ is said to provide $\epsilon$-differential privacy if for any two adjacent input datasets $\mathcal{T}_1$ and $\mathcal{T}_2$ and for any subset of possible outputs $\mathcal{S}$:*

$$\Pr[\mathcal{M}(\mathcal{T}_1) \in \mathcal{S}] \leq \exp(\epsilon) \cdot \Pr[\mathcal{M}(\mathcal{T}_2) \in \mathcal{S}].$$

In the mechanism that we use to generate $\mathcal{I}_k^t$ in ConSpar, $\mathcal{T}_1$ and $\mathcal{T}_2$ are the top-$\frac{K}{m}$ coordinate sets. Definition 2 leaves the definition of adjacent datasets open. In this work, coordinate sets $\mathcal{T}_1$ and $\mathcal{T}_2$ that satisfy $\mathcal{T}_1, \mathcal{T}_2 \subseteq [d]$ and $|\mathcal{T}_1| = |\mathcal{T}_2| = \frac{K}{m}$ are defined to be adjacent if $\mathcal{T}_1$ and $\mathcal{T}_2$ differ only on one element. Liu et al. (2020) provides DP guarantee for sparsification methods with only one selected coordinate. Our definition is more general and includes the one coordinate special case where $|\mathcal{T}_1| = |\mathcal{T}_2| = 1$. Now we show that the coordinate generation mechanism provides $\epsilon$-DP.

**Theorem 2.** *For any $\alpha \in (0, 1]$, the mechanism in consensus sparsification that takes the set of top coordinates $\mathcal{T}_k^t$ as an input and outputs $\mathcal{I}_k^t$ provides $\left(\ln\left(\frac{(1+\alpha) \cdot \frac{K}{m}(d - \frac{K}{m} + 1)}{2\alpha}\right)\right)$-differential privacy.*

Finally, we analyze the communication complexity of ConSpar. Clients need to send candidate coordinate set $\mathcal{I}_k^t$, receive $\mathcal{I}^t$, send local gradient in the form of $(\tilde{\mathbf{g}}_k^t)_{\mathcal{I}^t}$, and then receive $(\tilde{\mathbf{G}}^t)_{\mathcal{I}^t}$ in each iteration. Thus, each client needs to communicate no more than $(\frac{K}{m} + K)$ integers and $2K$ floating point numbers in each iteration. When each integer or floating point number is represented

---

[2]We use tilde to denote sparse vectors in this paper for easy distinguishment.

by 32 bits (4 bytes), the total communication load of each client is no more than $(96 + \frac{32}{m})K$ bits in each iteration. The communication load is not larger than that of vanilla top-$K$ sparsification, in which $4 \times 32K = 128K$ bits are transmitted in each iteration. Meanwhile, although ConSpar requires two communication rounds, the extra communication round is acceptable. For one reason, there is little computation between the two rounds, which will not significantly increase the risk of client disconnection during the aggregation process. For another reason, the cost of the extra communication round is negligible when combined with SecAgg since SecAgg already requires multiple communication rounds and can deal with offline clients.

### 3.3   FEDREP

As we have shown, ConSpar is compatible with each of robust aggregation and SecAgg. However, robust aggregation and SecAgg can not be simply applied together since SecAgg is originally designed for linear aggregation (such as summation and averaging) while robust aggregation is usually non-linear. This is also known as the tension between robustness and privacy (Kairouz et al., 2021). Buffers on server are widely studied in Byzantine-robust machine learning (Karimireddy et al., 2022; Velicheti et al., 2021; Wang et al., 2020; Yang & Li, 2021), which can be used to make such a trade-off between robustness and privacy. We also introduce buffers in FedREP. The details of FedREP are illustrated in Algorithm 1 and Algorithm 2 in Appendix A.

Let integer $s$ denote the buffer size. For simplicity, we assume client number $m$ is a multiple of buffer size $s$ and hence there are $\frac{m}{s}$ buffers on server. At the beginning of the $t$-th global iteration, each client_$k$ locally trains model using optimization algorithm $\mathcal{A}$ and training instances $\mathcal{D}_k$ based on $\mathbf{w}^t$ and obtain model parameter $\mathbf{w}_k^{t+1} = \mathcal{A}(\mathbf{w}^t; \mathcal{D}_k)$. The update to be sent is $\mathbf{g}_k^t = \mathbf{u}_k^t + (\mathbf{w}^t - \mathbf{w}_k^{t+1})$, where $\mathbf{u}_k^t$ is the local memory for error compensation with $\mathbf{u}_k^0 = \mathbf{0}$. Then client_$k$ generates coordinate set $\mathcal{I}_k^t$ by consensus sparsification (please see Section 3.2), and sends $\mathcal{I}_k^t$ to the server.

When server receives all clients' suggested coordinate sets, it broadcasts $\mathcal{I}^t = \cup_{k=1}^m \mathcal{I}_k^t$, which is the set of coordinates to be transmitted in the current iteration, to all clients. In addition, server will randomly assign a buffer for each client. More specifically, server randomly picks a permutation $\pi$ of $[m]$ and assign buffer $\mathbf{b}_l$ to clients $\{\pi(ls + k)\}_{k=1}^s$ $(l = 0, 1, \ldots, \frac{m}{s} - 1)$. Then for each buffer, $\mathbf{b}_l = \frac{1}{s} \sum_{k=1}^s (\tilde{\mathbf{g}}_{\pi(ls+k)}^t)_{\mathcal{I}^t}$ is obtained by secure aggregation[3] and global update $(\tilde{\mathbf{G}}^t)_{\mathcal{I}^t} = \mathbf{Agg}(\{\mathbf{b}_l\}_{l=1}^{m/s})$ is obtained by robust aggregation among buffers. During this time, clients could update the local memory for error compensation by computing $\mathbf{u}_k^{t+1} = \mathbf{g}_k^t - \tilde{\mathbf{g}}_k^t$. Finally, $(\tilde{\mathbf{G}}^t)_{\mathcal{I}^t}$ is broadcast to all clients for global updating by $\mathbf{w}^{t+1} = \mathbf{w}^t - \tilde{\mathbf{G}}^t$.

We have noticed that the consensus sparsification is similar to the cyclic local top-$K$ sparsification (CLT-$K$) (Chen et al., 2020), where all clients' non-zero coordinates are decided by one client in each communication round. However, there are significant differences between these two sparsification methods. CLT-$K$ is designed to be compatible with all-reduce while consensus sparsification is designed to be compatible with robust aggregation and SecAgg in FL. In addition, when there are Byzantine clients, CLT-$K$ does not satisfy the $d'$-contraction property since Byzantine clients may purposely send wrong non-zero coordinates. Meanwhile, some works (Karimireddy et al., 2022) show that averaging in groups before robust aggregation (as adopted in FedREP) can help to enhance the robustness of learning methods on heterogeneous datasets. We will further explore this aspect in future work since it is beyond the scope of this paper.

Finally, we would like to discuss more about privacy. In the ideal case, server would learning nothing more than the aggregated result. However, obtaining the ideal privacy-preserving property itself would be challenging, and even more so when we attempt to simultaneously guarantee Byzantine robustness and communication efficiency (Kairouz et al., 2021). In FedREP, server has access to the partially aggregated mean $\mathbf{b}_l$ and the coordinate set $\mathcal{I}_k^t$. However, as far as we know, the risk of privacy leakage increased by the two kinds of information is limited. Server does not know the momentum sent from each single client, and only has access to the coordinate set $\mathcal{I}_k^t$ without knowing the corresponding values or even the signs. Although it requires further work to study how much information can be obtained from the coordinates, to the best of our knowledge, there are almost no exsiting methods that can recover the training data only based on the coordinates.

---

[3]Random quantization can be simply adopted before secure aggregation to make the values on a finite field for more privacy preservation. However for simplicity, we do not include it in the description here.

## 4 CONVERGENCE

In this section, we theoretically prove the convergence of FedREP. Due to limited space, proof details are in Appendix B. Firstly, we present the definition of $d'$-contraction operator (Stich et al., 2018).

**Definition 3** ($d'$-contraction). $\mathcal{C} : \mathbb{R}^d \to \mathbb{R}^d$ is called a $d'$-contraction operator $(0 < d' \leq d)$ if

$$\mathbb{E}\|\mathbf{x} - \mathcal{C}(\mathbf{x})\|^2 \leq (1 - d'/d)\|\mathbf{x}\|^2, \quad \forall \mathbf{x} \in \mathbb{R}^d. \tag{4}$$

The $d'$-contraction property of consensus sparsification is shown in Proposition 2.

**Proposition 2.** *If the fraction of Byzantine clients is not larger than $\delta$ $(0 \leq \delta < \frac{1}{2})$, consensus sparsification is a $d'_{cons}$-contraction operator, where $d'_{cons} = d(1 - e^{-\frac{\alpha K[(1-\delta)m-1]}{md}}) + \frac{K}{m}e^{-\frac{\alpha K[(1-\delta)m-1]}{md}}$.*

Therefore, the existing convergence results of $d'$-contraction operator (Stich et al., 2018) can be directly applied to consensus sparsification when there is no Byzantine attack. Then we theoretically analyze the convergence of FedREP. For simplicity in the analysis, we consider the secure aggregation and the robust aggregation on server as a unit secure robust aggregator, which is denoted by $\mathbf{SRAgg}(\cdot)$. Therefore, $\tilde{\mathbf{G}}^t = \mathbf{SRAgg}(\{\tilde{\mathbf{g}}_k^t\}_{k=1}^m)$. The assumptions are listed below.

**Assumption 1** (Byzantine setting). *The fraction of Byzantine clients is not larger than $\delta$ $(0 \leq \delta < \frac{1}{2})$ and the secure robust aggregator $\mathbf{SRAgg}(\cdot)$ is $(\delta, c)$-robust with constant $c \geq 0$.*

**Assumption 2** (Lower bound). *$F(\mathbf{w})$ is bounded below: $\exists F^* \in \mathbb{R}, F(\mathbf{w}) \geq F^*, \forall \mathbf{w} \in \mathbb{R}^d$.*

**Assumption 3** ($L$-smoothness). *Global loss function $F(\mathbf{w})$ is differentiable and $L$-smooth: $\|\nabla F(\mathbf{w}) - \nabla F(\mathbf{w}')\| \leq L\|\mathbf{w} - \mathbf{w}'\|, \forall \mathbf{w}, \mathbf{w}' \in \mathbb{R}^d$.*

**Assumption 4** (Bounded bias). *$\forall k \in \mathcal{G}$, we have $\mathbb{E}[\nabla f_{i_k^t}(\mathbf{w})] = \nabla F_k(\mathbf{w})$ and there exists $B \geq 0$ such that $\|\nabla F_k(\mathbf{w}) - \nabla F(\mathbf{w})\| \leq B, \forall \mathbf{w} \in \mathbb{R}^d$.*

**Assumption 5** (Bounded gradient). *$\forall k \in \mathcal{G}$, stochastic gradient $\nabla f_{i_k^t}(\mathbf{w})$ has bounded expectation: $\exists D \in \mathbb{R}_+$, such that $\|\nabla F_k(\mathbf{w})\| \leq D, \forall \mathbf{w} \in \mathbb{R}^d$.*

Assumption 1 is common in Byzantine-robust distributed machine learning, which is consistent with previous works (Karimireddy et al., 2022). The rest assumptions are common in distributed stochastic optimization. Assumption 5 is widely used in the analysis of gradient compression methods with error compensation. We first analyze the convergence for a special case of FedREP where the training algorithm $\mathcal{A}$ is local SGD with learning rate $\eta$ and interval $I$. Specifically, $\mathbf{w}_k^{t+1}$ is computed by the following process: (i) $\mathbf{w}_k^{t+1,0} = \mathbf{w}^t$; (ii) $\mathbf{w}_k^{t+1,j+1} = \mathbf{w}_k^{t+1,j} - \eta \cdot \nabla f_{i_k^{t,j}}(\mathbf{w}_k^{t+1,j})$, $j = 0, 1, \ldots, I-1$; (iii) $\mathbf{w}_k^{t+1} = \mathbf{w}_k^{t+1,I}$, where $i_k^{t,j}$ is uniformly sampled from $\mathcal{D}_k$. Assumption 6 is made for this case.

**Assumption 6** (Bounded variance). *Stochastic gradient $\nabla f_{i_k^t}(\mathbf{w})$ is unbiased with bounded variance: $\mathbb{E}[\nabla f_{i_k^t}(\mathbf{w})] = F_k(\mathbf{w})$ and $\exists \sigma \in \mathbb{R}_+$, such that $\mathbb{E}\|\nabla f_{i_k^t}(\mathbf{w}) - \nabla F_k(\mathbf{w})\|^2 \leq \sigma^2, \forall \mathbf{w} \in \mathbb{R}^d, \forall k \in \mathcal{G}$.*

According to Assumption 5 and 6, the second order moment of stochastic gradient $\nabla f_{i_k^t}(\mathbf{w})$ is bounded by $(D^2 + \sigma^2)$. Let $\mathbf{u}^t = \frac{1}{|\mathcal{G}|}\sum_{k \in \mathcal{G}} \mathbf{u}_k^t$ and let $\mathbf{e}^t = \mathbf{SRAgg}(\{\tilde{\mathbf{g}}_k^t\}_{k=1}^m) - \frac{1}{|\mathcal{G}|}\sum_{k \in \mathcal{G}} \tilde{\mathbf{g}}_k^t$ denote the aggregation error . We first show that $\mathbb{E}\|\mathbf{u}_k^t\|^2$ and $\mathbb{E}\|\mathbf{e}^t\|^2$ are both bounded above.

**Lemma 1.** *Under Assumption 1, 2, 3, 4, 5 and 6, let constant $H = d/d'_{cons}$ and take learning rate $\eta_t = \eta > 0$, we have $\mathbb{E}\|\mathbf{u}_k^t\|^2 \leq 4H^2I^2(D^2 + \sigma^2) \cdot \eta^2, \quad \forall k \in \mathcal{G}$.*

**Lemma 2.** *Under the same conditions in Lemma 1, we have $\mathbb{E}\|\mathbf{e}^t\|^2 \leq 8c\delta I^2(4H^2+1)(D^2+\sigma^2) \cdot \eta^2$.*

Based on Lemma 1 and Lemma 2, we have the following theorem.

**Theorem 3.** *For FedREP, under the same conditions in Lemma 1 and Lemma 2, we have:*

$$\frac{1}{T}\sum_{t=0}^{T-1} \mathbb{E}\|\nabla F(\mathbf{w}^t)\|^2 \leq \frac{2[F(\hat{\mathbf{w}}^0) - F^*]}{\eta IT} + \eta\gamma_1 + \eta^2\gamma_2 + \Delta, \tag{5}$$

*where*

$$\gamma_1 = 2IL \cdot [2(1 - I^{-1})LD + 2HD\sqrt{D^2 + \sigma^2} + (D^2 + \sigma^2) + 8c\delta(4H^2 + 1)(D^2 + \sigma^2)],$$

$$\gamma_2 = 8H^2I^2L^2(D^2 + \sigma^2) \qquad and \qquad \Delta = 2BD + 4\sqrt{2c\delta(4H^2 + 1)(D^2 + \sigma^2)}D.$$

When taking $\eta = O(1/\sqrt{T})$, Theorem 3 guarantees that FedREP has a convergence rate of $O(1/\sqrt{T})$ with an extra error $\Delta$, which consists of two terms. The first term $2BD$ comes from the bias of stochastic gradients, which reflects the degree of heterogeneity between clients. The term vanishes in i.i.d. cases where $B = 0$. The second term $4\sqrt{2c\delta(4H^2 + 1)(D^2 + \sigma^2)}D$ comes from the aggregation error. The term vanishes when there is no Byzantine client ($\delta = 0$). Namely, the extra error $\Delta$ vanishes in i.i.d. cases without Byzantine clients. Then we analyze the convergence of FedREP with general local training algorithms that satisfy Assumption 7, which illustrates two important properties of a training algorithm.

**Assumption 7.** *Let* $\mathbf{w}' = \mathcal{A}(\mathbf{w}; \mathcal{D}_k)$. *There exist constants* $\eta_{\mathcal{A}} > 0$, $A_1 \geq 0$ *and* $A_2 > 0$ *such that local training algorithm* $\mathcal{A}$ *satisfies* $\|\mathbb{E}[\mathbf{G}_{\mathcal{A}}(\mathbf{w}; \mathcal{D}_k)] - \nabla F_k(\mathbf{w})\| \leq A_1$ *and* $\mathbb{E}\|\mathbf{G}_{\mathcal{A}}(\mathbf{w}; \mathcal{D}_k)\|^2 \leq (A_2)^2$, *where* $\mathbf{G}_{\mathcal{A}}(\mathbf{w}; \mathcal{D}_k) = (\mathbf{w} - \mathbf{w}')/\eta_{\mathcal{A}}$, $\forall k \in [m]$.

In Assumption 7, $\mathbf{G}_{\mathcal{A}}(\mathbf{w}; \mathcal{D}_k)$ can be deemed as an estimation of gradient $\nabla F_k(\mathbf{w})$ by algorithm $\mathcal{A}$ with bounded bias $A_1$ and bounded second order moment $(A_2)^2$. The expectation appears due to the randomness in algorithm $\mathcal{A}$. Many widely used algorithms satisfy Assumption 7. For vanilla SGD, let $\eta_{\mathcal{A}}$ be the learning rate and $\mathbf{G}_{\mathcal{A}}(\mathbf{w}; \mathcal{D}_k)$ is exactly the stochastic gradient. Thus, we have $A_1 = 0$ and $(A_2)^2 = D^2 + \sigma^2$ under Assumption 5 and 6. Moreover, previous works (Allen-Zhu et al., 2020; El-Mhamdi et al., 2020; Karimireddy et al., 2021) have shown that using history information such as momentum is necessary in Byzantine-robust machine learning. We show that local momentum SGD also satisfies Assumption 7 in Proposition 3 in Appendix B.

**Theorem 4.** *Let constant* $H = d/d'_{cons}$. *For FedREP, under Assumption 1, 2, 3, 4, 5 and 7, we have:*

$$\frac{1}{T} \sum_{t=0}^{T-1} \mathbb{E}\|\nabla F(\mathbf{w}^t)\|^2 \leq \frac{2[F(\hat{\mathbf{w}}^0) - F^*]}{\eta_{\mathcal{A}} T} + \eta_{\mathcal{A}} \gamma_{\mathcal{A},1} + (\eta_{\mathcal{A}})^2 \gamma_{\mathcal{A},2} + \Delta_{\mathcal{A}}, \quad (6)$$

*where*

$$\gamma_{\mathcal{A},1} = 2(A_2)^2 L + 4HA_2 DL + 16c\delta(4H^2 + 1)(A_2)^2 L,$$
$$\gamma_{\mathcal{A},2} = 8H^2(A_2)^2 L^2 \quad and \quad \Delta_{\mathcal{A}} = 2A_1 D + 2BD + 4\sqrt{2c\delta(4H^2 + 1)}A_2 D.$$

Compared to the error $\Delta$ in Theorem 3, there is an extra term $2A_1 D$ in $\Delta_{\mathcal{A}}$, which is caused by the bias of gradient estimation in algorithm $\mathcal{A}$. Meanwhile, we would like to point out that Theorem 4 provides convergence guarantee for general algorithms. For some specific algorithm, tighter upper bounds may be obtained by adopting particular analysis technique. We will leave it for future work since we mainly focus on a general framework in this paper.

## 5 EXPERIMENT

In this section, we evaluate the performance of FedREP and baselines on image classification task. Each method is evaluated on CIFAR-10 dataset (Krizhevsky et al., 2009) with a widely used deep learning model ResNet-20 (He et al., 2016). Training instances are equally and uniformly distributed to each client. All experiments in this work are conducted by PyTorch on a distributed platform with dockers. More specifically, we set 32 dockers as clients, among which 7 clients are Byzantine. One extra docker is set to be the server. Each docker is bound to an NVIDIA Tesla K80 GPU. Unless otherwise stated, we set local training algorithm $\mathcal{A}$ to be local momentum SGD with momentum hyper-parameter $\beta = 0.9$ (see Equation (182) in Appendix B) for FedREP. We run each method in the same environment for 120 epochs. Initial learning rate is chosen from $\{0.1, 0.2, 0.5, 1.0, 2.0, 5.0\}$. At the 80-th epoch, learning rate will be multiplied by $0.1$ as suggested in (He et al., 2016). The best top-1 accuracy w.r.t. epoch is used as final metrics. We test each method under bit-flipping attack, 'A Little is Enough' (ALIE) attack (Baruch et al., 2019) and 'Fall of Empires' (FoE) attack (Xie et al., 2020a). The updates sent by Byzantine clients with bit-flipping attack are in the opposite direction. ALIE and FoE are two omniscient attacks, where attackers are assumed to know the updates on all clients and use them for attack. We set attack magnitude hyper-parameter to be $0.5$ for FoE attack. For FedREP, we test the performance when the robust aggregator is geometric median (geoMed) (Chen et al., 2017), coordinate-wise trimmed-mean (TMean) (Yin et al., 2018) and centered-clipping (CClip) (Karimireddy et al., 2021), respectively. More specifically, we adopt Weiszfeld's algorithm (Pillutla et al., 2019) with iteration number set to be $5$ for computing geoMed. The trimming fraction in TMean is set to $7/16$. For CClip, we set clipping radius to be $0.5$ and iteration number to be $5$. Batch size is set to be $25$.

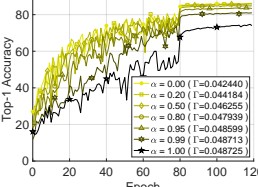 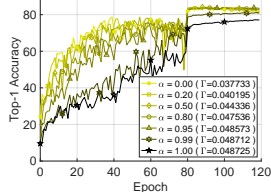 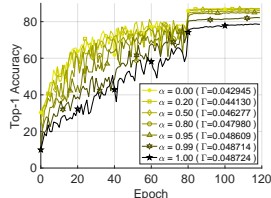

Figure 1: Top-1 accuracy w.r.t. epochs of FedREP with CClip when there are 7 Byzantine clients under bit-flipping attack (left), ALIE attack (middle) and FoE attack (right).

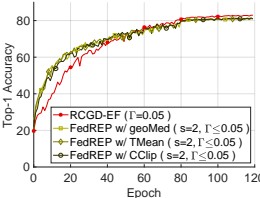 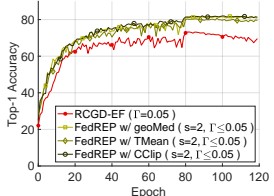 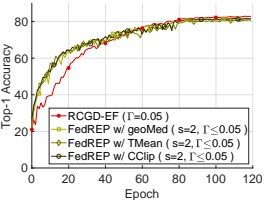

Figure 2: Top-1 accuracy w.r.t. epochs of FedREP and RCGD-EF when there are 7 Byzantine clients under bit-flipping attack (left), ALIE attack (middle) and FoE attack (right).

We first empirically evaluate the effect of $\alpha$ on the performance of FedREP. We set $I = 1$, $s = 2$ and $K = 0.05d$. We compare the performance of FedREP when $\alpha = 0, 0.2, 0.5, 0.8, 0.95, 0.99$ and $1$. As illustrated in Figure 1, the performance of FedREP with CClip changes little when $\alpha$ ranges from 0 to 0.95. Final accuracy will decrease rapidly when $\alpha$ continues to increase. A possible reason is that some coordinates could grow very large in local error compensation memory when $\alpha$ is near 1. More results of FedREP with geoMed and TMean are presented in Appendix C.1. Since the effect of $\alpha$ is small when $0 \leq \alpha \leq 0.95$, we set $\alpha = 0$ in the following experiments. In addition, as the empirical results in Appendix C.2 show, Byzantine attacks on coordinates have little effect on the performance of FedREP. Thus, we assume no attacks on the coordinates in the following experiments.

Then we compare FedREP with a Byzantine-robust communication-efficient baseline called Robust Compressed Gradient Descent with Error Feedback (RCGD-EF) (Ghosh et al., 2021). For fairness, we set the compression operator $\mathcal{Q}(\cdot)$ in RCGD-EF to be top-$K$ sparsification, and set $\Gamma = 0.05$, which is the ratio of transmitted dimension number to total dimension number, for both FedREP and RCGD-EF. Local updating interval $I$ is set to 5 for each method. As illustrated in Figure 2, FedREP has comparable performance to RCGD-EF under bit-flipping and FoE attack, but outperforms RCGD-EF under ALIE attack. Meanwhile, FedREP is naturally a two-way sparsification method while RCGD-EF is not. Moreover, FedREP provides extra privacy preservation compared to RCGD-EF.

Due to limited space, more empirical results are presented in the appendices. Empirical results in Appendix C.3 show the effect of momentum hyper-parameter $\beta$. Empirical results in Appendix C.4 show that FedREP can significantly outperform the communication-efficient privacy-preserving baseline SparseSecAgg (Ergun et al., 2021). Empirical results in Appendix C.5 show that compared with the Byzantine-robust privacy-preserving baseline SHARE (Velicheti et al., 2021), FedREP has comparable convergence rate and accuracy with much smaller communication cost.

## 6 CONCLUSION

In this paper, we theoretically analyze the tension among Byzantine robustness, communication efficiency and privacy preservation in FL. Motivated by the analysis results, we propose a novel Byzantine-robust, communication-efficient and privacy-preserving FL framework called *FedREP*. Theoretical guarantees for the Byzantine robustness and the convergence of FedREP are provided. Empirical results show that FedREP can significantly outperform communication-efficient privacy-preserving baselines. Furthermore, compared with Byzantine-robust communication-efficient baselines, FedREP can achieve comparable accuracy with an extra advantage of privacy preservation.

**Reproducibility Statement.** In this work, we empirically test the performance of FedREP and the baselines on the public dataset CIFAR-10 (Krizhevsky et al., 2009) with the widely used deep learning model ResNet-20 (He et al., 2016). Common settings for all the experiments in this work are presented at the beginning of Section 5. More settings for each single experiment are presented along with the empirical results in Section 5 and Appendix C. In addition, we provide the core part of our code in the supplementary material. All the proof details for the theoretical results in this work can be found in Appendix B.

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

# A  DETAILS OF FEDREP

The detailed algorithms of FedREP on server and clients are illustrated in Algorithm 1 and Algorithm 2, respectively.

---

**Algorithm 1** FedREP (Server)

---

**Input:** client number $m$, iteration number $T$, buffer size $s$, robust aggregator $\mathbf{Agg}(\cdot)$;

**for** $t = 0$ **to** $T - 1$ **do**
    Receive $\{\mathcal{I}_k^t\}_{k=1}^m$ from all clients and compute $\mathcal{I}^t = \cup_{k=1}^m \mathcal{I}_k^t$;
    Broadcast $\mathcal{I}^t$ to all clients;
    Pick a random permutation $\pi$ of $[m]$;
    Assign buffer $\mathbf{b}_l$ to clients $\{\pi(ls + k)\}_{k=1}^s$ for $l = 0, 1, \ldots, \frac{m}{s} - 1$;
    **for** $l = 0$ **to** $\frac{m}{s} - 1$ **do**
        Obtain $\mathbf{b}_l = \frac{1}{s} \sum_{k=1}^s (\tilde{\mathbf{g}}_{\pi(ls+k)}^t)_{\mathcal{I}^t}$ via SecAgg protocol;
    **end for**
    Compute: $(\tilde{\mathbf{G}}^t)_{\mathcal{I}^t} = \mathbf{Agg}(\{\mathbf{b}_l\}_{l=1}^{m/s})$;
    Broadcast $(\tilde{\mathbf{G}}^t)_{\mathcal{I}^t}$ to all clients;
**end for**

---

**Algorithm 2** FedREP (Client)

---

**Input:** client number $m$, iteration number $T$,
        local optimization algorithm $\mathcal{A}$ and sparsification size $K$;
**Initialization:** model parameter $\mathbf{w}^0$ and local memory $\mathbf{u}_k^0 = \mathbf{0}$;
**for** $t = 0$ **to** $T - 1$ **do**

    */* Locally train learning model and update error compensation */*
    Locally train learning model and obtain parameter $\mathbf{w}_k^{t+1} = \mathcal{A}(\mathbf{w}^t; \mathcal{D}_k)$;
    Compute $\mathbf{g}_k^t = \mathbf{u}_k^t + (\mathbf{w}^t - \mathbf{w}_k^{t+1})$;

    */* Two-stage aggregation with sparsification */*
    Generate coordinate set $\mathcal{I}_k^t$ by consensus sparsification (see Section 3.2 in the main text);
    Send $\mathcal{I}_k^t$ to the server;
    Receive coordinate set $\mathcal{I}^t$ and assigned buffer number $l$ from the server;
    Compute $(\tilde{\mathbf{g}}_k^t)_{\mathcal{I}^t}$ and send it to the assigned buffer $\mathbf{b}_l$ via SecAgg protocol;
    Update memory: $\mathbf{u}_k^{t+1} = \mathbf{g}_k^t - \tilde{\mathbf{g}}_k^t$;
    Receive $(\tilde{\mathbf{G}}^t)_{\mathcal{I}^t}$ from the server and recover $\tilde{\mathbf{G}}^t$ according to $\mathcal{I}^t$;

    */* Update model parameters*/*
    Update parameters: $\mathbf{w}^{t+1} = \mathbf{w}^t - \tilde{\mathbf{G}}^t$;
**end for**
Output model parameter $\mathbf{w}^T$;

---

# B    PROOF DETAILS

In this section, we present the proof details of the theoretical results in the paper.

## B.1    PROOF OF THEOREM 1

*Proof.* For any fixed $k, k' \in [m]$,

$$\mathbb{E}\|\mathcal{C}(\mathbf{v}_k) - \mathcal{C}(\mathbf{v}_{k'})\|^2$$

$$=\mathbb{E}\left[\sum_{j \in \mathcal{N}_k \cap \mathcal{N}_{k'}} [(\mathbf{v}_k)_j - (\mathbf{v}_{k'})_j]^2\right] + \mathbb{E}\left[\sum_{j \in \mathcal{N}_k \setminus \mathcal{N}_{k'}} (\mathbf{v}_k)_j^2\right] + \mathbb{E}\left[\sum_{j \in \mathcal{N}_{k'} \setminus \mathcal{N}_k} (\mathbf{v}_{k'})_j^2\right] \tag{7}$$

$$=\mathbb{E}\left[\sum_{j \in \mathcal{N}_k \cap \mathcal{N}_{k'}} \xi_{k,k',j}(\rho_{k,k'})^2\right] + \mathbb{E}\left[\sum_{j \in \mathcal{N}_k \setminus \mathcal{N}_{k'}} \zeta_{k,j}(\mu_k)^2\right] + \mathbb{E}\left[\sum_{j \in \mathcal{N}_{k'} \setminus \mathcal{N}_k} \zeta_{k',j}(\mu_{k'})^2\right] \tag{8}$$

$$= \sum_{j \in [d]} \xi_{k,k',j}(\rho_{k,k'})^2 \cdot \Pr[j \in \mathcal{N}_k \cap \mathcal{N}_{k'}]$$

$$+ \sum_{j \in [d]} \zeta_{k,j}(\mu_k)^2 \cdot \Pr[j \in \mathcal{N}_k \setminus \mathcal{N}_{k'}] + \sum_{j \in [d]} \zeta_{k',j}(\mu_{k'})^2 \cdot \Pr[j \in \mathcal{N}_{k'} \setminus \mathcal{N}_k] \tag{9}$$

$$=(\rho_{k,k'})^2 \cdot \sum_{j \in [d]} \left(\xi_{k,k',j} \Pr[j \in \mathcal{N}_k \cap \mathcal{N}_{k'}]\right)$$

$$+ (\mu_k)^2 \cdot \sum_{j \in [d]} \left(\zeta_{k,j} \Pr[j \in \mathcal{N}_k \setminus \mathcal{N}_{k'}]\right) + (\mu_{k'})^2 \cdot \sum_{j \in [d]} \left(\zeta_{k',j} \Pr[j \in \mathcal{N}_{k'} \setminus \mathcal{N}_k]\right). \tag{10}$$

$\square$

## B.2    PROOF OF PROPOSITION 1

*Proof.* Let $\mathcal{I}^t$ denote the set of non-zero coordinates after consensus sparsification. In general cases, for any fixed $k, k' \in [m]$, we have:

$$\mathbb{E}\|\tilde{\mathbf{g}}_k^t - \tilde{\mathbf{g}}_{k'}^t\|^2 = \sum_{j \in \mathcal{I}^t} \mathbb{E}[(\mathbf{g}_k^t)_j - (\mathbf{g}_{k'}^t)_j]^2 \tag{11}$$

$$\leq \sum_{j \in [d]} \mathbb{E}[(\mathbf{g}_k^t)_j - (\mathbf{g}_{k'}^t)_j]^2 \tag{12}$$

$$= \mathbb{E}\|\mathbf{g}_k^t - \mathbf{g}_{k'}^t\|^2. \tag{13}$$

$\square$

## B.3    PROOF OF THEOREM 2

*Proof.* Let $\mathcal{M}$ be the the mechanism in consensus sparsification that takes the set of top coordinates $\mathcal{T}$ as an input and outputs a random coordinate set. $\mathcal{T}_1, \mathcal{T}_2 \subseteq [d]$ are two arbitrary adjacent input coordinate sets that satisfy $|\mathcal{T}_1| = |\mathcal{T}_2| = \frac{K}{m}$ and $\mathcal{S}$ is any subset of possible outputs of $\mathcal{M}$. When $\mathcal{S}$ is empty, $\Pr[\mathcal{M}(\mathcal{T}_1) \in \mathcal{S}] = \Pr[\mathcal{M}(\mathcal{T}_2) \in \mathcal{S}] = 0$. Thus, for any $\epsilon > 0$, we have:

$$0 = \Pr[\mathcal{M}(\mathcal{T}_1) \in \mathcal{S}] \leq \exp(\epsilon) \cdot \Pr[\mathcal{M}(\mathcal{T}_2) \in \mathcal{S}] = 0. \tag{14}$$

Without loss of generality, we suppose that $\mathcal{S}$ is non-empty. For any $\mathcal{I} \in \mathcal{S}$, let $|\mathcal{T}_1 \cap \mathcal{I}| = U_1$ and $|\mathcal{T}_2 \cap \mathcal{I}| = U_2$. $\mathcal{T}_1$ and $\mathcal{T}_2$ only differ on one element since they are adjacent. Thus, we have $U_2 = U_1 - 1, U_1$ or $U_1 + 1$. Set $\tilde{\mathcal{T}}_1$ is generated by randomly selecting $(\frac{K}{m} - r_1)$ elements from $\mathcal{T}_1$, where $r_1$ follows the binomial distribution $\Pr(\frac{K}{m}, \alpha)$. Thus, $\forall i = 0, 1, \dots, K/m$,

$$\Pr[r_1 = i] = \binom{K/m}{i} \alpha^i (1 - \alpha)^{K/m - i}. \tag{15}$$

To obtain $\mathcal{I}$ as the final output, only the elements in $\mathcal{T}_1 \cap \mathcal{I}$ can be selected. Thus, $r_1$ should equal or be larger than $|\mathcal{T}_1 \setminus \mathcal{I}| = \frac{K}{m} - U_1$. Furthermore, for $r_1 \geq \frac{K}{m} - U_1$, the probability that all elements are selected from $\mathcal{T}_1 \cap \mathcal{I}$ is $\binom{U_1}{K/m - r_1} / \binom{K/m}{K/m - r_1} = \binom{U_1}{r_1 - (K/m - U_1)} / \binom{K/m}{r_1}$. Finally, the $r_1$ elements in $\mathcal{I} \setminus \tilde{\mathcal{T}}_1$ should be selected from $[d] \setminus \tilde{\mathcal{T}}_1$, of which the probability is $1/\binom{d - K/m + r_1}{r_1}$ since $|[d] \setminus \tilde{\mathcal{T}}_1| = d - (K/m - r_1) = d - K/m + r_1$. Thus, we have:

$$\Pr[\mathcal{M}(\mathcal{T}_1) = \mathcal{I}]$$

$$= \sum_{i=K/m-U_1}^{K/m} \left\{ \Pr[r_1 = i] \times \frac{\binom{U_1}{i-(K/m-U_1)}}{\binom{K/m}{i}} \times \frac{1}{\binom{d-K/m+i}{i}} \right\} \tag{16}$$

$$= \sum_{i=K/m-U_1}^{K/m} \left\{ \left[ \binom{K/m}{i} \alpha^i (1-\alpha)^{K/m-i} \right] \times \frac{\binom{U_1}{i-(K/m-U_1)}}{\binom{K/m}{i}} \times \frac{1}{\binom{d-K/m+i}{i}} \right\} \tag{17}$$

$$= \sum_{i=K/m-U_1}^{K/m} \left\{ \alpha^i (1-\alpha)^{K/m-i} \binom{U_1}{i-(K/m-U_1)} \times \frac{1}{\binom{d-K/m+i}{i}} \right\} \tag{18}$$

$$= \sum_{i=0}^{U_1} \left\{ \alpha^{K/m-i} (1-\alpha)^i \binom{U_1}{U_1-i} \times \frac{1}{\binom{d-i}{K/m-i}} \right\} \tag{19}$$

$$= \alpha^{K/m} \cdot \sum_{i=0}^{U_1} \left\{ \left( \frac{1-\alpha}{\alpha} \right)^i \binom{U_1}{i} \times \frac{1}{\binom{d-i}{d-K/m}} \right\}. \tag{20}$$

Thus, $\Pr[\mathcal{M}(\mathcal{T}_1) = \mathcal{I}]$ is monotonically increasing with respect to $U_1$. Similarly,

$$\Pr[\mathcal{M}(\mathcal{T}_2) = \mathcal{I}] = \alpha^{K/m} \cdot \sum_{i=0}^{U_2} \left\{ \left( \frac{1-\alpha}{\alpha} \right)^i \binom{U_2}{i} \times \frac{1}{\binom{d-i}{d-K/m}} \right\}, \tag{21}$$

which is monotonically increasing with respect to $U_2$. Thus, $\frac{\Pr[\mathcal{M}(\mathcal{T}_1)=\mathcal{I}]}{\Pr[\mathcal{M}(\mathcal{T}_2)=\mathcal{I}]}$ takes the maximum value when $U_1 = U_2 + 1$. Therefore,

$$\frac{\Pr[\mathcal{M}(\mathcal{T}_1) = \mathcal{I}]}{\Pr[\mathcal{M}(\mathcal{T}_2) = \mathcal{I}]} \leq \frac{\alpha^{K/m} \cdot \sum_{i=0}^{U_2+1} \left[ \left( \frac{1-\alpha}{\alpha} \right)^i \binom{U_2+1}{i} \times \frac{1}{\binom{d-i}{d-K/m}} \right]}{\alpha^{K/m} \cdot \sum_{i=0}^{U_2} \left[ \left( \frac{1-\alpha}{\alpha} \right)^i \binom{U_2}{i} \times \frac{1}{\binom{d-i}{d-K/m}} \right]} \tag{22}$$

$$= \frac{\sum_{i=0}^{U_2+1} \left[ \left( \frac{1-\alpha}{\alpha} \right)^i \frac{1}{\binom{d-i}{d-K/m}} \times \binom{U_2+1}{i} \right]}{\sum_{i=0}^{U_2} \left[ \left( \frac{1-\alpha}{\alpha} \right)^i \frac{1}{\binom{d-i}{d-K/m}} \times \binom{U_2}{i} \right]} \tag{23}$$

$$= \frac{\frac{1}{\binom{d}{d-K/m}} + \sum_{i=1}^{U_2+1} \left[ \left( \frac{1-\alpha}{\alpha} \right)^i \frac{1}{\binom{d-i}{d-K/m}} \times \binom{U_2+1}{i} \right]}{\frac{1}{\binom{d}{d-K/m}} + \sum_{i=1}^{U_2} \left[ \left( \frac{1-\alpha}{\alpha} \right)^i \frac{1}{\binom{d-i}{d-K/m}} \times \binom{U_2}{i} \right]} \tag{24}$$

$$= \frac{1 + \frac{1-\alpha}{\alpha} \cdot \sum_{i=1}^{U_2+1} \left[ \left( \frac{1-\alpha}{\alpha} \right)^{i-1} \frac{\binom{d}{d-K/m}}{\binom{d-i}{d-K/m}} \times \binom{U_2+1}{i} \right]}{1 + \frac{1-\alpha}{\alpha} \cdot \sum_{i=1}^{U_2} \left[ \left( \frac{1-\alpha}{\alpha} \right)^{i-1} \frac{\binom{d}{d-K/m}}{\binom{d-i}{d-K/m}} \times \binom{U_2}{i} \right]}. \tag{25}$$

Let

$$S_0(\alpha) = \sum_{i=1}^{U_2} \left[ \left( \frac{1-\alpha}{\alpha} \right)^{i-1} \frac{\binom{d}{d-K/m}}{\binom{d-i}{d-K/m}} \times \binom{U_2}{i} \right] \tag{26}$$

and

$$S_1(\alpha) = \sum_{i=1}^{U_2+1} \left[ \left( \frac{1-\alpha}{\alpha} \right)^{i-1} \frac{\binom{d}{d-K/m}}{\binom{d-i}{d-K/m}} \times \binom{U_2+1}{i} \right]. \tag{27}$$

We have

$$\frac{\Pr[\mathcal{M}(\mathcal{T}_1) = \mathcal{I}]}{\Pr[\mathcal{M}(\mathcal{T}_2) = \mathcal{I}]} \leq \frac{1 + \frac{1-\alpha}{\alpha} \cdot S_1(\alpha)}{1 + \frac{1-\alpha}{\alpha} \cdot S_0(\alpha)} \tag{28}$$

$$= 1 + \frac{\frac{1-\alpha}{\alpha} \cdot (S_1(\alpha) - S_0(\alpha))}{1 + \frac{1-\alpha}{\alpha} \cdot S_0(\alpha)} \tag{29}$$

$$\leq 1 + \frac{\frac{1-\alpha}{\alpha} \cdot (S_1(\alpha) - S_0(\alpha))}{\frac{1-\alpha}{\alpha} \cdot S_0(\alpha)} \tag{30}$$

$$= \frac{S_1(\alpha)}{S_0(\alpha)}. \tag{31}$$

Since $U_1 = U_2 + 1 \leq K/m$, we have $U_2 \leq K/m - 1$. Thus,

$$S_1(\alpha) = \sum_{i=1}^{U_2+1} \frac{\left(\frac{1-\alpha}{\alpha}\right)^{i-1} \binom{d}{d-K/m}\binom{U_2+1}{i}}{\binom{d-i}{d-K/m}} \tag{32}$$

$$\leq \sum_{i=1}^{U_2} \frac{\left(\frac{1-\alpha}{\alpha}\right)^{i-1} \binom{d}{d-K/m}\binom{U_2+1}{i}}{\binom{d-i}{d-K/m}} + \sum_{i=2}^{U_2+1} \frac{\left(\frac{1-\alpha}{\alpha}\right)^{i-1} \binom{d}{d-K/m}\binom{U_2+1}{i}}{\binom{d-i}{d-K/m}} \tag{33}$$

$$= \sum_{i=1}^{U_2} \frac{\left(\frac{1-\alpha}{\alpha}\right)^{i-1} \binom{d}{d-K/m}\binom{U_2+1}{i}}{\binom{d-i}{d-K/m}} + \sum_{i=1}^{U_2} \frac{\left(\frac{1-\alpha}{\alpha}\right)^{i} \binom{d}{d-K/m}\binom{U_2+1}{i+1}}{\binom{d-i-1}{d-K/m}} \tag{34}$$

$$= \sum_{i=1}^{U_2} \frac{\left(\frac{1-\alpha}{\alpha}\right)^{i-1} \binom{d}{d-K/m}\binom{U_2}{i}\frac{U_2+1}{U_2+1-i}}{\binom{d-i}{d-K/m}} + \sum_{i=1}^{U_2} \frac{\left(\frac{1-\alpha}{\alpha}\right)\left(\frac{1-\alpha}{\alpha}\right)^{i-1} \binom{d}{d-K/m}\binom{U_2}{i}\frac{U_2+1}{i+1}}{\binom{d-i}{d-K/m}\frac{K/m-i}{d-i}} \tag{35}$$

$$\leq \sum_{i=1}^{U_2} \frac{\left(\frac{1-\alpha}{\alpha}\right)^{i-1} \binom{d}{d-K/m}\binom{U_2}{i}\frac{U_2+1}{U_2+1-U_2}}{\binom{d-i}{d-K/m}} + \sum_{i=1}^{U_2} \frac{\left(\frac{1-\alpha}{\alpha}\right)\left(\frac{1-\alpha}{\alpha}\right)^{i-1} \binom{d}{d-K/m}\binom{U_2}{i}\frac{U_2+1}{1+1}}{\binom{d-i}{d-K/m}\frac{K/m-U_2}{d-U_2}} \tag{36}$$

$$= \left[(U_2 + 1) + \frac{\frac{1-\alpha}{\alpha} \cdot \frac{U_2+1}{2}}{\frac{K/m-U_2}{d-U_2}}\right] \cdot \sum_{i=1}^{U_2} \left[\left(\frac{1-\alpha}{\alpha}\right)^{i-1} \frac{\binom{d}{d-K/m}}{\binom{d-i}{d-K/m}} \times \binom{U_2}{i}\right] \tag{37}$$

$$\leq \left[(K/m - 1 + 1) + \frac{\frac{1-\alpha}{\alpha} \cdot \frac{K/m-1+1}{2}}{\frac{K/m-K/m+1}{d-K/m+1}}\right] \cdot S_0(\alpha) \tag{38}$$

$$= \left(\frac{K}{m} + \frac{\frac{K}{m}(1-\alpha)(d - \frac{K}{m} + 1)}{2\alpha}\right) \cdot S_0(\alpha) \tag{39}$$

$$\leq \left(\frac{K}{m}(d - \frac{K}{m} + 1) + \frac{\frac{K}{m}(1-\alpha)(d - \frac{K}{m} + 1)}{2\alpha}\right) \cdot S_0(\alpha) \tag{40}$$

$$\leq \frac{(1+\alpha) \cdot \frac{K}{m}(d - \frac{K}{m} + 1)}{2\alpha} \cdot S_0(\alpha). \tag{41}$$

Therefore,

$$\frac{\Pr[\mathcal{M}(\mathcal{T}_1) = \mathcal{I}]}{\Pr[\mathcal{M}(\mathcal{T}_2) = \mathcal{I}]} \leq \frac{S_1(\alpha)}{S_0(\alpha)} \leq \frac{(1+\alpha) \cdot \frac{K}{m}(d - \frac{K}{m} + 1)}{2\alpha}. \tag{42}$$

Consequently,

$$\Pr[\mathcal{M}(\mathcal{T}_1) = \mathcal{I}] \leq \exp\left(\ln\left(\frac{(1+\alpha) \cdot \frac{K}{m}(d - \frac{K}{m} + 1)}{2\alpha}\right)\right) \cdot \Pr[\mathcal{M}(\mathcal{T}_2) = \mathcal{I}], \tag{43}$$

which shows that $\mathcal{M}$ provides $\ln\left(\frac{(1+\alpha) \cdot \frac{K}{m}(d - \frac{K}{m} + 1)}{2\alpha}\right)$-DP. $\qquad\square$

Then we provide a stronger result for the case where $\alpha$ is close to 1. Specifically, when $\frac{1}{2} \leq \alpha \leq 1$,

$$\frac{\Pr[\mathcal{M}(\mathcal{T}_1) = \mathcal{I}]}{\Pr[\mathcal{M}(\mathcal{T}_2) = \mathcal{I}]} \leq 1 + \frac{\frac{1-\alpha}{\alpha} \cdot (S_1(\alpha) - S_0(\alpha))}{1 + \frac{1-\alpha}{\alpha} \cdot S_0(\alpha)} \leq 1 + \frac{1-\alpha}{\alpha} \cdot (S_1(\alpha) - S_0(\alpha)). \quad (44)$$

Since $U_1 = U_2 + 1 \leq K/m$, we have $U_2 \leq K/m - 1$. Thus,

$$S_1(\alpha) - S_0(\alpha) = \sum_{i=1}^{U_2+1} \frac{\left(\frac{1-\alpha}{\alpha}\right)^{i-1} \binom{d}{d-K/m} \binom{U_2+1}{i}}{\binom{d-i}{d-K/m}} - \sum_{i=1}^{U_2} \frac{\left(\frac{1-\alpha}{\alpha}\right)^{i-1} \binom{d}{d-K/m} \binom{U_2}{i}}{\binom{d-i}{d-K/m}} \quad (45)$$

$$= \sum_{i=1}^{U_2} \frac{\left(\frac{1-\alpha}{\alpha}\right)^{i-1} \binom{d}{d-K/m} \left[\binom{U_2+1}{i} - \binom{U_2}{i}\right]}{\binom{d-i}{d-K/m}} + \frac{\left(\frac{1-\alpha}{\alpha}\right)^{U_2} \binom{d}{d-K/m}}{\binom{d-U_2-1}{d-K/m}} \quad (46)$$

$$= \sum_{i=1}^{U_2} \frac{\left(\frac{1-\alpha}{\alpha}\right)^{i-1} \binom{d}{d-K/m} \binom{U_2}{i-1}}{\binom{d-i}{d-K/m}} + \frac{\left(\frac{1-\alpha}{\alpha}\right)^{U_2} \binom{d}{d-K/m}}{\binom{d-U_2-1}{d-K/m}} \quad (47)$$

$$\leq \frac{\binom{d}{d-K/m}}{\binom{d-U_2}{d-K/m}} \sum_{i=1}^{U_2} \left(\frac{1-\alpha}{\alpha}\right)^{i-1} \binom{U_2}{i-1} + \frac{\left(\frac{1-\alpha}{\alpha}\right)^{U_2} \binom{d}{d-K/m}}{\binom{d-U_2-1}{d-K/m}} \quad (48)$$

$$\leq \frac{\binom{d}{d-K/m}}{\binom{d-U_2}{d-K/m}} \sum_{i=0}^{U_2} \left(\frac{1-\alpha}{\alpha}\right)^{i} \binom{U_2}{i} + \frac{\left(\frac{1-\alpha}{\alpha}\right)^{U_2} \binom{d}{d-K/m}}{\binom{d-U_2-1}{d-K/m}} \quad (49)$$

$$= \frac{\binom{d}{d-K/m}}{\binom{d-U_2}{d-K/m}} \left(1 + \frac{1-\alpha}{\alpha}\right)^{U_2} + \frac{\left(\frac{1-\alpha}{\alpha}\right)^{U_2} \binom{d}{d-K/m}}{\binom{d-U_2-1}{d-K/m}} \quad (50)$$

$$\leq \frac{\binom{d}{d-K/m} \left(\frac{1}{\alpha}\right)^{U_2}}{\binom{d-U_2-1}{d-K/m}} + \frac{\left(\frac{1-\alpha}{\alpha}\right)^{U_2} \binom{d}{d-K/m}}{\binom{d-U_2-1}{d-K/m}} \quad (51)$$

$$\leq \frac{\binom{d}{d-K/m} 2^{U_2}}{\binom{d-U_2-1}{d-K/m}} + \frac{\left(\frac{1-\frac{1}{2}}{\frac{1}{2}}\right)^{U_2} \binom{d}{d-K/m}}{\binom{d-U_2-1}{d-K/m}} \quad (52)$$

$$= (2^{U_2} + 1) \cdot \frac{\binom{d}{d-K/m}}{\binom{d-U_2-1}{d-K/m}} \quad (53)$$

$$\leq (2^{K/m} + 1) \cdot \binom{d}{K/m} \quad (54)$$

$$\leq (2^{K/m} + 1) \cdot d^{K/m}. \quad (55)$$

Consequently,

$$\Pr[\mathcal{M}(\mathcal{T}_1) = \mathcal{I}] \leq \exp\left(\ln\left(1 + \frac{1-\alpha}{\alpha} \cdot (2^{K/m} + 1) \cdot d^{K/m}\right)\right) \cdot \Pr[\mathcal{M}(\mathcal{T}_2) = \mathcal{I}]. \quad (56)$$

It shows that when $\frac{1}{2} \leq \alpha \leq 1$, $\mathcal{M}$ provides $\epsilon_{\mathcal{M}}$-DP, where $\epsilon_{\mathcal{M}} = \ln\left(1 + \frac{1-\alpha}{\alpha} \cdot (2^{K/m} + 1) \cdot d^{K/m}\right)$. Specially, when $\alpha = 1$, $\epsilon_{\mathcal{M}} = \ln(1 + 0) = 0$. It is consistent to that the coordinate set is totally random when $\alpha = 1$.

### B.4  PROOF OF PROPOSITION 2

*Proof.* $\forall k \in \mathcal{G}, \forall 0 \leq t < T$, we have:

$$\mathcal{I}^t = \bigcup_{k' \in [m]} \mathcal{I}_{k'}^t \supseteq \bigcup_{k' \in \mathcal{G}} \mathcal{I}_{k'}^t = \left[\left(\bigcup_{k' \in \mathcal{G} \setminus \{k\}} \mathcal{I}_{k'}^t\right) \cup \mathcal{I}_k^t\right]. \quad (57)$$

Therefore,

$$\mathbb{E}[\|\tilde{\mathbf{g}}_k^t\|^2|\mathcal{I}_k^t] = \mathbb{E}\left[\sum_{j\in\mathcal{I}^t}(\mathbf{g}_k^t)_j^2\bigg|\mathcal{I}_k^t\right] \tag{58}$$

$$= \mathbb{E}\left[\sum_{j\in\mathcal{I}_k^t}(\mathbf{g}_k^t)_j^2\bigg|\mathcal{I}_k^t\right] + \mathbb{E}\left[\sum_{j\in(\mathcal{I}^t\setminus\mathcal{I}_k^t)}(\mathbf{g}_k^t)_j^2\bigg|\mathcal{I}_k^t\right] \tag{59}$$

$$= \sum_{j\in\mathcal{I}_k^t}(\mathbf{g}_k^t)_j^2 + \mathbb{E}\left[\sum_{j\in(\mathcal{I}^t\setminus\mathcal{I}_k^t)}(\mathbf{g}_k^t)_j^2\bigg|\mathcal{I}_k^t\right] \tag{60}$$

$$= \sum_{j\in\mathcal{I}_k^t}(\mathbf{g}_k^t)_j^2 + \sum_{j\notin\mathcal{I}_k^t}(\mathbf{g}_k^t)_j^2 \cdot \Pr\left[j\in\mathcal{I}^t|\mathcal{I}_k^t\right]. \tag{61}$$

For any $j\notin\mathcal{I}_k^t$,

$$\Pr\left[j\in\mathcal{I}^t|\mathcal{I}_k^t\right]$$

$$\geq \Pr\left[j\in\left(\bigcup_{k'\in\mathcal{G}\setminus\{k\}}\mathcal{I}_{k'}^t\right)\bigg|\mathcal{I}_k^t\right] \tag{62}$$

$$= \Pr\left[j\in\left(\bigcup_{k'\in\mathcal{G}\setminus\{k\}}\left(\tilde{\mathcal{T}}_{k'}^t\cup\mathcal{R}_{k'}^t\right)\right)\bigg|\mathcal{I}_k^t\right] \tag{63}$$

$$= \Pr\left[j\in\left(\bigcup_{k'\in\mathcal{G}\setminus\{k\}}\tilde{\mathcal{T}}_{k'}^t\right)\cup\left(\bigcup_{k'\in\mathcal{G}\setminus\{k\}}\mathcal{R}_{k'}^t\right)\bigg|\mathcal{I}_k^t\right] \tag{64}$$

$$= \Pr\left[j\in\left(\bigcup_{k'\in\mathcal{G}\setminus\{k\}}\tilde{\mathcal{T}}_{k'}^t\right)\bigg|\mathcal{I}_k^t\right] + \Pr\left[j\in\left(\bigcup_{k'\in\mathcal{G}\setminus\{k\}}\mathcal{R}_{k'}^t\right)\setminus\left(\bigcup_{k'\in\mathcal{G}\setminus\{k\}}\tilde{\mathcal{T}}_{k'}^t\right)\bigg|\mathcal{I}_k^t\right]. \tag{65}$$

For simplicity, let

$$\nu = \Pr\left[j\in\left(\bigcup_{k'\in\mathcal{G}\setminus\{k\}}\tilde{\mathcal{T}}_{k'}^t\right)\bigg|\mathcal{I}_k^t\right]\in[0,1], \tag{66}$$

and we have:

$$\Pr\left[j\in\mathcal{I}^t|\mathcal{I}_k^t\right]$$

$$= \nu + (1-\nu)\cdot\Pr\left[j\in\left(\bigcup_{k'\in\mathcal{G}\setminus\{k\}}\mathcal{R}_{k'}^t\right)\bigg|\mathcal{I}_k^t, j\notin\left(\bigcup_{k'\in\mathcal{G}\setminus\{k\}}\tilde{\mathcal{T}}_{k'}^t\right)\right] \tag{67}$$

$$= \nu + (1-\nu)\cdot\left\{1-\Pr\left[j\notin\left(\bigcup_{k'\in\mathcal{G}\setminus\{k\}}\mathcal{R}_{k'}^t\right)\bigg|\mathcal{I}_k^t, j\notin\left(\bigcup_{k'\in\mathcal{G}\setminus\{k\}}\tilde{\mathcal{T}}_{k'}^t\right)\right]\right\} \tag{68}$$

$$= \nu + (1-\nu)\cdot\left\{1-\prod_{k'\in\mathcal{G}\setminus\{k\}}\Pr\left[j\notin\mathcal{R}_{k'}^t\bigg|\mathcal{I}_k^t, j\notin\left(\bigcup_{k'\in\mathcal{G}\setminus\{k\}}\tilde{\mathcal{T}}_{k'}^t\right)\right]\right\} \tag{69}$$

$$\overset{(i)}{=} \nu + (1-\nu)\cdot\left\{1-\prod_{k'\in\mathcal{G}\setminus\{k\}}\left[\sum_{i=0}^{K/m}\Pr[r_{k'}^t=i]\cdot\left(1-\frac{i}{d-K/m+i}\right)\right]\right\} \tag{70}$$

$$\overset{(ii)}{\geq} \nu + (1-\nu)\cdot\left\{1-\prod_{k'\in\mathcal{G}\setminus\{k\}}\left[\sum_{i=0}^{K/m}\Pr[r_{k'}^t=i]\cdot\left(1-\frac{i}{d}\right)\right]\right\} \tag{71}$$

$$= \nu + (1-\nu) \cdot \left\{ 1 - \prod_{k' \in \mathcal{G} \setminus \{k\}} \left[ \sum_{i=0}^{K/m} \Pr[r_{k'}^t = i] - \sum_{i=0}^{K/m} \frac{i \cdot \Pr[r_{k'}^t = i]}{d} \right] \right\} \tag{72}$$

$$= \nu + (1-\nu) \cdot \left\{ 1 - \prod_{k' \in \mathcal{G} \setminus \{k\}} \left( 1 - \frac{\mathbb{E}[r_{k'}^t]}{d} \right) \right\} \tag{73}$$

$$\overset{\text{(iii)}}{=} \nu + (1-\nu) \cdot \left\{ 1 - \prod_{k' \in \mathcal{G} \setminus \{k\}} \left( 1 - \frac{\alpha K}{md} \right) \right\} \tag{74}$$

$$= \nu + (1-\nu) \cdot \left[ 1 - \left( 1 - \frac{\alpha K}{md} \right)^{|\mathcal{G}|-1} \right] \tag{75}$$

$$\geq \nu + (1-\nu) \cdot \left[ 1 - \left( 1 - \frac{\alpha K}{md} \right)^{(1-\delta)m-1} \right] \tag{76}$$

$$\overset{\text{(iv)}}{\geq} \nu \cdot \left[ 1 - \left( 1 - \frac{\alpha K}{md} \right)^{(1-\delta)m-1} \right] + (1-\nu) \cdot \left[ 1 - \left( 1 - \frac{\alpha K}{md} \right)^{(1-\delta)m-1} \right] \tag{77}$$

$$= 1 - \left( 1 - \frac{\alpha K}{md} \right)^{(1-\delta)m-1}, \tag{78}$$

where (i) holds because when $r_k^t = i$, the probability that element $j$ is among the $i$ randomly selected elements from $[d] \setminus \tilde{\mathcal{T}}_k^t$ is $\frac{i}{d - K/m + i}$ since $|[d] \setminus \tilde{\mathcal{T}}_k^t| = d - K/m + i$. Inequality (ii) holds because $i \leq K/m$. Equation (iii) holds because $r_{k'}^t$ follows the binomial distribution $\mathrm{B}(\frac{K}{m}, \alpha)$. Inequality (iv) holds because $1 - \left( 1 - \frac{\mathbb{E}[r_k^t]}{d} \right)^{(1-\delta)m-1} \leq 1$.

Since $0 \leq \alpha \leq 1$ and $0 < \frac{K}{m} < d$, we have $0 \leq \frac{\alpha K}{md} < 1$. Thus,

$$\left( 1 - \frac{\alpha K}{md} \right)^{(1-\delta)m-1} = \left[ \left( 1 - \frac{\alpha K}{md} \right)^{-\frac{md}{\alpha K}} \right]^{-\frac{\alpha K[(1-\delta)m-1]}{md}} \leq e^{-\frac{\alpha K[(1-\delta)m-1]}{md}}. \tag{79}$$

Therefore,

$$\Pr\left[ j \in \mathcal{I}^t | \mathcal{I}_k^t \right] \geq 1 - e^{-\frac{\alpha K[(1-\delta)m-1]}{md}}. \tag{80}$$

Substituting it into (61), it is obtained that

$$\mathbb{E}[\|\tilde{\mathbf{g}}_k^t\|^2 | \mathcal{I}_k^t] \geq \sum_{j \in \mathcal{I}_k^t} (\mathbf{g}_k^t)_j^2 + \left( 1 - e^{-\frac{\alpha K[(1-\delta)m-1]}{md}} \right) \cdot \sum_{j \notin \mathcal{I}_k^t} (\mathbf{g}_k^t)_j^2 \tag{81}$$

$$= \sum_{j \in \mathcal{I}_k^t} (\mathbf{g}_k^t)_j^2 + \left( 1 - e^{-\frac{\alpha K[(1-\delta)m-1]}{md}} \right) \cdot \left( \|\mathbf{g}_k^t\|^2 - \sum_{j \in \mathcal{I}_k^t} (\mathbf{g}_k^t)_j^2 \right) \tag{82}$$

$$= \left( 1 - e^{-\frac{\alpha K[(1-\delta)m-1]}{md}} \right) \cdot \|\mathbf{g}_k^t\|^2 + e^{-\frac{\alpha K[(1-\delta)m-1]}{md}} \sum_{j \in \mathcal{I}_k^t} (\mathbf{g}_k^t)_j^2. \tag{83}$$

Take total expectation and we have:

$$\mathbb{E}\|\tilde{\mathbf{g}}_k^t\|^2 = \mathbb{E}[\mathbb{E}[\|\tilde{\mathbf{g}}_k^t\|^2 | \mathcal{I}_k^t]] = \left( 1 - e^{-\frac{\alpha K[(1-\delta)m-1]}{md}} \right) \cdot \|\mathbf{g}_k^t\|^2 + e^{-\frac{\alpha K[(1-\delta)m-1]}{md}} \cdot \mathbb{E}\left[ \sum_{j \in \mathcal{I}_k^t} (\mathbf{g}_k^t)_j^2 \right]. \tag{84}$$

Also,

$$\mathbb{E}\left[ \sum_{j \in \mathcal{I}_k^t} (\mathbf{g}_k^t)_j^2 \Big| r_k^t \right]$$

$$= \mathbb{E}\left[\sum_{j \in \tilde{\mathcal{T}}_k^t} (\mathbf{g}_k^t)_j^2 \Big| r_k^t\right] + \mathbb{E}\left[\sum_{j \in \mathcal{R}_k^t} (\mathbf{g}_k^t)_j^2 \Big| r_k^t\right] \tag{85}$$

$$= \mathbb{E}\left[\sum_{j \in \tilde{\mathcal{T}}_k^t} (\mathbf{g}_k^t)_j^2 \Big| r_k^t\right] + \frac{r_k^t}{d - K/m + r_k^t} \cdot \mathbb{E}\left[\sum_{j \notin \tilde{\mathcal{T}}_k^t} (\mathbf{g}_k^t)_j^2 \Big| r_k^t\right] \tag{86}$$

$$= \mathbb{E}\left[\sum_{j \in \tilde{\mathcal{T}}_k^t} (\mathbf{g}_k^t)_j^2 \Big| r_k^t\right] + \frac{r_k^t}{d - K/m + r_k^t} \cdot \left(\|\mathbf{g}_k^t\|^2 - \mathbb{E}\left[\sum_{j \in \tilde{\mathcal{T}}_k^t} (\mathbf{g}_k^t)_j^2 \Big| r_k^t\right]\right) \tag{87}$$

$$= \frac{r_k^t}{d - K/m + r_k^t} \cdot \|\mathbf{g}_k^t\|^2 + \frac{d - K/m}{d - K/m + r_k^t} \cdot \mathbb{E}\left[\sum_{j \in \tilde{\mathcal{T}}_k^t} (\mathbf{g}_k^t)_j^2 \Big| r_k^t\right] \tag{88}$$

$$= \frac{r_k^t}{d - K/m + r_k^t} \cdot \|\mathbf{g}_k^t\|^2 + \frac{d - K/m}{d - K/m + r_k^t} \cdot \frac{K/m - r_k^t}{K/m} \cdot \mathbb{E}\left[\sum_{j \in \mathcal{T}_k^t} (\mathbf{g}_k^t)_j^2 \Big| r_k^t\right] \tag{89}$$

$$\geq \frac{r_k^t}{d - K/m + r_k^t} \cdot \|\mathbf{g}_k^t\|^2 + \frac{d - K/m}{d - K/m + r_k^t} \cdot \frac{K/m - r_k^t}{K/m} \cdot \frac{K/m}{d} \cdot \|\mathbf{g}_k^t\|^2 \tag{90}$$

$$= \frac{r_k^t}{d - K/m + r_k^t} \cdot \|\mathbf{g}_k^t\|^2 + \frac{d - K/m}{d - K/m + r_k^t} \cdot \frac{K/m - r_k^t}{d} \cdot \|\mathbf{g}_k^t\|^2 \tag{91}$$

$$= \frac{dr_k^t + (d - K/m)(K/m - r_k^t)}{d(d - K/m + r_k^t)} \cdot \|\mathbf{g}_k^t\|^2 \tag{92}$$

$$= \frac{(K/m) \cdot (d - K/m + r_k^t)}{d(d - K/m + r_k^t)} \cdot \|\mathbf{g}_k^t\|^2 \tag{93}$$

$$= \frac{K}{md}\|\mathbf{g}_k^t\|^2. \tag{94}$$

Thus,

$$\mathbb{E}\left[\sum_{j \in \mathcal{I}_k^t} (\mathbf{g}_k^t)_j^2\right] = \mathbb{E}\left[\mathbb{E}\left[\sum_{j \in \mathcal{I}_k^t} (\mathbf{g}_k^t)_j^2 \Big| r_k^t\right]\right] \geq \mathbb{E}\left[\frac{K}{md}\|\mathbf{g}_k^t\|^2\right] = \frac{K}{md}\|\mathbf{g}_k^t\|^2. \tag{95}$$

Substituting (95) into (84), we have:

$$\mathbb{E}\|\tilde{\mathbf{g}}_k^t\|^2 \geq \left(1 - e^{-\frac{\alpha K[(1-\delta)m-1]}{md}} + \frac{K}{md} e^{-\frac{\alpha K[(1-\delta)m-1]}{md}}\right) \cdot \|\mathbf{g}_k^t\|^2 \tag{96}$$

$$= \left(1 - \frac{(d - \frac{K}{m}) e^{-\frac{\alpha K[(1-\delta)m-1]}{md}}}{d}\right) \cdot \|\mathbf{g}_k^t\|^2. \tag{97}$$

Since $\tilde{\mathbf{g}}_k^t$ is the consensus sparsification result of $\mathbf{g}_k^t$, we have:

$$\mathbb{E}\|\tilde{\mathbf{g}}_k^t - \mathbf{g}_k^t\|^2 = \mathbb{E}\left[\sum_{j \in \mathcal{G} \setminus \mathcal{I}^t} (\mathbf{g}_k^t)_j^2\right] \tag{98}$$

$$= \mathbb{E}\left[\sum_{j \in \mathcal{G}} (\mathbf{g}_k^t)_j^2 - \sum_{j \in \mathcal{I}^t} (\mathbf{g}_k^t)_j^2\right] \tag{99}$$

$$= \|\mathbf{g}_k^t\|^2 - \mathbb{E}\|\tilde{\mathbf{g}}_k^t\|^2 \tag{100}$$

$$\leq \frac{(d - \frac{K}{m}) e^{-\frac{\alpha K[(1-\delta)m-1]}{md}}}{d} \cdot \|\mathbf{g}_k^t\|^2 \tag{101}$$

$$= \left(1 - \frac{d(1 - e^{-\frac{\alpha K[(1-\delta)m-1]}{md}}) + \frac{K}{m} e^{-\frac{\alpha K[(1-\delta)m-1]}{md}}}{d}\right) \cdot \|\mathbf{g}_k^t\|^2. \tag{102}$$

By definition, consensus sparsification is a $d'_{cons}$-contraction operator, where

$$d'_{cons} = d\left(1 - e^{-\frac{\alpha K[(1-\delta)m-1]}{md}}\right) + \frac{K}{m}e^{-\frac{\alpha K[(1-\delta)m-1]}{md}}.$$

$\square$

### B.5   PROOF OF LEMMA 1

*Proof.* When training algorithm $\mathcal{A}$ is $I$-iteration local SGD with learning rate $\eta$, we have $\mathbf{w}_k^{t+1,0} = \mathbf{w}^t$, $\mathbf{w}_k^{t+1,j+1} = \mathbf{w}_k^{t+1,j} - \eta_t \cdot \nabla f_{i_k^{t,j}}(\mathbf{w}_k^{t+1,j})$ ($j = 0, 1, \ldots, I-1$) and $\mathbf{w}_k^{t+1} = \mathbf{w}_k^{t+1,I}$, where $i_k^{t,j}$ is uniformly sampled from $\mathcal{D}_k$. Therefore, we have the following inequality for all $k \in \mathcal{G}$:

$$\mathbb{E}\|\mathbf{u}_k^{t+1}\|^2 = \mathbb{E}\|\mathbf{g}_k^t - \tilde{\mathbf{g}}_k^t\|^2 \tag{103}$$

$$\overset{(i)}{\leq} \left(1 - \frac{d'_{cons}}{d}\right)\mathbb{E}\|\mathbf{g}_k^t\|^2 \tag{104}$$

$$= \left(1 - \frac{d'_{cons}}{d}\right)\mathbb{E}\|\mathbf{u}_k^t + (\mathbf{w}^t - \mathbf{w}_k^{t+1})\|^2 \tag{105}$$

$$\overset{(ii)}{\leq} \left(1 - \frac{d'_{cons}}{d}\right)\left[(1 + \frac{d'_{cons}}{2d})\mathbb{E}\|\mathbf{u}_k^t\|^2 + (1 + \frac{2d}{d'_{cons}})\mathbb{E}\|\mathbf{w}_k^{t+1,0} - \mathbf{w}_k^{t+1,I}\|^2\right] \tag{106}$$

$$\overset{(iii)}{\leq} \left(1 - \frac{d'_{cons}}{2d}\right)\mathbb{E}\|\mathbf{u}_k^t\|^2 + \frac{2d}{d'_{cons}}\mathbb{E}\|\mathbf{w}_k^{t+1,0} - \mathbf{w}_k^{t+1,I}\|^2 \tag{107}$$

$$\leq \left(1 - \frac{d'_{cons}}{2d}\right)\mathbb{E}\|\mathbf{u}_k^t\|^2 + \frac{2Id}{d'_{cons}}\sum_{j=0}^{I-1}\mathbb{E}\|\mathbf{w}_k^{t+1,i} - \mathbf{w}_k^{t+1,i+1}\|^2 \tag{108}$$

$$= \left(1 - \frac{d'_{cons}}{2d}\right)\mathbb{E}\|\mathbf{u}_k^t\|^2 + \frac{2Id}{d'_{cons}}\sum_{j=0}^{I-1}\mathbb{E}\|\eta_t \cdot \nabla f_{i_k^{t,j}}(\mathbf{w}_k^{t+1,j})\|^2 \tag{109}$$

$$\overset{(iv)}{\leq} \left(1 - \frac{d'_{cons}}{2d}\right)\mathbb{E}\|\mathbf{u}_k^t\|^2 + \frac{2I^2 d}{d'_{cons}}(\eta_t)^2(D^2 + \sigma^2), \tag{110}$$

where (i) is derived based on Proposition 2. (ii) is derived based on that $\|\mathbf{x} + \mathbf{y}\|^2 \leq (1+\theta)\|\mathbf{x}\|^2 + (1+\theta^{-1})\|\mathbf{y}\|^2$ for any constant $\theta > 0$. (iii) is derived based on that $(1 - \frac{d'_{cons}}{d})(1 + \frac{d'_{cons}}{2d}) < 1 - \frac{d'_{cons}}{2d}$ and $(1 - \frac{d'_{cons}}{d})(1 + \frac{2d}{d'_{cons}}) < \frac{2d}{d'_{cons}}$. (iv) is derived based on Assumption 5 and Assumption 6.

When $\eta_t = \frac{b}{\sqrt{t+\lambda}}$ where constant $b > 0$ and $\lambda = \frac{4d}{d'_{cons}}$, the second term on the RHS

$$\frac{2I^2 d}{d'_{cons}}(\eta_t)^2(D^2 + \sigma^2) = \frac{2I^2 d}{d'_{cons}}(D^2 + \sigma^2) \cdot \frac{b^2}{t+\lambda} \tag{111}$$

$$= \left(\frac{8I^2 d^2 b^2}{(d'_{cons})^2}(D^2 + \sigma^2)\right) \cdot \frac{1}{t+\lambda} \cdot \frac{d'_{cons}}{4d} \tag{112}$$

$$= \left(\frac{8I^2 d^2 b^2}{(d'_{cons})^2}(D^2 + \sigma^2)\right) \cdot \frac{1}{t+\lambda} \cdot (\frac{d'_{cons}}{2d} - \frac{d'_{cons}}{4d}) \tag{113}$$

$$= \left(\frac{8I^2 d^2 b^2}{(d'_{cons})^2}(D^2 + \sigma^2)\right) \cdot \frac{1}{t+\lambda} \cdot (\frac{d'_{cons}}{2d} - \frac{1}{\lambda}) \tag{114}$$

$$\leq \left(\frac{8I^2 d^2 b^2}{(d'_{cons})^2}(D^2 + \sigma^2)\right) \cdot \frac{1}{t+\lambda} \cdot (\frac{d'_{cons}}{2d} - \frac{1}{t+\lambda+1}) \tag{115}$$

$$= \left(\frac{8I^2 d^2 b^2}{(d'_{cons})^2}(D^2 + \sigma^2)\right) \cdot \left(\frac{\frac{d'_{cons}}{2d}(t+\lambda+1) - 1}{(t+\lambda)(t+\lambda+1)}\right) \tag{116}$$

$$= \left(\frac{8I^2 d^2 b^2}{(d'_{cons})^2}(D^2 + \sigma^2)\right) \cdot \left(\frac{1}{t+\lambda+1} - \frac{(1 - \frac{d'_{cons}}{2d})}{t+\lambda}\right). \tag{117}$$

Combining (110) and (117), we have

$$\mathbb{E}\|\mathbf{u}_k^{t+1}\|^2 \leq \left(1 - \frac{d'_{cons}}{2d}\right)\mathbb{E}\|\mathbf{u}_k^t\|^2 + \left(\frac{8I^2d^2b^2}{(d'_{cons})^2}(D^2 + \sigma^2)\right) \cdot \left(\frac{1}{t+\lambda+1} - \frac{(1 - \frac{d'_{cons}}{2d})}{t+\lambda}\right) \cdot \tag{118}$$

Therefore,

$$\left(\mathbb{E}\|\mathbf{u}_k^{t+1}\|^2 - \frac{8I^2d^2b^2(D^2 + \sigma^2)}{(d'_{cons})^2(t+\lambda+1)}\right) \leq \left(1 - \frac{d'_{cons}}{2d}\right)\left(\mathbb{E}\|\mathbf{u}_k^t\|^2 - \frac{8I^2d^2b^2(D^2 + \sigma^2)}{(d'_{cons})^2(t+\lambda)}\right). \tag{119}$$

Recursively using (119), we have

$$\left(\mathbb{E}\|\mathbf{u}_k^t\|^2 - \frac{8I^2d^2b^2(D^2 + \sigma^2)}{(d'_{cons})^2(t+\lambda)}\right) \leq \left(1 - \frac{d'_{cons}}{2d}\right)^t\left(\mathbb{E}\|\mathbf{u}_k^0\|^2 - \frac{8I^2d^2b^2(D^2 + \sigma^2)}{(d'_{cons})^2\lambda}\right) < 0. \tag{120}$$

Thus,

$$\mathbb{E}\|\mathbf{u}_k^t\|^2 \leq \frac{8I^2d^2(D^2 + \sigma^2)}{(d'_{cons})^2} \cdot \frac{b^2}{t+\lambda} = \frac{8I^2d^2(D^2 + \sigma^2)}{(d'_{cons})^2} \cdot (\eta_t)^2. \tag{121}$$

Finally,

$$\mathbb{E}\|\mathbf{u}^t\|^2 = \mathbb{E}\|\frac{1}{|\mathcal{G}|}\sum_{k\in\mathcal{G}}\mathbf{u}_k^t\|^2 \leq \frac{1}{|\mathcal{G}|}\sum_{k\in\mathcal{G}}\mathbb{E}\|\mathbf{u}_k^t\|^2 \leq \frac{8I^2d^2(D^2 + \sigma^2)}{(d'_{cons})^2} \cdot (\eta_t)^2. \tag{122}$$

When $\eta_t = \eta$, by (110), we have

$$\mathbb{E}\|\mathbf{u}_k^{t+1}\|^2 \leq \left(1 - \frac{d'_{cons}}{2d}\right)\mathbb{E}\|\mathbf{u}_k^t\|^2 + \frac{2I^2d}{d'_{cons}}\eta^2(D^2 + \sigma^2) \tag{123}$$

$$\left(\mathbb{E}\|\mathbf{u}_k^{t+1}\|^2 - \frac{4I^2d^2}{(d'_{cons})^2}\eta^2(D^2 + \sigma^2)\right) \leq \left(1 - \frac{d'_{cons}}{2d}\right) \cdot \left(\mathbb{E}\|\mathbf{u}_k^t\|^2 - \frac{4I^2d^2}{(d'_{cons})^2}\eta^2(D^2 + \sigma^2)\right). \tag{124}$$

Recursively using (124), we have

$$\mathbb{E}\|\mathbf{u}_k^t\|^2 - \frac{4I^2d^2}{(d'_{cons})^2}\eta^2(D^2 + \sigma^2) \leq \left(1 - \frac{d'_{cons}}{2d}\right)^t \cdot \left(\mathbb{E}\|\mathbf{u}_k^0\|^2 - \frac{4I^2d^2}{(d'_{cons})^2}\eta^2(D^2 + \sigma^2)\right) < 0. \tag{125}$$

Thus,

$$\mathbb{E}\|\mathbf{u}_k^t\|^2 \leq \frac{4I^2d^2(D^2 + \sigma^2)}{(d'_{cons})^2} \cdot \eta^2. \tag{126}$$

Finally,

$$\mathbb{E}\|\mathbf{u}^t\|^2 = \mathbb{E}\|\frac{1}{|\mathcal{G}|}\sum_{k\in\mathcal{G}}\mathbf{u}_k^t\|^2 \leq \frac{1}{|\mathcal{G}|}\sum_{k\in\mathcal{G}}\mathbb{E}\|\mathbf{u}_k^t\|^2 \leq \frac{4I^2d^2(D^2 + \sigma^2)}{(d'_{cons})^2} \cdot \eta^2. \tag{127}$$

$\square$

## B.6 PROOF OF LEMMA 2

*Proof.* Based on Assumption 6 and Assumption 5, we have that $\forall k \in \mathcal{G}$,

$$\mathbb{E}\|\tilde{\mathbf{g}}_k^t\|^2 \leq \mathbb{E}\|\mathbf{g}_k^t\|^2 = \mathbb{E}\|\mathbf{u}_k^t + (\mathbf{w}^t - \mathbf{w}_k^{t+1})\|^2 \tag{128}$$

$$\leq 2\mathbb{E}\|\mathbf{u}_k^t\|^2 + 2\mathbb{E}\|\mathbf{w}^t - \mathbf{w}_k^{t+1}\|^2 \tag{129}$$

$$\leq 2\mathbb{E}\|\mathbf{u}_k^t\|^2 + 2I^2(\eta_t)^2(D^2 + \sigma^2). \tag{130}$$

By Lemma 1, if $\eta_t = \frac{b}{\sqrt{t+\lambda}}$ where constant $b > 0$ and $\lambda = \frac{4d}{d'_{cons}}$, we have

$$\mathbb{E}\|\tilde{\mathbf{g}}_k^t\|^2 \leq 2I^2(8H^2 + 1)(D^2 + \sigma^2) \cdot (\eta_t)^2, \quad \forall k \in \mathcal{G}. \tag{131}$$

$$\mathbb{E}_{k\neq k'}\left[\|\tilde{\mathbf{g}}_k^t - \tilde{\mathbf{g}}_{k'}^t\|^2\right] \leq 2\mathbb{E}\|\tilde{\mathbf{g}}_k^t\|^2 + 2\mathbb{E}\|\tilde{\mathbf{g}}_{k'}^t\|^2 \leq 8I^2(8H^2+1)(D^2+\sigma^2)\cdot(\eta_t)^2. \tag{132}$$

Therefore, by Definition 1 and (132),

$$\mathbb{E}\|\mathbf{e}^t\|^2 = \mathbb{E}\left\|\mathbf{SRAgg}(\{\tilde{\mathbf{g}}_k^t\}_{k=1}^m) - \frac{1}{|\mathcal{G}|}\sum_{k\in\mathcal{G}}\tilde{\mathbf{g}}_k^t\right\|^2 \tag{133}$$

$$\leq c\delta\cdot\mathbb{E}_{k\neq k'}\left[\|\tilde{\mathbf{g}}_k^t - \tilde{\mathbf{g}}_{k'}^t\|^2\right] \tag{134}$$

$$\leq 8c\delta I^2(8H^2+1)(D^2+\sigma^2)\cdot(\eta_t)^2. \tag{135}$$

Similarly, if $\eta_t = \eta > 0$, we have

$$\mathbb{E}\|\tilde{\mathbf{g}}_k^t\|^2 \leq 2I^2(4H^2+1)(D^2+\sigma^2)\cdot\eta^2, \quad \forall k\in\mathcal{G}. \tag{136}$$

and

$$\mathbb{E}\|\mathbf{e}^t\|^2 \leq 8c\delta I^2(4H^2+1)(D^2+\sigma^2)\cdot\eta^2. \tag{137}$$

$\square$

## B.7 PROOF OF THEOREM 3

*Proof.* Let $\mathbf{u}^t = \frac{1}{|\mathcal{G}|}\sum_{k\in\mathcal{G}}\mathbf{u}_k^t$ be the averaging memory of non-Byzantine clients. $\hat{\mathbf{w}}$ is defined as $\hat{\mathbf{w}}^t = \mathbf{w}^t - \mathbf{u}^t$. The iteration rule for $\hat{\mathbf{w}}$ is derived as follows:

$$\hat{\mathbf{w}}^{t+1} = \mathbf{w}^{t+1} - \mathbf{u}^{t+1} \tag{138}$$

$$= \left(\mathbf{w}^t - \mathbf{SRAgg}(\{\tilde{\mathbf{g}}_k^t\}_{k=1}^m)\right) - \frac{1}{|\mathcal{G}|}\sum_{k\in\mathcal{G}}\left(\mathbf{u}_k^t + (\mathbf{w}^t - \mathbf{w}_k^{t+1}) - \tilde{\mathbf{g}}_k^t\right) \tag{139}$$

$$= \mathbf{w}^t - \mathbf{SRAgg}(\{\tilde{\mathbf{g}}_k^t\}_{k=1}^m) - \left(\mathbf{u}^t + \frac{1}{|\mathcal{G}|}\sum_{k\in\mathcal{G}}(\mathbf{w}^t - \mathbf{w}_k^{t+1}) - \frac{1}{|\mathcal{G}|}\sum_{k\in\mathcal{G}}\tilde{\mathbf{g}}_k^t\right) \tag{140}$$

$$= (\mathbf{w}^t - \mathbf{u}^t) - \frac{1}{|\mathcal{G}|}\sum_{k\in\mathcal{G}}(\mathbf{w}^t - \mathbf{w}_k^{t+1}) - \left(\mathbf{SRAgg}(\{\tilde{\mathbf{g}}_k^t\}_{k=1}^m) - \frac{1}{|\mathcal{G}|}\sum_{k\in\mathcal{G}}\tilde{\mathbf{g}}_k^t\right) \tag{141}$$

$$= \hat{\mathbf{w}}^t - \left(\mathbf{w}^t - \frac{1}{|\mathcal{G}|}\sum_{k\in\mathcal{G}}\mathbf{w}_k^{t+1}\right) - \mathbf{e}^t, \tag{142}$$

where $\mathbf{e}^t = \mathbf{SRAgg}(\{\tilde{\mathbf{g}}_k^t\}_{k=1}^m) - \frac{1}{|\mathcal{G}|}\sum_{k\in\mathcal{G}}\tilde{\mathbf{g}}_k^t$ is the estimation error of $\frac{1}{|\mathcal{G}|}\sum_{k\in\mathcal{G}}\tilde{\mathbf{g}}_k^t$.

Let $\bar{\mathbf{G}}^t = (\mathbf{w}^t - \frac{1}{|\mathcal{G}|}\sum_{k\in\mathcal{G}}\mathbf{w}_k^{t+1})/(\eta I)$. Then we have

$$\bar{\mathbf{G}}^t = \frac{1}{I|\mathcal{G}|}\sum_{k\in\mathcal{G}}\sum_{j=0}^{I-1}\nabla f_{i_k^{t,j}}(\mathbf{w}_k^{t+1,j}), \quad t = 0, 1, \ldots, T-1, \tag{143}$$

and

$$\hat{\mathbf{w}}^{t+1} = \hat{\mathbf{w}}^t - \eta I\cdot\bar{\mathbf{G}}^t - \mathbf{e}^t, \quad t = 0, 1, \ldots, T-1. \tag{144}$$

The equation can be interpreted as that $\hat{\mathbf{w}}^{t+1}$ is obtained by performing an SGD step on $\hat{\mathbf{w}}^t$ with learning rate $\eta I$, gradient approximation $\bar{\mathbf{G}}^t$ and error $\mathbf{e}^t$.
Based on Assumption 3 and the inequality that $\|\mathbf{x}+\mathbf{y}\|^2 \leq 2\|\mathbf{x}\|^2 + 2\|\mathbf{y}\|^2$,

$$F(\hat{\mathbf{w}}^{t+1}) = F(\hat{\mathbf{w}}^t - \eta I\cdot\bar{\mathbf{G}}^t - \mathbf{e}^t) \tag{145}$$

$$\leq F(\hat{\mathbf{w}}^t) - \nabla F(\hat{\mathbf{w}}^t)^T(\eta I\cdot\bar{\mathbf{G}}^t + \mathbf{e}^t) + \frac{L}{2}\|\eta I\cdot\bar{\mathbf{G}}^t + \mathbf{e}^t\|^2 \tag{146}$$

$$\leq F(\hat{\mathbf{w}}^t) - \eta I\cdot\nabla F(\hat{\mathbf{w}}^t)^T\bar{\mathbf{G}}^t - \nabla F(\hat{\mathbf{w}}^t)^T\mathbf{e}^t + \eta^2 I^2 L\|\bar{\mathbf{G}}^t\|^2 + L\|\mathbf{e}^t\|^2 \tag{147}$$

$$= F(\hat{\mathbf{w}}^t) - \eta I\cdot\left(\|\nabla F(\hat{\mathbf{w}}^t)\|^2 + \nabla F(\hat{\mathbf{w}}^t)^T[\bar{\mathbf{G}}^t - \nabla F(\hat{\mathbf{w}}^t)]\right)$$

$$- \nabla F(\hat{\mathbf{w}}^t)^T\mathbf{e}^t + \eta^2 I^2 L\|\bar{\mathbf{G}}^t\|^2 + L\|\mathbf{e}^t\|^2 \tag{148}$$

$$=F(\hat{\mathbf{w}}^t) - \eta I \cdot \|\nabla F(\hat{\mathbf{w}}^t)\|^2 - \eta I \cdot \nabla F(\hat{\mathbf{w}}^t)^T[\bar{\mathbf{G}}^t - \nabla F(\hat{\mathbf{w}}^t)]$$

$$- \nabla F(\hat{\mathbf{w}}^t)^T\mathbf{e}^t + \eta^2 I^2 L \|\bar{\mathbf{G}}^t\|^2 + L\|\mathbf{e}^t\|^2. \tag{149}$$

Taking expectation on both sides, we have

$$\mathbb{E}[F(\hat{\mathbf{w}}^{t+1})|\mathbf{w}^t, \mathbf{u}^t] \leq F(\hat{\mathbf{w}}^t) - \eta I \cdot \|\nabla F(\hat{\mathbf{w}}^t)\|^2 - \eta I \cdot \mathbb{E}\Big[\nabla F(\hat{\mathbf{w}}^t)^T[\bar{\mathbf{G}}^t - \nabla F(\hat{\mathbf{w}}^t)]\Big|\mathbf{w}^t, \mathbf{u}^t\Big]$$

$$- \mathbb{E}[\nabla F(\hat{\mathbf{w}}^t)^T\mathbf{e}^t|\mathbf{w}^t, \mathbf{u}^t] + \eta^2 I^2 L \cdot \mathbb{E}[\|\bar{\mathbf{G}}^t\|^2|\mathbf{w}^t, \mathbf{u}^t] + L \cdot \mathbb{E}[\|\mathbf{e}^t\|^2|\mathbf{w}^t, \mathbf{u}^t]. \tag{150}$$

Based on Assumption 3 and that $-\|\mathbf{x}\|^2 \leq -\frac{1}{2}\|\mathbf{y}\|^2 + \|\mathbf{x} - \mathbf{y}\|^2$, we have:

$$-\|\nabla F(\hat{\mathbf{w}}^t)\|^2 = -\|\nabla F(\mathbf{w}^t) + [\nabla F(\hat{\mathbf{w}}^t) - \nabla F(\mathbf{w}^t)]\|^2 \tag{151}$$

$$\leq -\frac{1}{2}\|\nabla F(\mathbf{w}^t)\|^2 + \|\nabla F(\hat{\mathbf{w}}^t) - \nabla F(\mathbf{w}^t)\|^2 \tag{152}$$

$$= -\frac{1}{2}\|\nabla F(\mathbf{w}^t)\|^2 + \|\nabla F(\mathbf{w}^t - \mathbf{u}^t) - \nabla F(\mathbf{w}^t)\|^2 \tag{153}$$

$$\leq -\frac{1}{2}\|\nabla F(\mathbf{w}^t)\|^2 + L^2\|\mathbf{u}^t\|^2. \tag{154}$$

In addition, using Assumption 3, Assumption 4, Assumption 5 and Equation (143), we have:

$$- \mathbb{E}\Big[\nabla F(\hat{\mathbf{w}}^t)^T[\bar{\mathbf{G}}^t - \nabla F(\hat{\mathbf{w}}^t)]\Big|\mathbf{w}^t, \mathbf{u}^t\Big]$$

$$= -\nabla F(\hat{\mathbf{w}}^t)^T \cdot \mathbb{E}[\bar{\mathbf{G}}^t - \nabla F(\hat{\mathbf{w}}^t)|\mathbf{w}^t, \mathbf{u}^t]$$

$$\leq \|\nabla F(\hat{\mathbf{w}}^t)\| \cdot \Big\|\mathbb{E}[\bar{\mathbf{G}}^t - \nabla F(\hat{\mathbf{w}}^t)|\mathbf{w}^t, \mathbf{u}^t]\Big\| \tag{155}$$

$$\leq D \cdot \left\|\mathbb{E}\left[\nabla F(\hat{\mathbf{w}}^t) - \frac{1}{I|\mathcal{G}|}\sum_{k\in\mathcal{G}}\sum_{j=0}^{I-1}\nabla f_{i_k^{t,j}}(\mathbf{w}_k^{t+1,j})\Big|\mathbf{w}^t, \mathbf{u}^t\right]\right\| \tag{156}$$

$$\leq D \cdot \left\|\frac{1}{I|\mathcal{G}|}\sum_{k\in\mathcal{G}}\sum_{j=0}^{I-1}\mathbb{E}\left[\nabla F(\mathbf{w}^t - \mathbf{u}^t) - \nabla F_k(\mathbf{w}_k^{t+1,j})\Big|\mathbf{w}^t, \mathbf{u}^t\right]\right\| \tag{157}$$

$$\leq \frac{D}{I|\mathcal{G}|}\sum_{k\in\mathcal{G}}\sum_{j=0}^{I-1}\mathbb{E}\left[\Big\|\nabla F(\mathbf{w}_k^{t+1,0} - \mathbf{u}^t) - \nabla F(\mathbf{w}_k^{t+1,j})\Big\|\Big|\mathbf{w}^t, \mathbf{u}^t\right]$$

$$+ \frac{D}{I|\mathcal{G}|}\sum_{k\in\mathcal{G}}\sum_{j=0}^{I-1}\mathbb{E}\left[\Big\|\nabla F(\mathbf{w}_k^{t+1,j}) - \nabla F_k(\mathbf{w}_k^{t+1,j})\Big\|\Big|\mathbf{w}^t, \mathbf{u}^t\right] \tag{158}$$

$$\leq \frac{D}{I|\mathcal{G}|}\sum_{k\in\mathcal{G}}\sum_{j=0}^{I-1}\left(L \cdot \mathbb{E}\left[\Big\|\mathbf{w}_k^{t+1,0} - \mathbf{u}^t - \mathbf{w}_k^{t+1,j}\Big\|\Big|\mathbf{w}^t, \mathbf{u}^t\right] + B\right) \tag{159}$$

$$\leq \frac{DL}{I|\mathcal{G}|}\sum_{k\in\mathcal{G}}\sum_{j=0}^{I-1}\left(\|\mathbf{u}^t\| + \mathbb{E}\left[\Big\|\mathbf{w}_k^{t+1,0} - \mathbf{w}_k^{t+1,j}\Big\|\Big|\mathbf{w}^t, \mathbf{u}^t\right]\right) + BD \tag{160}$$

$$\leq \frac{DL}{I|\mathcal{G}|}\sum_{k\in\mathcal{G}}\sum_{j=0}^{I-1}\left(\sum_{j'=0}^{j-1}\mathbb{E}\left[\Big\|\mathbf{w}_k^{t+1,j'} - \mathbf{w}_k^{t+1,j'+1}\Big\|\Big|\mathbf{w}^t, \mathbf{u}^t\right]\right) + DL \cdot \|\mathbf{u}^t\| + BD. \tag{161}$$

With Assumption 5, we have:

$$\mathbb{E}\left[\Big\|\mathbf{w}_k^{t+1,j'} - \mathbf{w}_k^{t+1,j'+1}\Big\|\Big|\mathbf{w}^t, \mathbf{u}^t\right] = \mathbb{E}\left[\Big\|\eta \cdot \nabla f_{i_k^{t,j'}}(\mathbf{w}_k^{t+1,j'})\Big\|\Big|\mathbf{w}^t, \mathbf{u}^t\right] \leq \eta D. \tag{162}$$

Therefore,

$$- \mathbb{E}\Big[\nabla F(\hat{\mathbf{w}}^t)^T[\bar{\mathbf{G}}^t - \nabla F(\hat{\mathbf{w}}^t)]\Big|\mathbf{w}^t, \mathbf{u}^t\Big]$$

$$\leq \frac{DL}{I|\mathcal{G}|} \sum_{k\in\mathcal{G}} \sum_{j=0}^{I-1} \Big( \sum_{j'=0}^{j-1} \eta D \Big) + DL \cdot \|\mathbf{u}^t\| + BD \tag{163}$$

$$= \frac{4L^2}{I|\mathcal{G}|} \sum_{k\in\mathcal{G}} \sum_{j=0}^{I-1} \Big( j\eta D \Big) + DL \cdot \|\mathbf{u}^t\| + BD \tag{164}$$

$$= \frac{4L^2}{I|\mathcal{G}|} \cdot |\mathcal{G}| \frac{I(I-1)}{2} \eta D + DL \cdot \|\mathbf{u}^t\| + BD \tag{165}$$

$$= 2(I-1)\eta DL^2 + DL \cdot \|\mathbf{u}^t\| + BD. \tag{166}$$

Note that $\mathbb{E}[XY] \leq \sqrt{\mathbb{E}[X^2] \cdot \mathbb{E}[Y^2]}$. Using Assumption 5 and Lemma 2, we have:

$$-\mathbb{E}[\nabla F(\hat{\mathbf{w}}^t)^T \mathbf{e}^t | \mathbf{w}^t, \mathbf{u}^t] \leq \mathbb{E}[\|\nabla F(\hat{\mathbf{w}}^t)\| \cdot \|\mathbf{e}^t\| | \mathbf{w}^t, \mathbf{u}^t] \tag{167}$$

$$\leq \sqrt{\mathbb{E}[\|\nabla F(\hat{\mathbf{w}}^t)\|^2 | \mathbf{w}^t, \mathbf{u}^t] \cdot \mathbb{E}[\|\mathbf{e}^t\|^2 | \mathbf{w}^t, \mathbf{u}^t]} \tag{168}$$

$$\leq \sqrt{8c\delta I^2(4H^2+1)D^2(D^2+\sigma^2) \cdot \eta^2} \tag{169}$$

$$= \eta I \sqrt{8c\delta(4H^2+1)(D^2+\sigma^2)}D. \tag{170}$$

According to Assumption 5 and 6,

$$\mathbb{E}[\|\bar{\mathbf{G}}^t\|^2 | \mathbf{w}^t, \mathbf{u}^t] = \mathbb{E}\left[ \left\| \frac{1}{I|\mathcal{G}|} \sum_{k\in\mathcal{G}} \sum_{j=0}^{I-1} \nabla f_{i_k^{t,j}}(\mathbf{w}_k^{t+1,j}) \right\|^2 \middle| \mathbf{w}^t, \mathbf{u}^t \right] \tag{171}$$

$$\leq \frac{1}{I|\mathcal{G}|} \sum_{k\in\mathcal{G}} \sum_{j=0}^{I-1} \mathbb{E}\left[ \left\| \nabla f_{i_k^{t,j}}(\mathbf{w}_k^{t+1,j}) \right\|^2 \middle| \mathbf{w}^t, \mathbf{u}^t \right] \tag{172}$$

$$\leq \frac{1}{I|\mathcal{G}|} \sum_{k\in\mathcal{G}} \sum_{j=0}^{I-1} (D^2+\sigma^2) \tag{173}$$

$$= D^2 + \sigma^2. \tag{174}$$

Substituting (137), (154), (166), (170) and (174) into (150), we have:

$$\mathbb{E}[F(\hat{\mathbf{w}}^{t+1})|\mathbf{w}^t, \mathbf{u}^t] \leq F(\hat{\mathbf{w}}^t) - \eta I \cdot \|\nabla F(\hat{\mathbf{w}}^t)\|^2 - \eta I \cdot \mathbb{E}\Big[\nabla F(\hat{\mathbf{w}}^t)^T[\bar{\mathbf{G}}^t - \nabla F(\hat{\mathbf{w}}^t)]\Big|\mathbf{w}^t, \mathbf{u}^t\Big]$$
$$- \mathbb{E}[\nabla F(\hat{\mathbf{w}}^t)^T \mathbf{e}^t | \mathbf{w}^t, \mathbf{u}^t] + \eta^2 I^2 L \cdot \mathbb{E}[\|\bar{\mathbf{G}}^t\|^2 | \mathbf{w}^t, \mathbf{u}^t] + L \cdot \mathbb{E}[\|\mathbf{e}^t\|^2 | \mathbf{w}^t, \mathbf{u}^t] \tag{175}$$

$$\leq F(\hat{\mathbf{w}}^t) - \frac{\eta I}{2} \|\nabla F(\mathbf{w}^t)\|^2 + \eta I L^2 \|\mathbf{u}^t\|^2$$
$$+ \eta I \Big[2(I-1)\eta DL^2 + DL\|\mathbf{u}^t\| + BD\Big]$$
$$+ \eta I \sqrt{8c\delta(4H^2+1)(D^2+\sigma^2)}D$$
$$+ \eta^2 I^2 L(D^2+\sigma^2) + L \cdot [8c\delta I^2(4H^2+1)(D^2+\sigma^2) \cdot \eta^2]. \tag{176}$$

Note that $\mathbb{E}\|\mathbf{u}^t\| = \sqrt{[\mathbb{E}\|\mathbf{u}^t\|]^2} \leq \sqrt{[\mathbb{E}\|\mathbf{u}^t\|^2]}$. Taking total expectation on both sides and using that $\mathbb{E}\|\mathbf{u}^t\|^2 \leq 4H^2 I^2(D^2+\sigma^2) \cdot \eta^2$, we have:

$$\mathbb{E}[F(\hat{\mathbf{w}}^{t+1})] \leq \mathbb{E}[F(\hat{\mathbf{w}}^t)] - \frac{\eta I}{2} \mathbb{E}\|\nabla F(\mathbf{w}^t)\|^2 + \eta I L^2[4H^2 I^2(D^2+\sigma^2) \cdot \eta^2]$$
$$+ \eta I \Big[2(I-1)\eta DL^2 + DL \cdot \sqrt{4H^2 I^2(D^2+\sigma^2) \cdot \eta^2} + BD\Big]$$
$$+ \eta I \sqrt{8c\delta(4H^2+1)(D^2+\sigma^2)}D$$
$$+ \eta^2 I^2 L(D^2+\sigma^2) + L \cdot [8c\delta I^2(4H^2+1)(D^2+\sigma^2) \cdot \eta^2]. \tag{177}$$

Namely,

$$
\begin{aligned}
\mathbb{E}[F(\hat{\mathbf{w}}^{t+1})] \leq\ & \mathbb{E}[F(\hat{\mathbf{w}}^{t})] - \frac{\eta I}{2}\mathbb{E}\|\nabla F(\mathbf{w}^{t})\|^{2} \\
& + (\eta I)^{2}L\Big[2(1-I^{-1})DL + 2HD\sqrt{D^{2}+\sigma^{2}} + (D^{2}+\sigma^{2}) + 8c\delta(4H^{2}+1)(D^{2}+\sigma^{2})\Big] \\
& + (\eta I)^{3}\Big[4H^{2}L^{2}(D^{2}+\sigma^{2})\Big] + (\eta I)\Big[BD + \sqrt{8c\delta(4H^{2}+1)(D^{2}+\sigma^{2})}D\Big].
\end{aligned}
\tag{178}
$$

By taking summation from $t = 0$ to $T-1$, we have:

$$
\begin{aligned}
\mathbb{E}[F(\hat{\mathbf{w}}^{T})] \leq\ & \mathbb{E}[F(\hat{\mathbf{w}}^{0})] - \frac{\eta I}{2}\cdot\sum_{t=0}^{T-1}\mathbb{E}\|\nabla F(\mathbf{w}^{t})\|^{2} \\
& + T(\eta I)^{2}L\Big[2(1-I^{-1})DL + 2HD\sqrt{D^{2}+\sigma^{2}} + (D^{2}+\sigma^{2}) + 8c\delta(4H^{2}+1)(D^{2}+\sigma^{2})\Big] \\
& + T(\eta I)^{3}\Big[4H^{2}L^{2}(D^{2}+\sigma^{2})\Big] + T(\eta I)\Big[BD + \sqrt{8c\delta(4H^{2}+1)(D^{2}+\sigma^{2})}D\Big].
\end{aligned}
\tag{179}
$$

Note that $\hat{\mathbf{w}}^{0} = \mathbf{w}^{0}$ and $F(\hat{\mathbf{w}}^{T}) \geq F^{*}$. Thus,

$$
\begin{aligned}
\frac{1}{T}\sum_{t=0}^{T-1}\mathbb{E}\|\nabla F(\mathbf{w}^{t})\|^{2} \leq\ & \frac{2[F(\hat{\mathbf{w}}^{0}) - F^{*}]}{\eta I T} \\
& + \eta \cdot 2IL\Big[2(1-I^{-1})DL + 2HD\sqrt{D^{2}+\sigma^{2}} + (D^{2}+\sigma^{2}) + 8c\delta(4H^{2}+1)(D^{2}+\sigma^{2})\Big] \\
& + \eta^{2}\cdot\Big[8H^{2}I^{2}L^{2}(D^{2}+\sigma^{2})\Big] + 2\Big[BD + \sqrt{8c\delta(4H^{2}+1)(D^{2}+\sigma^{2})}D\Big].
\end{aligned}
\tag{180}
$$

In summary,

$$
\frac{1}{T}\sum_{t=0}^{T-1}\mathbb{E}\|\nabla F(\mathbf{w}^{t})\|^{2} \leq \frac{2[F(\hat{\mathbf{w}}^{0}) - F^{*}]}{\eta I T} + \eta\gamma_{1} + \eta^{2}\gamma_{2} + \Delta,
\tag{181}
$$

where $\gamma_{1} = 2IL\cdot[2(1-I^{-1})LD + 2HD\sqrt{D^{2}+\sigma^{2}} + (D^{2}+\sigma^{2}) + 8c\delta(4H^{2}+1)(D^{2}+\sigma^{2})]$, $\gamma_{2} = 8H^{2}I^{2}L^{2}(D^{2}+\sigma^{2})$ and $\Delta = 2BD + 4\sqrt{2c\delta(4H^{2}+1)(D^{2}+\sigma^{2})}D$.

$\square$

## B.8 ANALYSIS FOR LOCAL MOMENTUM SGD

We present the following proposition, which illustrates that Assumption 7 holds when $\mathcal{A}$ is set to be local momentum SGD.

**Proposition 3.** *Under Assumption 3, 5 and 6, local momentum SGD satisfies Assumption 7. Moreover, for local momentum SGD with learning rate $\eta > 0$, update interval $I \in \mathbb{N}_{+}$ and momentum hyper-parameter $\beta \in [0, 1)$, we have $\eta_{\mathcal{A}} = \eta I$, $A_{1} = \frac{\beta(1-\beta^{I})}{I(1-\beta)}D + \sqrt{D^{2}+\sigma^{2}} + (\frac{I-1}{2} + \frac{\beta^{2}(1-\beta^{I-1})}{I(1-\beta)^{2}} - \frac{\beta(I-1)}{I(1-\beta)})\cdot L\sqrt{D^{2}+\sigma^{2}}$ and $(A_{2})^{2} = D^{2} + \sigma^{2}$.*

*Proof.* When $\mathcal{A}$ is set to be local momentum SGD with learning rate $\eta$, update interval $I$ and momentum hyper-parameter $\beta$, let $\mathbf{m}_{k}^{0,j} = \mathbf{0}$ be the initial momentum and $\mathbf{w}_{k}^{t+1}$ ($t = 0, 1, \ldots, T-1$) is computed by the following process:

$$
\begin{cases}
\mathbf{m}_{k}^{t+1,0} = \mathbf{m}_{k}^{t,I}; \\
\mathbf{w}_{k}^{t+1,0} = \mathbf{w}^{t}; \\
\mathbf{m}_{k}^{t+1,j+1} = \beta\cdot\mathbf{m}_{k}^{t+1,j} + (1-\beta)\cdot\nabla f_{i_{k}^{t,j}}(\mathbf{w}_{k}^{t+1,j}), & j = 0, 1, \ldots, I-1; \\
\mathbf{w}_{k}^{t+1,j+1} = \mathbf{w}_{k}^{t+1,j} - \eta\cdot\mathbf{m}_{k}^{t+1,j+1}, & j = 0, 1, \ldots, I-1; \\
\mathbf{w}_{k}^{t+1} = \mathbf{w}_{k}^{t+1,I}.
\end{cases}
\tag{182}
$$

Let $\eta_{\mathcal{A}} = \eta I$, we have

$$\mathbf{G}_{\mathcal{A}}(\mathbf{w}^t; \mathcal{D}_k) = (\mathbf{w}^t - \mathbf{w}_k^{t+1})/(\eta I) = \frac{1}{I}\sum_{j=0}^{I-1}(\mathbf{w}_k^{t+1,j} - \mathbf{w}_k^{t+1,j+1}) = \frac{1}{I}\sum_{j=0}^{I-1}\mathbf{m}_k^{t+1,j+1}. \quad (183)$$

In addition,

$$\mathbf{m}_k^{t+1,j+1} = \beta \cdot \mathbf{m}_k^{t+1,j} + (1-\beta) \cdot \nabla f_{i_k^{t,j}}(\mathbf{w}_k^{t+1,j}) \quad (184)$$

$$= \beta \cdot (\beta \cdot \mathbf{m}_k^{t+1,j-1} + (1-\beta) \cdot \nabla f_{i_k^{t,j-1}}(\mathbf{w}_k^{t+1,j-1})) + (1-\beta) \cdot \nabla f_{i_k^{t,j}}(\mathbf{w}_k^{t+1,j}) \quad (185)$$

$$= \beta^2 \cdot \mathbf{m}_k^{t+1,j-1} + \beta(1-\beta) \cdot \nabla f_{i_k^{t,j-1}}(\mathbf{w}_k^{t+1,j-1}) + (1-\beta) \cdot \nabla f_{i_k^{t,j}}(\mathbf{w}_k^{t+1,j}) \quad (186)$$

$$= \ldots\ldots\ldots\ldots$$

$$= \beta^{j+1}\mathbf{m}_k^{t+1,0} + (1-\beta)\sum_{j'=0}^{j}\beta^{j-j'}\nabla f_{i_k^{t,j'}}(\mathbf{w}_k^{t+1,j'}). \quad (187)$$

Now we prove that $\mathbb{E}\|\mathbf{m}_k^{t,j}\|^2 \leq D^2 + \sigma^2$ $(j = 0, 1, \ldots, I)$ by deduction on $t$.

Step 1. When $t = 0$, we have $\mathbb{E}\|\mathbf{m}_k^{0,j}\|^2 = 0 \leq D^2 + \sigma^2$.

Step 2 (deduction). Suppose $\mathbb{E}\|\mathbf{m}_k^{t,j}\|^2 \leq D^2 + \sigma^2$, we have $\mathbb{E}\|\mathbf{m}_k^{t+1,0}\|^2 = \mathbb{E}\|\mathbf{m}_k^{t,I}\|^2 \leq D^2 + \sigma^2$ and $\mathbb{E}\|\nabla f_{i_k^{t,j'}}(\mathbf{w}_k^{t+1,j'})\|^2 \leq D^2 + \sigma^2$. Since

$$\beta^{j+1} + (1-\beta)\sum_{j'=0}^{j}\beta^{j-j'} = \beta^{j+1} + (1-\beta)\frac{1-\beta^{j+1}}{1-\beta} = 1, \quad (188)$$

$\mathbf{m}_k^{t+1,j+1}$ can be deemed as a weighted averaging of $\mathbf{m}_k^{t+1,0}$ and $\{\nabla f_{i_k^{t,j'}}(\mathbf{w}_k^{t+1,j'})\}_{j'=0}^{j}$. Thus,

$$\mathbb{E}\|\mathbf{m}_k^{t+1,j+1}\|^2 \leq D^2 + \sigma^2, \quad j = 1, 2, \ldots, I. \quad (189)$$

By mathematical deduction, $\forall t = 0, 1, \ldots, T$, we have

$$\mathbb{E}\|\mathbf{m}_k^{t,j}\|^2 \leq D^2 + \sigma^2, \quad j = 0, 1, 2, \ldots, I. \quad (190)$$

Therefore,

$$\mathbb{E}\|\mathbf{G}_{\mathcal{A}}(\mathbf{w}^t; \mathcal{D}_k)\|^2 = \mathbb{E}\left\|\frac{1}{I}\sum_{j=0}^{I-1}\mathbf{m}_k^{t+1,j+1}\right\|^2 \leq D^2 + \sigma^2. \quad (191)$$

Substituting (187) into (183), we have:

$$\mathbf{G}_{\mathcal{A}}(\mathbf{w}^t; \mathcal{D}_k) = \frac{1}{I}\sum_{j=0}^{I-1}\left[\beta^{j+1}\mathbf{m}_k^{t+1,0} + (1-\beta)\sum_{j'=0}^{j}\beta^{j-j'}\nabla f_{i_k^{t,j'}}(\mathbf{w}_k^{t+1,j'})\right] \quad (192)$$

$$= \frac{\beta(1-\beta^I)}{I(1-\beta)}\mathbf{m}_k^{t+1,0} + \frac{1-\beta}{I}\sum_{j=0}^{I-1}\left[\sum_{j'=0}^{j}\beta^{j-j'}\nabla f_{i_k^{t,j'}}(\mathbf{w}_k^{t+1,j'})\right] \quad (193)$$

$$= \frac{\beta(1-\beta^I)}{I(1-\beta)}\mathbf{m}_k^{t+1,0} + \frac{1-\beta}{I}\sum_{j'=0}^{I-1}\left[\sum_{j=j'}^{I-1}\beta^{j-j'}\nabla f_{i_k^{t,j'}}(\mathbf{w}_k^{t+1,j'})\right] \quad (194)$$

$$= \frac{\beta(1-\beta^I)}{I(1-\beta)}\mathbf{m}_k^{t+1,0} + \frac{1-\beta}{I}\sum_{j'=0}^{I-1}\left[\frac{1-\beta^{I-j'}}{1-\beta}\nabla f_{i_k^{t,j'}}(\mathbf{w}_k^{t+1,j'})\right] \quad (195)$$

$$= \frac{1}{I}\left[\frac{\beta(1-\beta^I)}{1-\beta}\mathbf{m}_k^{t+1,0} + \sum_{j'=0}^{I-1}(1-\beta^{I-j'})\nabla f_{i_k^{t,j'}}(\mathbf{w}_k^{t+1,j'})\right]. \quad (196)$$

Therefore,

$$\mathbb{E}[\mathbf{G}_{\mathcal{A}}(\mathbf{w}^t; \mathcal{D}_k) - \nabla F_k(\mathbf{w}^t)] = \frac{1}{I}\left[\frac{\beta(1-\beta^I)}{1-\beta} \cdot \mathbb{E}[\mathbf{m}_k^{t+1,0} - \nabla F_k(\mathbf{w}^t)]\right.$$
$$\left. + \sum_{j'=0}^{I-1}(1-\beta^{I-j'}) \cdot \mathbb{E}[\nabla f_{i_k^{t,j'}}(\mathbf{w}_k^{t+1,j'}) - \nabla F_k(\mathbf{w}^t)]\right]. \quad (197)$$

Since $\mathbb{E}\|\mathbf{m}_k^{t+1,0} - \nabla F_k(\mathbf{w}^t)\| \leq \sqrt{D^2 + \sigma^2} + D$ and that

$$\mathbb{E}\|\nabla f_{i_k^{t,j'}}(\mathbf{w}_k^{t+1,j'}) - \nabla F_k(\mathbf{w}^t)\|$$
$$\leq \mathbb{E}\|\nabla f_{i_k^{t,j'}}(\mathbf{w}_k^{t+1,j'}) - \nabla F_k(\mathbf{w}_k^{t+1,j'})\| + \mathbb{E}\|\nabla F_k(\mathbf{w}_k^{t+1,j'}) - \nabla F_k(\mathbf{w}^t)\| \quad (198)$$
$$\leq \sqrt{D^2 + \sigma^2} + L \cdot \|\mathbf{w}_k^{t+1,j'} - \mathbf{w}^t\| \quad (199)$$
$$\leq \sqrt{D^2 + \sigma^2} + L\sum_{j''=0}^{j'-1}\|\mathbf{m}_k^{t+1,j''+1}\| \quad (200)$$
$$\leq \sqrt{D^2 + \sigma^2} + j'L\sqrt{D^2 + \sigma^2}, \quad (201)$$

we have

$$\|\mathbb{E}[\mathbf{G}_{\mathcal{A}}(\mathbf{w}^t; \mathcal{D}_k)] - \nabla F_k(\mathbf{w}^t)\|$$
$$\leq \frac{1}{I}\left[\frac{\beta(1-\beta^I)}{1-\beta} \cdot [\sqrt{D^2 + \sigma^2} + D] + \sum_{j'=0}^{I-1}(1-\beta^{I-j'}) \cdot [\sqrt{D^2 + \sigma^2} + j'L\sqrt{D^2 + \sigma^2}]\right] \quad (202)$$
$$= \frac{1}{I}\left[\frac{\beta(1-\beta^I)}{1-\beta}D + I\sqrt{D^2 + \sigma^2} + L\sqrt{D^2 + \sigma^2} \cdot \sum_{j'=0}^{I-1}(j' - j'\beta^{I-j'})\right] \quad (203)$$
$$= \frac{1}{I}\left[\frac{\beta(1-\beta^I)}{1-\beta}D + I\sqrt{D^2 + \sigma^2} + L\sqrt{D^2 + \sigma^2} \cdot \left(\frac{I(I-1)}{2} + \frac{\beta^2(1-\beta^{I-1})}{(1-\beta)^2} - \frac{\beta(I-1)}{1-\beta}\right)\right] \quad (204)$$
$$= \frac{\beta(1-\beta^I)}{I(1-\beta)}D + \sqrt{D^2 + \sigma^2} + \left(\frac{I-1}{2} + \frac{\beta^2(1-\beta^{I-1})}{I(1-\beta)^2} - \frac{\beta(I-1)}{I(1-\beta)}\right) \cdot L\sqrt{D^2 + \sigma^2}. \quad (205)$$

$$\square$$

## B.9 Proof of Theorem 4

*Proof.* Similar to Lemma 1 and Lemma 2, we have the following inequalities to bound the local memory and the aggregation error, respectively, for general training algorithm $\mathcal{A}$ that satisfies Assumption 7:

$$\mathbb{E}\|\mathbf{u}_k^{t+1}\|^2 = \mathbb{E}\|\mathbf{g}_k^t - \tilde{\mathbf{g}}_k^t\|^2 \quad (206)$$
$$\overset{(i)}{\leq} \left(1 - \frac{d'_{cons}}{d}\right)\mathbb{E}\|\mathbf{g}_k^t\|^2 \quad (207)$$
$$= \left(1 - \frac{d'_{cons}}{d}\right)\mathbb{E}\|\mathbf{u}_k^t + (\mathbf{w}^t - \mathbf{w}_k^{t+1})\|^2 \quad (208)$$
$$\overset{(ii)}{\leq} \left(1 - \frac{d'_{cons}}{d}\right)\left[(1 + \frac{d'_{cons}}{2d})\mathbb{E}\|\mathbf{u}_k^t\|^2 + (1 + \frac{2d}{d'_{cons}})\mathbb{E}\|\eta_{\mathcal{A}} \cdot \mathbf{G}_{\mathcal{A}}(\mathbf{w}^t; \mathcal{D}_k)\|^2\right] \quad (209)$$
$$\overset{(iii)}{\leq} \left(1 - \frac{d'_{cons}}{2d}\right)\mathbb{E}\|\mathbf{u}_k^t\|^2 + \frac{2d}{d'_{cons}}(\eta_{\mathcal{A}})^2 \cdot \mathbb{E}\|\mathbf{G}_{\mathcal{A}}(\mathbf{w}^t; \mathcal{D}_k)\|^2 \quad (210)$$
$$\overset{(iv)}{\leq} \left(1 - \frac{d'_{cons}}{2d}\right)\mathbb{E}\|\mathbf{u}_k^t\|^2 + \frac{2d}{d'_{cons}}(\eta_{\mathcal{A}})^2(A_2)^2, \quad (211)$$

where (i) is derived based on Proposition 2. (ii) is derived based on that $\|\mathbf{x} + \mathbf{y}\|^2 \leq (1+\theta)\|\mathbf{x}\|^2 + (1+\theta^{-1})\|\mathbf{y}\|^2$ for any constant $\theta > 0$. (iii) is derived based on that $(1 - \frac{d'_{cons}}{d})(1 + \frac{d'_{cons}}{2d}) < 1 - \frac{d'_{cons}}{2d}$ and $(1 - \frac{d'_{cons}}{d})(1 + \frac{2d}{d'_{cons}}) < \frac{2d}{d'_{cons}}$. (iv) is derived based on Assumption 7. Therefore,

$$\left( \mathbb{E}\|\mathbf{u}_k^{t+1}\|^2 - \frac{4d^2}{(d'_{cons})^2}(\eta_{\mathcal{A}})^2(A_2)^2 \right) \leq \left( 1 - \frac{d'_{cons}}{2d} \right) \cdot \left( \mathbb{E}\|\mathbf{u}_k^t\|^2 - \frac{4d^2}{(d'_{cons})^2}(\eta_{\mathcal{A}})^2(A_2)^2 \right). \tag{212}$$

Recursively using (212), we have

$$\left( \mathbb{E}\|\mathbf{u}_k^t\|^2 - \frac{4d^2}{(d'_{cons})^2}(\eta_{\mathcal{A}})^2(A_2)^2 \right) \leq \left( 1 - \frac{d'_{cons}}{2d} \right)^t \cdot \left( \mathbb{E}\|\mathbf{u}_k^0\|^2 - \frac{4d^2}{(d'_{cons})^2}(\eta_{\mathcal{A}})^2(A_2)^2 \right) < 0. \tag{213}$$

Thus,

$$\mathbb{E}\|\mathbf{u}_k^t\|^2 \leq \frac{4d^2(A_2)^2}{(d'_{cons})^2} \cdot (\eta_{\mathcal{A}})^2. \tag{214}$$

Let $H = d'_{cons}/d$. Finally, it is obtained that

$$\mathbb{E}\|\mathbf{u}^t\|^2 = \mathbb{E}\|\frac{1}{|\mathcal{G}|}\sum_{k\in\mathcal{G}}\mathbf{u}_k^t\|^2 \leq \frac{1}{|\mathcal{G}|}\sum_{k\in\mathcal{G}}\mathbb{E}\|\mathbf{u}_k^t\|^2 \leq \frac{4d^2(A_2)^2}{(d'_{cons})^2} \cdot (\eta_{\mathcal{A}})^2 = 4H^2(A_2)^2(\eta_{\mathcal{A}})^2. \tag{215}$$

Based on Assumption 7, we have that $\forall k \in \mathcal{G}$,

$$\mathbb{E}\|\tilde{\mathbf{g}}_k^t\|^2 \leq \mathbb{E}\|\mathbf{g}_k^t\|^2 = \mathbb{E}\|\mathbf{u}_k^t + (\mathbf{w}^t - \mathbf{w}_k^{t+1})\|^2 \tag{216}$$
$$\leq 2\mathbb{E}\|\mathbf{u}_k^t\|^2 + 2\mathbb{E}\|\mathbf{w}^t - \mathbf{w}_k^{t+1}\|^2 \tag{217}$$
$$\leq 2\mathbb{E}\|\mathbf{u}_k^t\|^2 + 2(\eta_{\mathcal{A}})^2(A_2)^2 \tag{218}$$
$$\leq 2(4H^2 + 1)(A_2)^2 \cdot (\eta_{\mathcal{A}})^2. \tag{219}$$

Thus,

$$\mathbb{E}_{k\neq k'}\left[ \|\tilde{\mathbf{g}}_k^t - \tilde{\mathbf{g}}_{k'}^t\|^2 \right] \leq 2\mathbb{E}\|\tilde{\mathbf{g}}_k^t\|^2 + 2\mathbb{E}\|\tilde{\mathbf{g}}_{k'}^t\|^2 \leq 8(4H^2 + 1)(A_2)^2 \cdot (\eta_{\mathcal{A}})^2. \tag{220}$$

Therefore, by Definition 1 and (220),

$$\mathbb{E}\|\mathbf{e}^t\|^2 = \mathbb{E}\left\|\mathbf{SRAgg}(\{\tilde{\mathbf{g}}_k^t\}_{k=1}^m) - \frac{1}{|\mathcal{G}|}\sum_{k\in\mathcal{G}}\tilde{\mathbf{g}}_k^t\right\|^2 \tag{221}$$
$$\leq c\delta \cdot \mathbb{E}_{k\neq k'}\left[ \|\tilde{\mathbf{g}}_k^t - \tilde{\mathbf{g}}_{k'}^t\|^2 \right] \tag{222}$$
$$\leq 8c\delta(4H^2 + 1)(A_2)^2 \cdot (\eta_{\mathcal{A}})^2. \tag{223}$$

Let $\bar{\mathbf{w}}^{t+1} = \frac{1}{|\mathcal{G}|}\sum_{k\in\mathcal{G}}\mathbf{w}_k^{t+1}$. Combining with Equation (142), we have:

$$\hat{\mathbf{w}}^{t+1} = \hat{\mathbf{w}}^t - (\mathbf{w}^t - \bar{\mathbf{w}}^{t+1}) - \mathbf{e}^t. \tag{224}$$

The equation can be interpreted as that $\hat{\mathbf{w}}^{t+1}$ is obtained by adding a small term $-(\mathbf{w}^t - \bar{\mathbf{w}}^{t+1})$ on $\hat{\mathbf{w}}^t$ with error $\mathbf{e}^t$. Therefore,

$$F(\hat{\mathbf{w}}^{t+1}) = F(\hat{\mathbf{w}}^t - (\mathbf{w}^t - \bar{\mathbf{w}}^{t+1}) - \mathbf{e}^t) \tag{225}$$
$$\leq F(\hat{\mathbf{w}}^t) - \nabla F(\hat{\mathbf{w}}^t)^T(\mathbf{w}^t - \bar{\mathbf{w}}^{t+1} + \mathbf{e}^t) + \frac{L}{2}\|\mathbf{w}^t - \bar{\mathbf{w}}^{t+1} + \mathbf{e}^t\|^2 \tag{226}$$
$$\leq F(\hat{\mathbf{w}}^t) - \nabla F(\hat{\mathbf{w}}^t)^T(\mathbf{w}^t - \bar{\mathbf{w}}^{t+1}) - \nabla F(\hat{\mathbf{w}}^t)^T\mathbf{e}^t + \eta^2 I^2 L\|\mathbf{w}^t - \bar{\mathbf{w}}^{t+1}\|^2 + L\|\mathbf{e}^t\|^2 \tag{227}$$
$$= F(\hat{\mathbf{w}}^t) - \frac{1}{|\mathcal{G}|}\sum_{k\in\mathcal{G}}\nabla F(\hat{\mathbf{w}}^t)^T(\mathbf{w}^t - \mathbf{w}_k^{t+1})$$
$$- \nabla F(\hat{\mathbf{w}}^t)^T\mathbf{e}^t + L\left\|\mathbf{w}^t - \frac{1}{|\mathcal{G}|}\sum_{k\in\mathcal{G}}\mathbf{w}_k^{t+1}\right\|^2 + L\|\mathbf{e}^t\|^2 \tag{228}$$

$$\leq F(\hat{\mathbf{w}}^t) - \eta_{\mathcal{A}} \|\nabla F(\hat{\mathbf{w}}^t)\|^2 - \frac{1}{|\mathcal{G}|} \sum_{k \in \mathcal{G}} \nabla F(\hat{\mathbf{w}}^t)^T [\mathbf{w}^t - \mathbf{w}_k^{t+1} - \eta_{\mathcal{A}} \cdot \nabla F(\hat{\mathbf{w}}^t)]$$

$$- \nabla F(\hat{\mathbf{w}}^t)^T \mathbf{e}^t + \frac{L}{|\mathcal{G}|} \sum_{k \in \mathcal{G}} \left\| \mathbf{w}^t - \mathbf{w}_k^{t+1} \right\|^2 + L \|\mathbf{e}^t\|^2 \tag{229}$$

$$= F(\hat{\mathbf{w}}^t) - \eta_{\mathcal{A}} \|\nabla F(\hat{\mathbf{w}}^t)\|^2 - \frac{1}{|\mathcal{G}|} \sum_{k \in \mathcal{G}} \nabla F(\hat{\mathbf{w}}^t)^T [\eta_{\mathcal{A}} \cdot \mathbf{G}_{\mathcal{A}}(\mathbf{w}^t; \mathcal{D}_k) - \eta_{\mathcal{A}} \cdot \nabla F(\hat{\mathbf{w}}^t)]$$

$$- \nabla F(\hat{\mathbf{w}}^t)^T \mathbf{e}^t + \frac{L}{|\mathcal{G}|} \sum_{k \in \mathcal{G}} \left\| \eta_{\mathcal{A}} \cdot \mathbf{G}_{\mathcal{A}}(\mathbf{w}^t; \mathcal{D}_k) \right\|^2 + L \|\mathbf{e}^t\|^2. \tag{230}$$

Taking expectation on both sides, we have

$$\mathbb{E}[F(\hat{\mathbf{w}}^{t+1})|\mathbf{w}^t, \mathbf{u}^t]$$

$$\leq F(\hat{\mathbf{w}}^t) - \eta_{\mathcal{A}} \|\nabla F(\hat{\mathbf{w}}^t)\|^2 - \frac{\eta_{\mathcal{A}}}{|\mathcal{G}|} \sum_{k \in \mathcal{G}} \mathbb{E}\left[ \nabla F(\hat{\mathbf{w}}^t)^T [\mathbf{G}_{\mathcal{A}}(\mathbf{w}^t; \mathcal{D}_k) - \nabla F(\hat{\mathbf{w}}^t)] \middle| \mathbf{w}^t, \mathbf{u}^t \right]$$

$$- \mathbb{E}[\nabla F(\hat{\mathbf{w}}^t)^T \mathbf{e}^t | \mathbf{w}^t, \mathbf{u}^t] + \frac{(\eta_{\mathcal{A}})^2 L}{|\mathcal{G}|} \sum_{k \in \mathcal{G}} \mathbb{E}\left[ \left\| \mathbf{G}_{\mathcal{A}}(\mathbf{w}^t; \mathcal{D}_k) \right\|^2 \middle| \mathbf{w}^t, \mathbf{u}^t \right] + L \cdot \mathbb{E}[\|\mathbf{e}^t\|^2 | \mathbf{w}^t, \mathbf{u}^t]. \tag{231}$$

By using Assumption 3, Assumption 4, Assumption 7, we have:

$$- \mathbb{E}\left[ \nabla F(\hat{\mathbf{w}}^t)^T [\mathbf{G}_{\mathcal{A}}(\mathbf{w}^t; \mathcal{D}_k) - \nabla F(\hat{\mathbf{w}}^t)] \middle| \mathbf{w}^t, \mathbf{u}^t \right]$$

$$= - \nabla F(\hat{\mathbf{w}}^t)^T \left[ \mathbb{E}[\mathbf{G}_{\mathcal{A}}(\mathbf{w}^t; \mathcal{D}_k)|\mathbf{w}^t, \mathbf{u}^t] - \nabla F(\hat{\mathbf{w}}^t) \right] \tag{232}$$

$$\leq \|\nabla F(\hat{\mathbf{w}}^t)\| \cdot \left\| \mathbb{E}[\mathbf{G}_{\mathcal{A}}(\mathbf{w}^t; \mathcal{D}_k)|\mathbf{w}^t, \mathbf{u}^t] - \nabla F(\hat{\mathbf{w}}^t) \right\| \tag{233}$$

$$\leq \|\nabla F(\hat{\mathbf{w}}^t)\| \cdot \left\{ \left\| \mathbb{E}[\mathbf{G}_{\mathcal{A}}(\mathbf{w}^t; \mathcal{D}_k)|\mathbf{w}^t, \mathbf{u}^t] - \nabla F_k(\mathbf{w}^t) \right\| \right.$$

$$\left. + \|\nabla F_k(\mathbf{w}^t) - \nabla F(\mathbf{w}^t)\| + \|\nabla F(\mathbf{w}^t) - \nabla F(\hat{\mathbf{w}}^t)\| \right\} \tag{234}$$

$$\leq D \cdot (A_1 + B + L \|\mathbf{w}^t - \hat{\mathbf{w}}^t\|) \tag{235}$$

$$= A_1 D + BD + DL \|\mathbf{u}^t\|. \tag{236}$$

Note that $\mathbb{E}[XY] \leq \sqrt{\mathbb{E}[X^2]\mathbb{E}[Y^2]}$. Based on Assumption 5, Assumption 7 and (223), we have:

$$-\mathbb{E}[\nabla F(\hat{\mathbf{w}}^t)^T \mathbf{e}^t | \mathbf{w}^t, \mathbf{u}^t] \leq \mathbb{E}[\|\nabla F(\hat{\mathbf{w}}^t)\| \cdot \|\mathbf{e}^t\| | \mathbf{w}^t, \mathbf{u}^t] \tag{237}$$

$$\leq \sqrt{\mathbb{E}[\|\nabla F(\hat{\mathbf{w}}^t)\|^2 | \mathbf{w}^t, \mathbf{u}^t] \cdot \mathbb{E}[\|\mathbf{e}^t\|^2 | \mathbf{w}^t, \mathbf{u}^t]} \tag{238}$$

$$\leq \sqrt{D^2 \cdot 8c\delta(4H^2 + 1)(A_2)^2(\eta_{\mathcal{A}})^2} \tag{239}$$

$$= \eta_{\mathcal{A}} \cdot \sqrt{8c\delta(4H^2 + 1)} A_2 D. \tag{240}$$

According to Assumption 7,

$$\mathbb{E}[\|\mathbf{G}_{\mathcal{A}}(\mathbf{w}^t; \mathcal{D}_k)\|^2 | \mathbf{w}^t, \mathbf{u}^t] \leq (A_2)^2. \tag{241}$$

Substituting (154), (223), (236), (240) and (241) into (231), we have:

$$\mathbb{E}[F(\hat{\mathbf{w}}^{t+1})|\mathbf{w}^t, \mathbf{u}^t] \leq F(\hat{\mathbf{w}}^t) - \frac{\eta_{\mathcal{A}}}{2} \|\nabla F(\mathbf{w}^t)\|^2 + \eta_{\mathcal{A}} L^2 \|\mathbf{u}^t\|^2$$

$$+ \eta_{\mathcal{A}} \left[ A_1 D + BD + DL \|\mathbf{u}^t\| \right] + \eta_{\mathcal{A}} \cdot \sqrt{8c\delta(4H^2 + 1)} A_2 D$$

$$+ (\eta_{\mathcal{A}})^2 L(A_2)^2 + L \cdot [8c\delta(4H^2 + 1)(A_2)^2 \cdot (\eta_{\mathcal{A}})^2]. \tag{242}$$

Note that $\mathbb{E}\|\mathbf{u}^t\| = \sqrt{[\mathbb{E}\|\mathbf{u}^t\|]^2} \leq \sqrt{[\mathbb{E}\|\mathbf{u}^t\|^2]}$ and that $\mathbb{E}\|\mathbf{u}^t\|^2 = 4H^2(A_2)^2(\eta_{\mathcal{A}})^2$. Taking total expectation on both sides, we have:

$$\mathbb{E}[F(\hat{\mathbf{w}}^{t+1})] \leq \mathbb{E}[F(\hat{\mathbf{w}}^t)] - \frac{\eta_{\mathcal{A}}}{2} \mathbb{E}\|\nabla F(\mathbf{w}^t)\|^2 + \eta_{\mathcal{A}} L^2 [4H^2(A_2)^2(\eta_{\mathcal{A}})^2]$$

$$+ \eta_{\mathcal{A}} \Big[ (A_1 D + BD + 2HA_2 DL\eta_{\mathcal{A}}) + \sqrt{8c\delta(4H^2 + 1)}A_2 D \Big]$$
$$+ (\eta_{\mathcal{A}})^2 L(A_2)^2 + L \cdot [8c\delta(4H^2 + 1)(A_2)^2 \cdot (\eta_{\mathcal{A}})^2]. \tag{243}$$

Namely,

$$\mathbb{E}[F(\hat{\mathbf{w}}^{t+1})] \le \mathbb{E}[F(\hat{\mathbf{w}}^t)] - \frac{\eta_{\mathcal{A}}}{2}\mathbb{E}\|\nabla F(\mathbf{w}^t)\|^2 + \eta_{\mathcal{A}}[A_1 D + BD + \sqrt{8c\delta(4H^2 + 1)}A_2 D]$$
$$+ (\eta_{\mathcal{A}})^3 [4H^2 (A_2)^2 L^2] + (\eta_{\mathcal{A}})^2 [(A_2)^2 L + 2HA_2 DL + 8c\delta(4H^2 + 1)(A_2)^2 L]. \tag{244}$$

By taking summation from $t = 0$ to $T - 1$, we have:

$$\mathbb{E}[F(\hat{\mathbf{w}}^T)] \le \mathbb{E}[F(\hat{\mathbf{w}}^0)] - \frac{\eta_{\mathcal{A}}}{2} \cdot \sum_{t=0}^{T-1} \mathbb{E}\|\nabla F(\mathbf{w}^t)\|^2 + T\eta_{\mathcal{A}}[A_1 D + BD + \sqrt{8c\delta(4H^2 + 1)}A_2 D]$$
$$+ T(\eta_{\mathcal{A}})^3 [4H^2 (A_2)^2 L^2] + T(\eta_{\mathcal{A}})^2 [(A_2)^2 L + 2HA_2 DL + 8c\delta(4H^2 + 1)(A_2)^2 L]. \tag{245}$$

Note that $\hat{\mathbf{w}}^0 = \mathbf{w}^0$ and $F(\hat{\mathbf{w}}^T) \ge F^*$. Thus,

$$\frac{1}{T} \sum_{t=0}^{T-1} \mathbb{E}\|\nabla F(\mathbf{w}^t)\|^2 \le \frac{2[F(\hat{\mathbf{w}}^0) - F^*]}{\eta_{\mathcal{A}} T} + [2A_1 D + 2BD + 4\sqrt{2c\delta(4H^2 + 1)}A_2 D]$$
$$+ (\eta_{\mathcal{A}})^2 \cdot [8H^2 (A_2)^2 L^2] + \eta_{\mathcal{A}} \cdot [2(A_2)^2 L + 4HA_2 DL + 16c\delta(4H^2 + 1)(A_2)^2 L]. \tag{246}$$

In summary,

$$\frac{1}{T} \sum_{t=0}^{T-1} \mathbb{E}\|\nabla F(\mathbf{w}^t)\|^2 \le \frac{2[F(\hat{\mathbf{w}}^0) - F^*]}{\eta_{\mathcal{A}} T} + \eta_{\mathcal{A}}\gamma_{\mathcal{A},1} + (\eta_{\mathcal{A}})^2 \gamma_{\mathcal{A},2} + \Delta_{\mathcal{A}}, \tag{247}$$

where $\gamma_{\mathcal{A},1} = 2(A_2)^2 L + 4HA_2 DL + 16c\delta(4H^2 + 1)(A_2)^2 L$, $\gamma_{\mathcal{A},2} = 8H^2 (A_2)^2 L^2$ and $\Delta_{\mathcal{A}} = 2A_1 D + 2BD + 4\sqrt{2c\delta(4H^2 + 1)}A_2 D$. $\qquad\square$

## C    MORE EXPERIMENTAL RESULTS

In this section, we present more empirical results, which are consistent to the ones in the main text of this paper and further support our conclusions.

### C.1    MORE EXPERIMENTS ABOUT THE EFFECT OF ALPHA

We present more empirical results about FedREP with aggregators geoMed and TMean in this section. The experimental settings are the same as those in the main text. As illustrated in Figure 3, the empirical results are consistent with that in the main text. In addition, we have also noticed that the performance of FedREP with TMean is not stable enough under ALIE attack. A possible reason is that the aggregator TMean is not robust enough against ALIE attack since FedREP with each of the other two aggregators (geoMed and CClip) has a relatively stable empirical results. We will further study this phenomenon in future works.

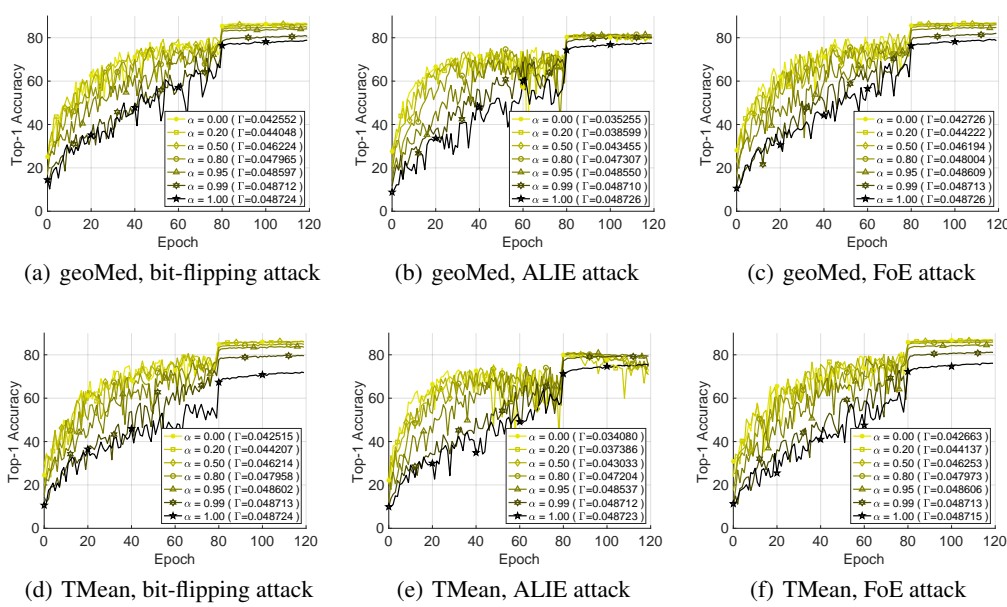

Figure 3: Top-1 accuracy w.r.t. epochs of FedREP with geoMed (top row) and TMean (bottom row) under bit-flipping attack (left column), ALIE attack (middle column) and FoE attack (right column).

### C.2    EXPERIMENTS ABOUT BYZANTINE ATTACKS ON COORDINATES

In each iteration of FedREP, clients will send the coordinate set $\mathcal{I}_k^t$ to server. However, Byzantine clients may send arbitrary coordinates. Although the theoretical analysis in the main text has included this case, we also provide empirical results about Byzantine behaviour on sending coordinates. We set $\alpha = 0$ for non-Byzantine clients and consider four different Byzantine settings, where Byzantine clients send the correct coordinates (noAtk), send the coordinates of $\frac{K}{m}$ smallest absolute values (minAtk), send random coordinates (randAtk) and send coordinates that is the same as a non-Byzantine client (sameAtk), respectively. We set $K = 0.065d$ while the other settings are the same as those in Section 5 in the main text. As illustrated in Figure 4, although the Byzantine attack on coordinates slightly changes the communication cost, it has little effect on the convergence rate and final top-1 accuracy. The main reason is that the top-$\frac{K}{m}$ coordinates of each non-Byzantine client can always be sent to server in FedREP, no matter what is sent from Byzantine clients.

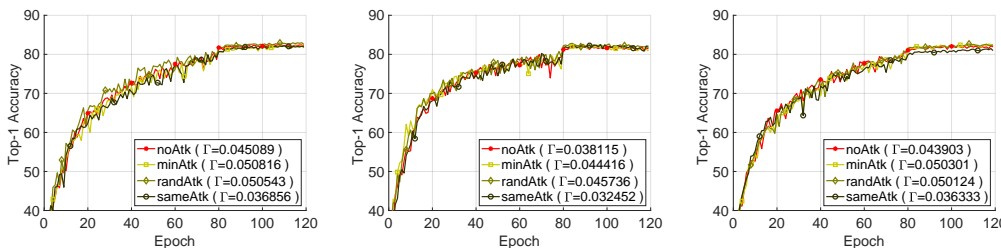

Figure 4: Top-1 accuracy w.r.t. epochs of FedREP with different Byzantine behaviour on sending coordinates when there are 7 Byzantine clients with bit-flipping attack (left), ALIE attack (middle) and FoE attack (right), respectively.

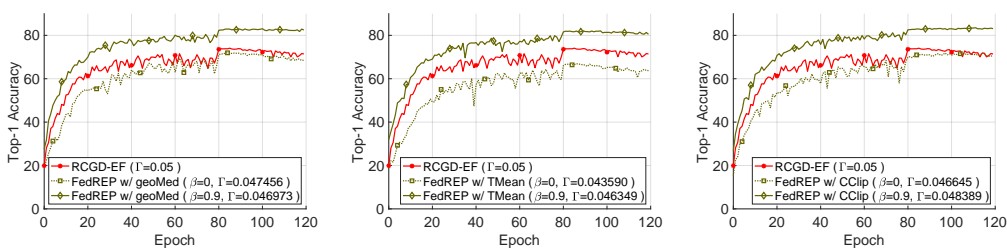

Figure 5: Top-1 accuracy w.r.t. epochs when there are 7 Byzantine clients with ALIE attack. $\beta$ is the hyper-parameter of local momentum. Local momentum is not used when $\beta = 0$. The robust aggregator in FedREP is set to be geoMed (left), TMean (middle) and CClip (right), respectively.

### C.3   EXPERIMENTS ABOUT LOCAL MOMENTUM

Previous works (Karimireddy et al., 2021) have shown that using momentum can help to reduce the variance of stochastic gradients and obtain stronger Byzantine robustness. We also provide empirical results about the momentum in this section. The experimental settings keep the same as in Section 5 in the main text. As illustrated in Figure 5, using local momentum can make FedREP more robust to Byzantine attack ALIE, which is consistent with previous works (Karimireddy et al., 2021).

### C.4   COMPARISON WITH SPARSESECAGG

We first empirically compare the performance of FedREP with the communication-efficient privacy-preserving FL baseline SparseSecAgg (Ergun et al., 2021) when there is no attack. We test the performance of FedREP with buffer size $s = 4$, 8 and 16, respectively. We set $\Gamma = 0.05$ and 0.1 for SparseSecAgg in the two experiments, respectively. Correspondingly, we set $K = 0.05d$ and $K = 0.1d$ in the two experiments for FedREP since the transmitted dimension number in FedREP is uncertain but not larger than $K$. Thus, we have $\Gamma \leq K/d$ for FedREP. The top-1 accuracy w.r.t. epochs is illustrated in Figure 6. The results show that FedREP can significantly outperform the existing communication-efficient privacy-preserving baseline SparseSecAgg when there is no Byzantine attack.

In addition, we have tried different learning rates for SparseSecAgg and it has the best performance when learning rate equals 5. As illustrated in Figure 7, FedREP significantly outperforms SparseSecAgg on top-1 accuracy when communication cost is similar. The communication cost of SparseSecAgg is much more than FedREP when the performance on top-1 accuracy is comparable. For one reason, FedREP is based on top-$K$ sparsification while SparseSecAgg is based on random-$K$ sparsification. For another reason, FedREP adopts error-compensation technique while SparseSecAgg does not.

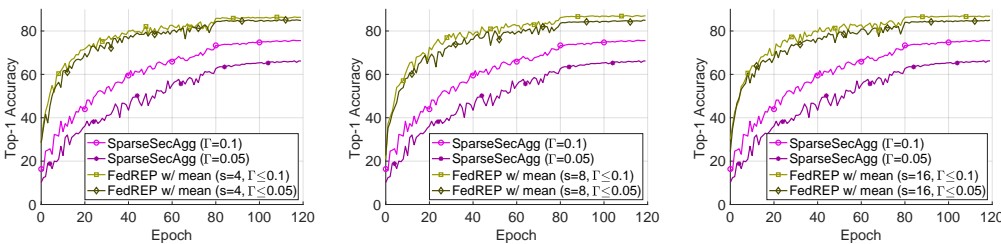

Figure 6: Top-1 accuracy w.r.t. epochs of FedREP and SparseSecAgg when there is no attack. The buffer size $s$ for FedREP is set to be $4$ (left), $8$ (middle) and $16$ (right), respectively.

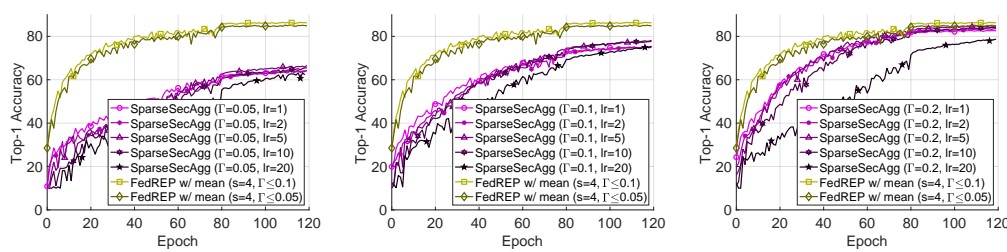

Figure 7: Top-1 accuracy w.r.t. epochs of FedREP and SparseSecAgg when there is no attack. $\Gamma$ for SparseSecAgg is set to $0.05$ (left), $0.1$ (middle) and $0.2$ (right), respectively. The learning rate (lr) for FedREP is set to $0.5$.

## C.5 COMPARISON WITH SHARE

We empirically compare the performance of FedREP and SHARE (Velicheti et al., 2021) in this section. Both FedREP and SHARE are FL frameworks that can work with various local training algorithms on clients. Compared to SHARE, the main advantage of FedREP is the consensus sparsification. Moreover, FedREP degenerates to SHARE when consensus sparsification hyper-parameter $K = md$. Therefore, we compare SHARE and FedREP with different $\Gamma$ when keeping other conditions the same. Specifically, we set buffer size (a.k.a. cluster size in SHARE) to be $4$ and local training algorithms to be vanilla SGD for each method. The other settings are the same as those in Section 5 of the main text of this paper. Since the actually transmitted dimension number is uncertain in each communication round for FedREP, we count the average transmitted dimension number of all communication rounds and use it as a measurement of communication cost. The experimental results of FedREP and SHARE when there are no Byzantine clients are illustrated in Figure 8. The experimental results of FedREP and SHARE when there are 3 Byzantine clients under bit-flipping attack, ALIE attack and FoE attack are illustrated in Figure 9, Figure 10 and Figure 11, respectively. As we can see from the empirical results, compared to SHARE, there is almost no loss on the convergence rate and final accuracy when $\Gamma$ is about $0.079$ for FedREP. In addition, there is only a little loss on final accuracy when $\Gamma$ is as low as about $0.017$.

Interestingly, when under ALIE attack, empirical results show that FedREP has an even higher final accuracy compared to SHARE. We conduct an extra experiment to compare the performance of FedREP and SHARE when there are 7 Byzantine clients with ALIE attack. We set local training algorithm to be momentum SGD with $\beta = 0.9$ and buffer size $s = 2$ in the extra experiment. The other settings are the same. As illustrated in Figure 12, FedREP can still outperform SHARE in this setting. A possible reason is that the consensus sparsification in FedREP can lower the dissimilarity between the updates of different clients and thus lower the aggregation error (please see Definition 1 in the main text for more details). However, it requires more effort to further explore this aspect and we leave it for future work. In summary, FedREP has a comparable performance to SHARE on convergence rate and final accuracy, but has much less communication cost than SHARE.

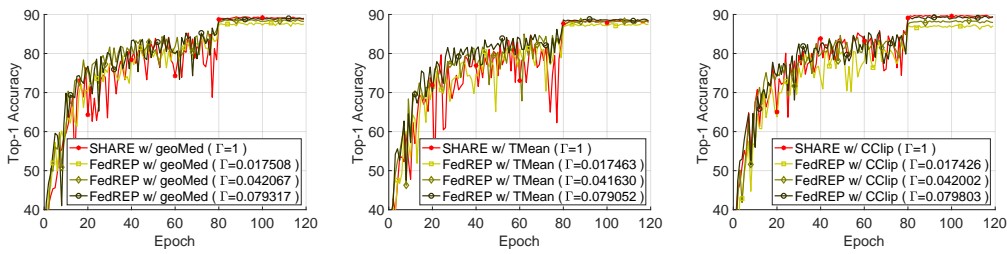

Figure 8: Top-1 accuracy w.r.t. epochs of FedREP and SHARE when there are no Byzantine clients.

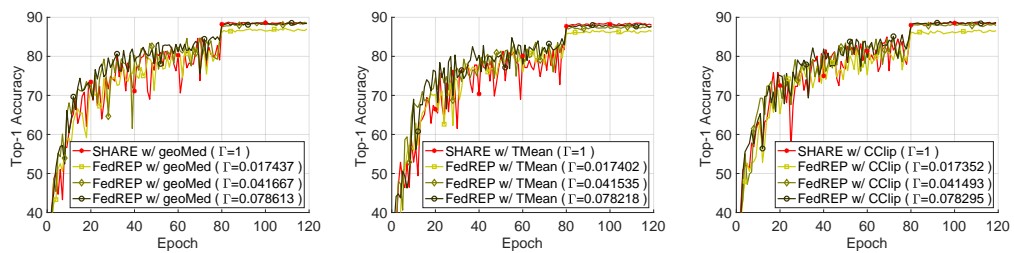

Figure 9: Top-1 accuracy w.r.t. epochs when there are 3 Byzantine clients with bit-flipping attack.

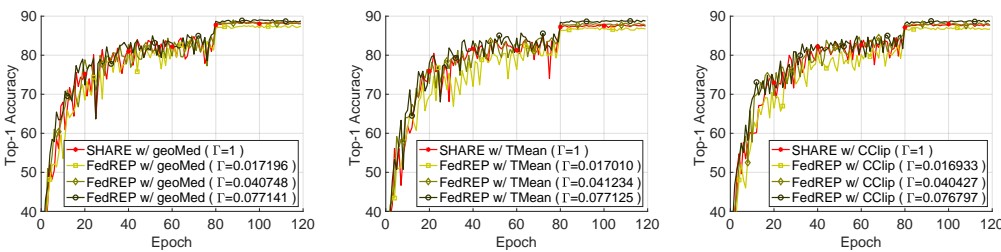

Figure 10: Top-1 accuracy w.r.t. epochs when there are 3 Byzantine clients with ALIE attack.

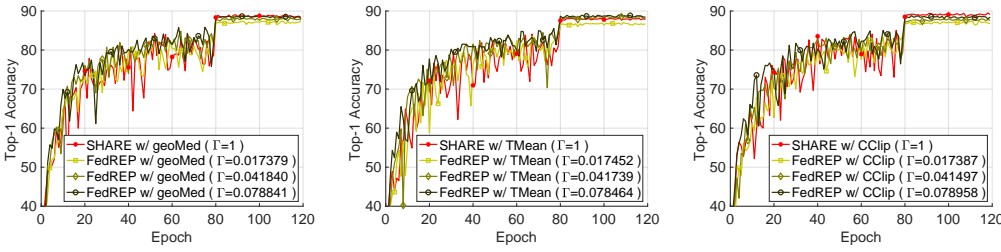

Figure 11: Top-1 accuracy w.r.t. epochs when there are 3 Byzantine clients with FoE attack.

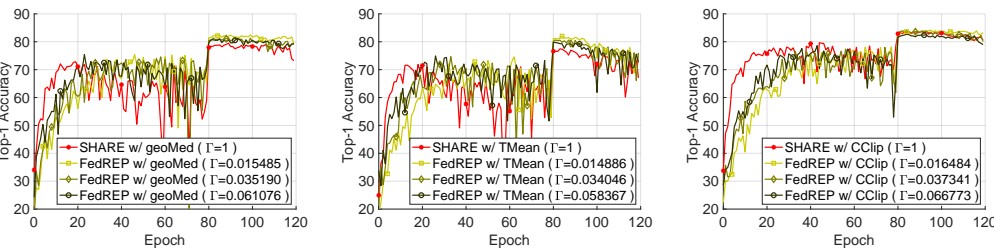

Figure 12: Top-1 accuracy w.r.t. epochs when there are 7 Byzantine clients with ALIE attack.

