# OpenReview forum: "FedREP: A Byzantine-Robust, Communication-Efficient and Privacy-Preserving Framework for Federated Learning"
_ICLR.cc/2023/Conference — Submitted to ICLR 2023_

### Official Review · Reviewer_jrY6 · 2022-10-19

**Confidence:** 4
**Correctness:** 2
**Technical Novelty And Significance:** 3
**Empirical Novelty And Significance:** 2
**Recommendation:** 3

**Clarity, Quality, Novelty And Reproducibility:**

General comments:
1. Not including the algorithm in the main text does incur inconvenience for the reading experience.
2. Please state Theorem 2 clearly: what is the algorithm $\mathcal{M}$, what are input datasets $\mathcal{T}_{1,2}$, and what are corresponding possible output $\mathcal{S}$. So far the statement of theorem 2 is unclear and therefore the proof is impossible to follow.
3. I recommend that authors submit core code for experiments in the supplementary materials.
4. The statement for theorem 3 and 4 should be very explicit to distinguish their differences (wrt to difference optimization algorithms for local clients). I recommend the authors polish the narrative of the paper: be concise in textual description, and be clear and specific in theorem statement.

Specific comments:
1) On page 5, the update formula for $\mathbf{u}^{t+1}_k$ is  $\mathbf{u}^{t+1}_k= \mathbf{u}^{t}_k - \widetilde{\mathbf{g}}^t_k$. This is inconsistent with the expression for $\mathbf{u}^{t+1}_k$ on page 6.
2) What is $\Gamma$ in Figure 2 in Section 5?
3) Buffers $\mathbf{b}_l$ are used on the FedREP Server side. I do not see any involvement of buffer quantities in the optimization proof. It is okay to only include buffer as a practice hack. For theory statement, please be specific.
4) To make the proof more helpful to wider audience, I suggest authors to add some explanation about the motivation of considering $\widehat{\mathrm{w}}^{t+1}$ in equations (145) and (224) respectively. These two steps are decisive for setting up the entire optimization argument.

**Strength And Weaknesses:**

Strength:

--> the structure of this submission is clear.

--> Proposition 2 about the contraction property of the proposed ConSpar algorithm is useful for the general community.

Weakness:

--> Some proof steps are problematic.

--> The writing should be overhauled.

**Summary Of The Paper:**

This submission introduces a local sparsification scheme in the federated learning setting to reduce the data transmission burden. The scheme starts with sampling certain ratio of coordinates with largest magnitude, before adding in some randomly selected coordinates to meet the pre-designated sampling budget.

The authors intend to prove that the suggest sparsification scheme has differential-privacy property, and also shows the convergence of the resulted federated learning algorithm.

**Summary Of The Review:**

The optimization proof did not include the buffer steps, which are stated in the main algorithm.

The writing of the paper needs major overhaul. The description of technical details should be direct to the point, specific and explicit, and the statement of theorems must be rigorous and detailed.

I recommend rejection.

---

> ### Author Response · Authors · 2022-11-16
> **Response to Reviewer jrY6**
>
> We greatly thank the reviewer for the valuable comments. We will respond to the general comments (GC) and specific comments (SC) point by point as follows.
>
> **GC1. Not including the algorithm in the main text does incur inconvenience for the reading experience.**
>
> We strongly agree with the reviewer that presenting the detailed algorithm in the main text could increase the readability. Actually, our work involves Byzantine robustness, communication efficiency, and privacy preservation. It takes much space to discuss the three properties, and there is little room left for the detailed algorithm. Given this, we have to move it to the top of the Appendix in the current version. Meanwhile, we promise to include the algorithm in the main text when more space is allowed.
>
> **GC2. Please state Theorem 2 clearly: what is the algorithm $\mathcal{M}$, what are input datasets $\mathcal{T}_{1,2}$, and what are corresponding possible output $\mathcal{S}$.**
>
> In Theorem 2, $\mathcal{M}$ is the mechanism in consensus sparsification that takes the set of top coordinates $\mathcal{T}_k^t$ as an input and outputs $\mathcal{I}_k^t$. In the proof of Theorem 2, $\mathcal{T}_1$ and $\mathcal{T}_2$ are two arbitrary adjacent coordinate sets that satisfy $|\mathcal{T}_1|=|\mathcal{T}_2|=\frac{ K}{m}$. We have modified the statement of Theorem 2 in the revised version and sincerely thank the reviewer for the constructive suggestion.
>
> **GC3. I recommend that authors submit core code for experiments in the supplementary materials.**
>
> Actually, we have submitted the core code of FedREP in the initial version. The remaining parts will be released when this work is published.
>
> **GC4. The statement for theorem 3 and 4 should be very explicit to distinguish their differences (wrt to difference optimization algorithms for local clients). I recommend the authors polish the narrative of the paper.**
>
> Theorem 3 is for FedREP with local SGD, and Theorem 4 is for FedREP with general optimization algorithms that satisfy Assumption 7. The order of convergence is the same in these two theorems. The main differences lie in the coefficients $\gamma_1$, $\gamma_2$, and $\Delta$. We have made the values of these coefficients more explicit in the revised version.
>
> **SC1. On page 5, the update formula for $u_k^{t+1}$ is inconsistent with the expression on page 6.**
>
> We have deleted the update formula for $u_k^{t+1}$ on page 5. Actually, we put the formula here by mistake in the initial version. In Section 3.2, we mainly introduce the core algorithm of ConSpar (without error feedback). In Section 3.3, we introduce FedREP, where ConSpar with error feedback is used. We sincerely thank the reviewer for pointing out the mistake and apologize for the caused inconvenience.
>
> **SC2. What is $\Gamma$ in Figure 2 in Section 5?**
>
> $\Gamma$ is defined to be the ratio of the transmitted dimension number to the total dimension number. We are sorry that the statement was wrongly put in the Appendix. We have added this to the main text in the revised version.
>
> **SC3. I do not see any involvement of buffer quantities in the optimization proof.**
>
> Actually, we consider the buffer secure aggregation and the robust aggregation as a unit $SRAgg(\cdot)$ in our proof, which is stated before Assumption 1. We use the notation by following the previous work (Karimireddy et al., 2022), where it is shown that many widely used robust aggregator combined with buffer averaging is $(\delta,c)$-robust. Different buffer size $s$ will lead to different values of $\delta$ and $c$ (Karimireddy et al., 2022).
>
> **SC4. To make the proof more helpful to wider audience, I suggest authors add some explanation about the motivation of considering $\hat{w}^{t+1}$ in equations (145) and (224) respectively.**
>
> We have added explanations about equations (144) and (224) in the Appendix in the revised version. For quick access, we present the added explanation below.
>
> $$\hat w^{t+1}=\hat{w}^t- \eta I\cdot\bar G^t - e^t. \qquad (144)$$
>
> Equation (144) can be interpreted as that $\hat{w}^{t+1}$ is obtained by performing an SGD step on $\hat{w}^t$ with learning rate $\eta I$, gradient approximation $\bar G^t$ and error $e^t$.
>
> $$\hat{w}^{t+1}
>     = \hat{w}^t- (w^t-\bar w^{t+1})- e^t.\qquad (224)$$
>
> Equation (224) can be interpreted as that $\hat{w}^{t+1}$ is obtained by adding a small term $- (w^t-\bar w^{t+1})$ on $\hat{w}^t$ with error $e^t$.
>
> ---
> In summary, we would like to politely point out that our proof includes the buffer steps, as shown in our response to SC3 above. Meanwhile, we have improved the writing accordingly. We sincerely thank the reviewer for the constructive comments and would greatly appreciate it if the reviewer could re-evaluate our work in light of our response.
>
> Sai Praneeth Karimireddy, Lie He, and Martin Jaggi. Byzantine-robust learning on heterogeneous datasets via bucketing. In Proceedings of the International Conference on Learning Representations, 2022.

---

> > ### Comment · Reviewer_jrY6 · 2022-12-13
> > **After rebuttal**
> >
> > I intend to keep my score as it is.
> >
> > GC3: I do not feel comfortable giving greenlight to a manuscript without seeing the core code.
> >
> > SC3: For either presentation purposes or for generalizability, it is unclear to me that incorporating the buffer into the SRAgg is justified, as I intend to think of SRAgg as some aggregation with corresponding deviation bounds.
> >
> > SC4: I do not find this explanation helpful to understand the motivation to build the Lyapunov function in the proof.

---

### Official Review · Reviewer_UUqC · 2022-10-21

**Confidence:** 4
**Correctness:** 2
**Technical Novelty And Significance:** 3
**Empirical Novelty And Significance:** 2
**Recommendation:** 5

**Clarity, Quality, Novelty And Reproducibility:**

The paper is mainly clear. In terms of quality, it is OK, which has room for substantial improvement. To the best of my knowledge, the idea of ConSpar is novel. However, SecAgg, local memory, local updating, and bucketing are pretty standard. The experimental setup is sufficiently described.

**Strength And Weaknesses:**

I think the paper is well-written. There are interesting algorithmic elements including the introduction of random variable $r_k^t$ with binomial distribution to improve privacy. The related work are sufficiently studied. I think ConSpar along with SecAgg is an interesting algorithm in terms of addressing communication costs and privacy risks. However, I am not sure it is much interesting in terms of  robustness.

1- The main weakness of this paper is that the considered attacks are very simple and not designed for ConSpar, which will be elaborated in the following:

- The type of attacks considered are mainly attacks effective against Krum and comed. What if an adversary constructs ${\cal I}_k^t$'s that target the **most informative coordinates of average update of honest clients** and make them zero. In that case, even if the server  perfectly detects all Byzantines, the server can only figure out the average of least informative coordinates of  good client, which does not help the server to learn an effective model. In particular, an attack, which tries to remove the most informative coordinates from the aggregate of good updates should be considered. I also checked the results in Appendix C.2. However, those experiments do not consider this attack.

2- Regarding FoE attack. I am not sure why  the hyperparameter is set to 0.5. Xie et al., 2020a proposed to apply a number of $\epsilon$'s including 0.1 and 10 so I am curious why the authors selected 0.5?

3- The idea is that sparsification makes the clients updates closer to each other. The smaller dissimilarity in Proposition 1 implies that it is easier fo find $\frac{1}{|{\cal G}|}\sum_{k=1}^{|{\cal G}|} \tilde g_k$. However, it does not guarantee anything about the quantity of interest $\frac{1}{|{\cal G}|}\sum_{k=1}^{|{\cal G}|} g_k$. Note that $\frac{1}{|{\cal G}|}\sum_{k=1}^{|{\cal G}|} \tilde g_k$ may be arbitrarily different from $\frac{1}{|{\cal G}|}\sum_{k=1}^{|{\cal G}|} g_k$.

4- Definition 1: The upper bound $\rho$ for (Karimireddy et al., 2021) holds only for those good clients but Definition 1 includes a bound on all $m$ clients.

5- "As shown in previous works (Karimireddy et al., 2021), many widely-used aggregators, such as Krum (Blanchard et al., 2017), geoMed (Chen et al., 2017) and coordinate-wise median (Yin et al., 2018), satisfy Definition 1."

- Could you specify where this is shown in (Karimireddy et al., 2021)? It is shown that Krum and coordinate-wise median are vulnerable to simple attacks (Fang et al., 2020; Xie et al., 2020). I am not sure how much Definition 1 is meaningful in terms of robustness in practical settings.

Minghong Fang, Xiaoyu Cao, Jinyuan Jia, and Neil Gong. Local model poisoning attacks to byzantine-robust federated learning. In Proc. USENIX Security Symposium, 2020.

Cong Xie, Oluwasanmi Koyejo, and Indranil Gupta. Fall of empires: Breaking byzantinetolerant SGD by inner product manipulation. In Proc. Uncertainty in Artificial Intelligence Conference, 2020.

6- Assumption 5 does not hold even for a quadratic loss function. Can the author clarify why Assumptions 3 and 4 are not sufficient to establish convergence guarantees?

7- "Even with quantization, ... suffers from heavy communication cost."

- One advantage of unbiased quantization schemes is that the same test accuracy as the uncompressed baseline is achievable while achieving communication efficiency. It is unclear how much test accuracy is lost under same hyperparameters tuned for uncompressed baseline if someones uses consensus spartifications under no attack scenario?





**Summary Of The Paper:**

The authors study the conditions that a communication compression method should satisfy to be compatible with existing Byzantine-robust methods and privacy-preserving methods. The core of their proposed method is consensus sparsification, which is a variant of top-$K$ sparsification with additional memory and a random selection component aiming at obscuring exact coordinates for sparsification. Their idea is that as long as the clients agree on non-zero coordinates, the dissimilarity among clients reduces, which is intuitive while they achieve communication saving through sparisification and preserve privacy by applying SecAgg off the shelf.

**Summary Of The Review:**

The key idea of this paper is consensus sparsification, which is a variant of top-$K$ sparsification with additional memory and a random selection component to improve privacy. ConSpar along with SecAgg is an interesting algorithm in terms of addressing communication costs and privacy risks. The main weakness is w.r.t. robustness. The attacks considered are very simple and not designed for ConSpar. I think the smaller dissimilarity in Proposition 1 does not imply improved robustness.

---

> ### Author Response · Authors · 2022-11-16
> **Response to Reviewer UUqC (part 2/2)**
>
>
> **Q6. Can the author clarify why Assumptions 3 and 4 are not sufficient to establish convergence guarantees?**
>
> The bounded gradient assumption (Assumption 5) is widely used in the analysis of communication compression with error compensation (Stich et al., 2018, Xie et al., 2020c). Specifically, it is used to provide upper bounds for error compensation vectors on each client. To the best of our knowledge, there are almost no works that establish convergence guarantees of communication compression with error compensation for non-convex cases without the bounded gradient assumption.
>
> **Q7. It is unclear how much test accuracy is lost under same hyperparameters tuned for uncompressed baseline if someones uses consensus spartifications under no attack scenario?**
>
> FedREP is equivalent to SHARE (Velicheti et al., 2021) when the sparsification hyper-parameter $K=md$ (i.e., no compression). We have empirically compared FedREP with SHARE in Appendix C.5. Specifically, the results of FedREP and SHARE no attack are illustrated in Figure 8. There is almost no loss in the convergence rate and final accuracy when about 7.9% of the coordinates are sent for FedREP. In addition, there is only a little loss in final accuracy when 1.7% of the coordinates are sent.
>
> ---
> We sincerely thank the reviewers again for helping us improve our work and hope that we have addressed all the raised concerns. Meanwhile, we would greatly appreciate it if the reviewer could re-evaluate our work in light of our response.
>
> ---
> Cong Xie, Oluwasanmi Koyejo, and Indranil Gupta. Fall of empires: Breaking Byzantine-tolerant sgd by inner product manipulation. In Proceedings of the Conference on Uncertainty in Artificial Intelligence, pp. 261–270, 2020a.
>
> Sebastian U Stich, Jean-Baptiste Cordonnier, and Martin Jaggi. Sparsified sgd with memory. In Advances in Neural Information Processing Systems, pp. 4447–4458, 2018.
>
> Sai Praneeth Karimireddy, Lie He, and Martin Jaggi. Byzantine-robust learning on heterogeneous datasets via bucketing. In Proceedings of the International Conference on Learning Representations, 2022.
>
> Cong Xie, Shuai Zheng, Oluwasanmi O Koyejo, Indranil Gupta, Mu Li, and Haibin Lin. CSER: Communication-efficient sgd with error reset. In Advances in Neural Information Processing Systems, pp. 12593–12603, 2020c.
>
> Raj Kiriti Velicheti, Derek Xia, and Oluwasanmi Koyejo. Secure Byzantine-robust distributed learning via clustering. arXiv preprint arXiv:2110.02940, 2021.

---

> ### Author Response · Authors · 2022-11-16
> **Response to Reviewer UUqC (part 1/2)**
>
> We greatly thank the reviewer for their valuable time and insightful comments. We will address the raised concerns point by point below.
>
> **Q1. What if an adversary constructs $\mathcal{I}_k^t$'s that target the most informative coordinates of average update of honest clients and make them zero?**
>
> The server determines the coordinates to be sent by taking the union of all $\mathcal{I_k^t} $'s (i.e., $\mathcal{I}^t=\cup_{k=1}^m \mathcal{I}_k^t $, as presented in Section 3.2). Thus, $\mathcal{I}_k^t\subseteq\mathcal{I}^t$, which means that a coordinate $j\in[d]$ will definitely be sent when any honest client $k$ includes it in $\mathcal{I}_k^t$. Therefore, Byzantine clients cannot prevent any coordinate $j\in\mathcal{I}_k^t$ from being sent. The only thing that Byzantine clients can do is not to introduce additional 'informative' coordinates. The three types of attacks (minAtk, randAtk, and sameAtk) on coordinates in Appendix C.2 are designed based on this. Meanwhile, we are willing to present more empirical results if the reviewer could let us know about some more specific attacks on coordinates.
>
> Besides, in our analysis (including the $d'$-contraction property in Proposition 2), we do not assume the behavior of Byzantine clients when sending $\mathcal{I}_k^t$. Therefore, the convergence of FedREP is theoretically guaranteed whatever value of $\mathcal{I}_k^t$ is sent from Byzantine clients.
>
> **Q2. (Xie et al., 2020a) proposed to apply a number of $\epsilon$'s including 0.1 and 10 so I am curious why the authors selected 0.5?**
>
> The main reason is that in FedREP, clients will be randomly grouped at each iteration, and the server can only know the average updates of each group. Thus, updates sent by Byzantine clients under FoE attack (Xie et al., 2020a) with $\epsilon=0.1$ will be diluted during the random grouping. Meanwhile, FoE attack with $\epsilon=10$ will be easily detected.
>
> We have tried different $\epsilon$ (ranging from $0.1$ to $100$) for FoE. The value of $\epsilon$ has little effect on the convergence rate and final accuracy of FedREP. We finally set $\epsilon=0.5$ since it leads to a little bit smaller accuracy compared to other settings of FoE.
>
> **Q3. The smaller dissimilarity in Proposition 1 implies that it is easier to find $\frac{1}{|\mathcal{G}|}\sum_{k=1}^{|\mathcal{G}|}\tilde{g_k}$. However, it does not guarantee anything about the quantity of interest $\frac{1}{|\mathcal{G}|}\sum_{k=1}^{|\mathcal{G}|}g_k$.**
>
> Proposition 2 shows that FedREP is a $d'$-contraction (Stich et al., 2018) operator. Specifically, it guarantees that $\mathbb{E}||g_k-\tilde{g_k}||^2\leq(1-\frac{d'_{cons}}{d})||g_k||^2$.
>
> **Q4. The upper bound for (Karimireddy et al., 2021) holds only for those good clients but Definition 1 includes a bound on all clients.**
>
> Actually, the upper bound in Definition 1 is for the true values $v_1,\ldots,v_m$ while a Byzantine client $k$ may send arbitrary value $\hat{v}_k$. Therefore, we do not assume the behavior of Byzantine clients and our definition is actually equivalent to that in (Karimireddy et al., 2021).
>
> In the revised version, to avoid misunderstanding, we have modified the statement of Definition 1, Theorem 1, and Proposition 1. We sincerely thank the reviewer's valuable comment, which greatly helps to improve our work.
>
> **Q5. "As shown in previous works (Karimireddy et al., 2021), many widely-used aggregators, such as Krum, geoMed, and coordinate-wise median, satisfy Definition 1." Could you specify where this is shown? It is shown that Krum and coordinate-wise median are vulnerable to simple attacks (Fang et al., 2020; Xie et al., 2020a). I am not sure how much Definition 1 is meaningful in terms of robustness in practical settings.**
>
> We are so sorry that there are some mistakes in this statement in the initial version. It is shown in the Appendix C.2 of **(Karimireddy et al., 2022)** that these aggregators **combined with averaging in buffers** are $(\delta,c)$-robust. We have modified the statement in the revised version and sincerely thank the reviewer for pointing out the mistake.
>
> It is shown in (Fang et al., 2020) and (Xie et al., 2020a) that Krum/median alone is vulnerable to attacks while (Karimireddy et al., 2022) shows that combining with averaging in buffers can make the two aggregators $(\delta,c)$-robust. Empirical results are also provided in (Karimireddy et al., 2022). There seems no explicit conflict between these works.

---

> > ### Comment · Reviewer_UUqC · 2022-11-18
> > **Response**
> >
> > Thanks for the response. Regarding the effective attack, suppose that the adversary knows ${\cal I}_k^t$'s for all honest clients. Then the adversary has the option to send coordinates that are "supporters" in the sense of (Baruch et al., 2019). If the adversary knows the set ${\cal I}^t$, it should be able to figure out
> >
> > $\{(\tilde {\bf g}^t_k)_{{\cal I}^t}\}$ for $k=1,\ldots,m$. Then it should be able to apply attacks similar to those of (Baruch et al., 2019; Fang et al., 2020; Xie et al., 2020).

---

> > > ### Author Response · Authors · 2022-11-18
> > > **Reply**
> > >
> > > We sincerely thank the reviewer for their quick response and insightful comment. Actually, we have considered the mentioned attacks in the experiments.
> > >
> > > Specifically, the 'minAtk' attack (in Appendix C.2) on coordinate sets $\mathcal{I_k^t}$ is designed with a similar motivation to (Baruch et al., 2019; Fang et al., 2020; Xie et al., 2020). It is designed to lure 'good' clients into sending coordinates with small absolute values. Due to the randomness in stochastic gradients, coordinates with mean values close to $0$ will be more likely to be opposite to the true direction and thus become the "supporters" under attacks. However, as the empirical results in Appendix C.2 show, the performance of FedREP is hardly affected by attackers that adopt both minAtk on sets $\mathcal{I_k^t}$ and existing attacks on coordinates $(\tilde{\textbf{g}}_k^t)_\mathcal{I^t}$. The empirical results are consistent with our theoretical results, which show the robustness of FedREP.
> > >
> > > In addition, the empirical performance of FedREP and the baselines under 'A Little is Enough' (ALIE) attack (Baruch et al., 2019) and 'Fall of Empires' (FoE) attack (Xie et al., 2020) are presented in Section 5. Appendix C. Specifically, the coordinates sent by clients under ALIE or FoE attack are computed based on $(\tilde{\textbf{g}}_k^t)_\mathcal{I^t}$ of all 'good' workers.
> > >
> > > We hope that our response will address the reviewer's concern.

---

> > > > ### Comment · Reviewer_UUqC · 2022-12-13
> > > > **After Rebuttal**
> > > >
> > > > I would like to thank the authors for their response. I would like to increase my score to 5.5 or so but still after reading all comments and responses, I think the paper needs a bit of revision and improvement for publication. In my view, the adversaries can collaborate to disrupt sparsification and robustness goals. Thanks for reminding 'minAtk' attack. But it is still unclear to me what happens if the compromised clients try to keep the maximum number of coordinates on the agreement set (disrupting sparsification) and apply the same attacks proposed in the literature given that some coordinates are known to be set to zero.
> > > >
> > > > Also the number of clients $m$ cannot be too large otherwise $K/m$ will be very small. I am also not very convinced with the responses to "The smaller dissimilarity in Proposition 1... " and "I am curious why the authors selected 0.5?"

---

### Official Review · Reviewer_LcWV · 2022-10-24

**Confidence:** 3
**Correctness:** 3
**Technical Novelty And Significance:** 2
**Empirical Novelty And Significance:** 2
**Recommendation:** 6

**Clarity, Quality, Novelty And Reproducibility:**

Clarity: The method is clear and correct.
Quality: The quality of the proposed method is good.
Novelty: The novelty of the proposed method is marginal.
Reproducibility: The authors provide the code for reproducibility. I do not have the chance to run it.

**Strength And Weaknesses:**

Strength:
1. Reconciling byzantine robustness, communication efficiency and privacy is an important topic in federated learning system design.
2. The evaluational results show reasonable trade-off between the three dimensions.


Weakness:
1. The authors did not discuss the potential malicious behavior during the consensus sparsification. What if some malicious client always upload misleading $I^t_k$? Will that affect the robustness or the privacy of the protocol?

**Summary Of The Paper:**

The paper propose the first federated learning system which supports privacy, communication efficiency and byzantine-robustness. Specifically, it designs a consensus sparsification protocol to compress the update and also keeps compatible with secure aggregation and robust aggregation.

**Summary Of The Review:**

The paper is a borderline one. I slightly tend towards acceptance but I am okay either way.

The biggest problem is that the authors do not discuss the influence of potential malicious attacks during the sparsification step. This makes the privacy guarantee questionable.

----------------------------------------------------------
Thank you for the rebuttal. I decide to keep the score after reading it.

---

> ### Author Response · Authors · 2022-11-16
> **Response to Reviewer LcWV**
>
> We greatly thank the reviewer for the valuable review and for the support of our work. We strongly agree with the reviewer that considering Byzantine robustness, communication efficiency, and privacy preservation in federated learning at the same time is an important topic. Our response to the reviewer's concerns is presented below.
>
> ---
> **Q: What if some malicious client always uploads misleading $\mathcal{I}_k^t$? Will that affect the robustness or the privacy of the protocol?**
>
> We thank the reviewer for the question, which greatly helps us to improve the paper. Actually, we do not assume that Byzantine clients follow the protocol when sending coordinate sets. All the theoretical results in our work hold even if Byzantine clients send arbitrary $\mathcal{I_k^t}$. We also empirically test the performance of FedREP when Byzantine clients send wrong coordinates in Appendix C.2. Specifically, we consider three different Byzantine attacks on coordinates, where Byzantine clients send: i) the coordinates of $K/m$ smallest absolute values (minAtk); ii) random coordinates (randAtk); and iii) coordinates that is the same as a non-Byzantine client (sameAtk, aim to lower the number of the sent coordinates $|\mathcal{I}^t|=|\cup_{k=1}^m \mathcal{I}_k^t| $), respectively. Empirical results show that attacks on the coordinates have little effect on the convergence rate and the final accuracy.
>
> In summary, we have considered the potential Byzantine behaviour on sending coordinate sets $\mathcal{I}_k^t$ in the theoretical analysis and the experiments. Both the theoretical results and the empirical results show the resilience of FedREP when malicious clients always upload misleading $\mathcal{I}_k^t$. To avoid misunderstanding, we have added the statement in Section 3.2, which says that *"Please note that we do not assume the behaviour of Byzantine clients, which may send arbitrary $\mathcal{I}_k^t$"*.
>
> ---
> We have uploaded the revised version and the changed parts have been marked in blue for quick recognition. We would greatly appreciate it if the reviewer could re-evaluate our work in light of our response. We sincerely thank the reviewer again for their valuable time.

---

### Official Review · Reviewer_VCzw · 2022-10-27

**Confidence:** 4
**Correctness:** 3
**Technical Novelty And Significance:** 2
**Empirical Novelty And Significance:** 2
**Recommendation:** 3

**Clarity, Quality, Novelty And Reproducibility:**

**Clarity and Quality:** The presentation has a good level of clarity and technical quality seems good enough.

**Novelty:** The novelty is a bit limited for the following reasons.
* As mentioned in the previous section, while the sparsification scheme is interesting, the novelty of FedREP is quite limited.
* The convergence analysis seems to follow directly from [Stich et al. 2018] and [Karimireddi et al. 2022]. It would be helpful if the authors can explicitly specify what are the new challenges in the analysis (if any).
* Empirical results are restricted to CIFAR-10 and ResNet-20. It would be good to consider other datasets (e.g., some of the LEAF datasets) and model architectures.


**Strength And Weaknesses:**

**Strengths:**
1. The paper considers a practically important and intellectually challenging problem of simultaneously achieving communication efficiency, Byzantine robustness, and privacy.

2. The analysis of sparsification schemes to ensure assumptions in Def. 1 ($(\delta, c)$-robust aggregator) is interesting.

**Weaknesses:**
1. For privacy, the paper proposes a hierarchical approach where clients are randomly partitioned and then an aggregate update is computed for each partition using secure aggregation. This is essentially the bucketing approach from [Karimireddy et al. ICLR 2022] with additional secure aggregation. From the perspective of novelty, this is fairly limited. More importantly, this method would only enhance privacy, but there is no analysis about how much is the enhancement. (In fact, the authors acknowledge at the end of Sec. 3 that further study would require to quantify the privacy leakage.) On the other hand, the paper claims in the Abstract and Introduction (Table 1) that their method is privacy-preserving. This seems like a bit of over claiming.

2. The analysis in Theorem 1 is to ensure the assumptions in Def. 1 ($(\delta, c)$-robust aggregator). The assumption of bounded pairwise distances is sufficient, but it is not discussed if it is necessary. Can there be another definition for a robust aggregator that does not need to assume bounded pairwise distances? There may be empirical robust aggregators that perform decently even without the assumption of bounded pairwise distances. It would be important to discuss this point.

3. The $\epsilon$ parameter of DP in Theorem 2 seems to be quite large. In particular, it is of the order $O\left(\frac{\frac{K}{m}(d - \frac{K}{m})}{\alpha}\right)$, which seems to be at least $O(d)$. Since deep neural networks have large $d$, the value of $\epsilon$ may be too high for practice. It would be important to discuss this further.


**Summary Of The Paper:**

This paper considers the problems of communication efficiency, Byzantine robustness, and privacy in federated learning (FL). The paper focuses on sparsification for compression (in order to achieve communication efficiency), and analyzes what are conditions on sparsification to ensure that the expected pairwise l2-distance between client updates after sparsification remains bounded. Next, the paper proposes a sparsification technique -- consensus sparsification, which is a two-phase scheme that ensures that all clients agree on the same set of coordinates. The paper then combines it with a hierarchical scheme to enhance privacy, wherein sparsified updates from a subset of clients are first aggregated using secure aggregation, and then a robust aggregation is applied on top for Byzantine robustness. The paper analyzes the convergence rate for the proposed scheme.

**Summary Of The Review:**

My main concerns are mentioned in the section on Weaknesses. Due to these issues along with a limited novelty, it seems to me that the paper is not yet ready for publication.

Additional suggestions and questions to improve the quality of the presentation are give below:

1. In Theorem 1, it is not clear why the conditions $\sum_{j\in[d]} \xi_{k,k’,j} =1$ and $\sum_{j\in[d]} \zeta_{k,j} =1$ will be satisfied. Is it assumed that update vectors are normalized?

2. Footnote 3 mentions that random quantization can be adopted for “more privacy preservation”. What do the authors mean by “more privacy preservation”? Why would random quantization increase the level of privacy?

3. On page 2, it is mentioned that there are two ways to achieve privacy: differential privacy and secure aggregation. This is a bit misleading. Secure aggregation does not guarantee the privacy of training data, but only ensures the privacy of computation (i.e., aggregation operation). In other words, SecAgg ensures that the clients (and the server) observe only the aggregate model and cannot learn individual model updates. DP would be necessary to ensure input privacy. (For instance, in case of only 2 clients, even when secure aggregation is used, each client can compute the model update of the other client from the average.) It will be helpful to discuss this.

4. Before stating the main contributions, the paper claims that: “we theoretically analyze the tension among Byzantine robustness, communication efficiency and privacy preservation, and propose a novel FL framework called FedREP”. The paper analyzes the tension between sparsification and the condition required by $(\delta, c)$-robust aggregator. In other words, the tension between communication efficiency and Byzantine robustness. It is not cleat where the tension between the privacy and the other objectives is analyzed. It would be good to give a bit more details.

---

> ### Author Response · Authors · 2022-11-16
> **Response to Reviewer VCzw (part 2/2)**
>
> **Q6. On page 2, it is mentioned that there are two ways to achieve privacy: differential privacy and secure aggregation. This is a bit misleading. ...........**
>
> We have revised the mentioned statements on page 2 accordingly and greatly thank the reviewer for the constructive suggestion on improving our work.
>
> **Q7. It is not clear where the tension between the privacy and the other objectives is analyzed. It would be good to give a bit more details.**
>
> Actually, we analyze the tension between secure aggregation and existing sparsification methods in the first paragraph of Section 3.1. Although the analysis is not complex, the result is indeed constructive and inspires the design of ConSpar and FedREP. Specifically, the two analyses in Section 3.1 from different perspectives indicate the same requirement, i.e., to make non-Byzantine clients agree on the non-zero coordinates in sparsification.
>
> **Q8. The novelty is a bit limited. The convergence analysis seems to follow directly from [Stich et al. 2018] and [Karimireddi et al. 2022]. It would be helpful if the authors can explicitly specify what are the new challenges in the analysis.**
>
> Firstly, we would like to point out that we do not pursue a tighter bound for convergence in this work. We mainly focus on how to simultaneously obtain Byzantine robustness, communication efficiency and privacy preservation in FL. Thus, it is reasonable that we follow the existing proof sketches of communication-efficient methods and Byzantine-robust methods.
>
> In addition, properly combining the analysis of communication-efficient methods (Stich et al. 2018) and Byzantine-robust methods (Karimireddi et al. 2022) is not a trivial work. The convergence of FedREP is proved smoothly due to the well-designed ConSpar. Specifically, the property of ConSpar presented in Proposition 1 ($\mathbb{E}||\tilde{g_k}^t-\tilde{g_{k'}}^t||^2\leq \mathbb{E}||g_k^t-g_{k'}^t||^2$) is fundamental in the convergence analysis of FedREP. For existing methods, the dissimilarity $\mathbb{E}||\tilde{g_k}^t-\tilde{g_{k'}}^t||^2$ could be enlarged after sparsification. As shown in our response to Q2 above, $O(\delta \rho^2)$ is the tightest order of error for a robust aggregator where $\mathbb{E}||\tilde{g_k}^t-\tilde{g_{k'}}^t||^2\leq\rho^2$. In other words, the analysis in [Stich et al. 2018] and [Karimireddi et al. 2022] cannot be simply combined for existing sparsification methods. The convergence analysis for FedREP is finished due to the well-designed sparsification method ConSpar.
>
> **Q9. It would be good to consider other datasets (e.g., some of the LEAF datasets) and model architectures.**
>
> We greatly thank the reviewer for the suggestion. We are conducting experiments on more datasets with more models. Meanwhile, we also hope that the reviewer could consider our empirical results in Section 5 and Appendix C.1 to C.5. The empirical results show the performance of FedREP from different perspectives and are consistent with our theoretical results.
>
> ---
>
> We sincerely thank the reviewer for the constructive comments. We have uploaded the revised version and would greatly appreciate it if the reviewer could re-evaluate our work in light of our response.

---

> > ### Comment · Reviewer_VCzw · 2022-12-01
> > **Thank you**
> >
> > I thank the authors for answering my questions. While several points are addressed, some key concerns, unfortunately, still remain. Therefore, I retain my original score.
> >
> > * My main concern is that the paper claims the protocol to be privacy-preserving without providing a rigorous threat model and formal privacy analysis of the overall method.
> >
> > In their response, to justify privacy, the authors mention the following:
> > > In FedREP, the set of coordinates $\mathcal{I}^t_k$ and the sparsified vector $\tilde{g}^t_k$ are transmitted in each training round. The differential privacy property of the mechanism that generates $\mathcal{I}^t_k$ is proved in Theorem 2. The sparsified vector $\tilde{g}^t_k$ is transmitted by secure aggregation, the privacy-preserving property of which has been adequately studied in previous works. Therefore, all the transmitted information has been properly protected by privacy-preserving techniques with theoretical guarantees.
> >
> > * To claim a protocol to be privacy-preserving, it is important to mention explicit threat model and formally prove privacy statements. Claiming a method to be privacy preserving with informal arguments as above can be misleading, since it often leads to attacks and it is not clear when the method is private.
> >
> >   * For instance, what happens if some clients collude with the server during consensus sparsification?
> >
> >   * In FedREP, secure aggregation is used only on a subset of clients. This leaks more information to the server than applying secure aggregation on all the clients together. A formal privacy analysis typically covers trade-offs between privacy leakage and communication efficiency/accuracy.

---

> > > ### Author Response · Authors · 2022-12-01
> > > **Follow-up Response to Reviewer VCzw**
> > >
> > > We thank the reviewer for letting us know about the remaining concern, which is mainly about privacy.
> > >
> > > **Q1. My main concern is that the paper claims the protocol to be privacy-preserving without providing a rigorous threat model and formal privacy analysis of the overall method.**
> > >
> > > **About threat model:**
> > > The Byzantine threat model and privacy threat model are presented in Section 2. The privacy threat model in this work is consistent with many previous works, where a malicious server inspects the received messages and attempts to recover the data on clients based on the received messages.
> > >
> > > **About privacy analysis:**
> > > Before answering the reviewer's question, we would like to first present the following statements in the frequently-cited overview about federated learning (Kairouz et al., 2021):
> > > >"Producing a system with all of the above ideal privacy properties would be a daunting feat on its own, and even moreso while also guaranteeing other desirable properties such as ease of use for all participants, the quality and fairness of the end user experiences (and the models that power them), the judicious use of communication and computation resources, resilience against attacks and failures, and so on.
> > > >
> > > >"Rather than allowing perfect to be the enemy of good, we advocate a strategy wherein the overall system is composed of modular units which can be studied and improved relatively independently, while also reminding ourselves that we must, in the end, measure the privacy properties of the complete system against our ideal privacy goals set out above."
> > > >
> > > In this work, we follow the suggestions presented above in (Kairouz et al., 2021) and thus separately study the property of each modular unit. $\mathcal{I}_k^t$'s are transmitted via secure aggregation, the privacy property of which has been fully studied in previous works. The privacy property of ConSpar has been presented in Theorem 2. After presenting the detailed algorithm, we evaluate the privacy properties of FedREP against ideal goals at the end of Section 3.
> > >
> > > **Q2. "In FedREP, ...... guarantees." Claiming a method to be privacy-preserving with informal arguments as above can be misleading.**
> > >
> > > Actually, the statement in our reply is a brief summary of the privacy property. We apologize for the caused misunderstanding. The privacy threat model is presented in Section 2. The privacy property of secure aggregation is fully studied in previous works (Bonawitz et al., 2017). The differential privacy property of ConSpar is formally presented in Definition 2 and Theorem 2.
> > >
> > >
> > > **Q3. What happens if some clients collude with the server during consensus sparsification?**
> > >
> > > We thank the reviewer for the comments. Clients have access to each $\mathcal{I}_k^t$. The set of original top-$K$ coordinates $\mathcal{T}_k^t$ is only stored on each client itself. Therefore, the server would know nothing more during consensus sparsification when some clients are colluders.
> > >
> > > **Q4. In FedREP, secure aggregation is used only on a subset of clients. This leaks more information to the server than applying secure aggregation on all the clients together. A formal privacy analysis typically covers trade-offs between privacy leakage and communication efficiency/accuracy.**
> > >
> > > In mathematics, the result of secure aggregation is exactly the same as that of the vanilla mean. In practice, due to the limited digits of computing devices, there could be a small rounding error, which is usually negligible. Then we would like to discuss about the trade off between privacy and efficiency.
> > >
> > > The communication cost of initial secure aggregation (SecAgg) is $O(m^2)$ (Bonawitz et al., 2017), where $m$ is the number of clients. In FedREP, there are $m/s$ buffers and the communication cost of secure aggregation in each buffer is $O(s^2)$. Therefore, the total communication cost of sharing $\tilde{g}_k^t$ in each round is $O(ms)$. Meanwhile, according to the property of $t$-out-of-$n$ Shamir's secret sharing in SecAgg, any set of at most $t-1$ shares gives nothing about any other shares (Bonawitz et al., 2017). Therefore, FedREP can tolerate up to $s-1$ colluders in each buffer. In summary, as $s$ increases, the communication cost will also increase, but more colluders can be tolerated in each buffer.
> > >
> > > We will add the discussion above in future versions since revisions are not allowed at the current stage.
> > >
> > > ---
> > > We thank the reviewer for their valuable time and insightful comments again. We hope that our response could address the reviewer's concern.

---

> > > ### Author Response · Authors · 2022-12-01
> > > **References**
> > >
> > > Due to the limited length of a single comment, the references mentioned in our follow-up reply are separately listed below.
> > >
> > > ---
> > >
> > > Peter Kairouz, H Brendan McMahan, Brendan Avent, Aure ́lien Bellet, Mehdi Bennis, Arjun Nitin Bhagoji, Kallista Bonawitz, Zachary Charles, Graham Cormode, Rachel Cummings, et al. Advances and open problems in federated learning. Foundations and Trends in Machine Learning, 14(1–2):1–210, 2021.
> > >
> > > Keith Bonawitz, Vladimir Ivanov, Ben Kreuter, Antonio Marcedone, H Brendan McMahan, Sarvar Patel, Daniel Ramage, Aaron Segal, and Karn Seth. Practical secure aggregation for privacy-preserving machine learning. In proceedings of the 2017 ACM SIGSAC Conference on Computer and Communications Security, pp. 1175–1191, 2017.

---

> ### Author Response · Authors · 2022-11-16
> **Response to Reviewer VCzw (part 1/2)**
>
> We greatly thank the reviewer for their valuable time and insightful comments. We answer the raised questions point by point below.
>
> **Q1. The claim that FedREP is privacy-preserving seems like a bit of overclaiming. More importantly, this method would only enhance privacy, but there is no analysis about how much is the enhancement. (In fact, the authors acknowledge at the end of Sec. 3 that further study would require to quantify the privacy leakage.)**
>
> In FedREP, the set of coordinates $\mathcal{I}_k^t$ and the sparsified vector $\tilde{g}_k^t$ are transmitted in each training round. The differential privacy property of the mechanism that generates $\mathcal{I}_k^t$ is proved in Theorem 2. The sparsified vector $\tilde{g}_k^t$ is transmitted by secure aggregation, the privacy-preserving property of which has been adequately studied in previous works. Therefore, all the transmitted information has been properly protected by privacy-preserving techniques with theoretical guarantees.
>
> Meanwhile, we would like to clarify that the discussion at the end of Section 3 is to inspire further study on potential privacy attacks. To the best of our knowledge, there are no existing methods that can recover training data based on the set of coordinates $\mathcal{I}_k^t$ in FedREP. Designing novel privacy attacks based on the set of coordinates is beyond the scope of this work.
>
> **Q2. Can there be another definition for a robust aggregator that does not need to assume bounded pairwise distances? There may be empirical robust aggregators that perform decently even without the assumption of bounded pairwise distances. It would be important to discuss this point.**
>
> Actually, we have pointed out that $O(\delta \rho^2)$ is the tightest order after Definition 1. Specifically, it is theoretically proved in (Karimireddy et al., 2021) that any aggregator necessarily has an error that is not smaller than $\delta \rho^2$. Theoretical bounds are provided for the worst case. An empirical aggregator may perform decently in some cases, but there exists an attack manner that will foil the aggregator.
>
> Sai Praneeth Karimireddy, Lie He, and Martin Jaggi. Learning from history for Byzantine robust optimization. In Proceedings of the International Conference on Machine Learning, pp. 5311–5319, 2021.
>
> **Q3. The $\epsilon$ parameter of DP in Theorem 2 seems to be quite large, which seems to be at least $O(d)$.**
>
> We politely guess that the reviewer might have omitted the $\ln(\cdot)$ operator. Actually, the $\epsilon$ parameter in Theorem 2 is $\ln\left(\frac{(1+\alpha)\cdot\frac{ K}{m}(d-\frac{ K}{m}+1)}{2\alpha}\right)$, which is of the order $O(\ln d)$ with respect to model dimension $d$. Moreover, we would like to point out that to the best of our knowledge, ConSpar is the first sparsification method where multiple coordinates are sent with a theoretical guarantee about DP property.
>
> **Q4. In Theorem 1, it is not clear why $\sum_{j\in[d]}\xi_{k,{k'},j}=1$ and $\sum_{j\in[d]}\zeta_{k,j}=1$. Is it assumed that update vectors are normalized?**
>
> Due to that $\mathbb{E}||v_k-v_{k'}||^2=\sum_{j\in[d]}\mathbb{E}[(v_k)j-(v_{k'})j]^2$
>
> *($(v_k)j$ denotes the $j$-th coordinate of vector $v_k$. $j$ should have been the subindex of $(v_k)$ here. However, an unknown compile error occurred when we wrote that. Thus, we have to present the formula in the current form and sincerely apologize for the inconvenience.)*
>
> where $\mathbb{E}||v_k-v_{k'}||^2=(\rho_{k,k'})^2$ and $\mathbb{E}[(v_k)j-(v_{k'})j]^2=\xi_{k,{k'},j}(\rho_{k,k'})^2$, it is obtained that $(\rho_{k,k'})^2=\sum_{j\in[d]}\xi_{k,{k'},j}(\rho_{k,k'})^2$. It directly implies that $\sum_{j\in[d]}\xi_{k,{k'},j}=1$. The equation that $\sum_{j\in[d]}\zeta_{k,j}=1$ can be obtained in a similar way.
>
> **Q5. Footnote 3 mentions that random quantization can be adopted for “more privacy preservation”. What do the authors mean by “more privacy preservation”? Why would random quantization increase the level of privacy?**
>
> Quantization can make each $g_k^t$ on a finite field. When each $g_k^t$ is on a finite field, secure aggregation (SecAgg) is guaranteed to make server know nothing about each $g_k^t$. When $g_k^t$ is on an infinite field (e.g., $\mathbb{R}^d$), SecAgg can make server not know the exact value of each $g_k^t$. However, server may obtain some information about the probability distribution of $g_k^t$. The noise scale of random masks in SecAgg decides how much information about the distribution of $g_k^t$ can be inferred. In summary, random quantization could increase the level of privacy because the quantized gradient is on a finite field.

---

### Author Response · Authors · 2022-11-16
**General response to all reviewers**

We sincerely thank all the reviewers for their valuable time and insightful comments. We have revised our work accordingly and submitted the latest version. The changed statements have been marked in blue for quick recognition. Besides, we would like to briefly summarize the strengths mentioned in the reviews below.

1. The problem that simultaneously obtaining Byzantine robustness, communication efficiency, and privacy preservation in FL is challenging and important; (Reviewer VCzw and LcWV)
2. The structure of this paper is clear; (all reviewers)
3. The idea of consensus sparsification is novel and interesting; (Reviewer VCzw and UUqc)
4. The quality of this work is good; (Reviewer VCzw and LcWV)

Meanwhile, more than one reviewer raises concerns about inadequate consideration of potential attacks on coordinate sets $\mathcal{I}_k^t$. However, we politely think that it might be a misunderstanding. Our theoretical analysis allows Byzantine clients to send arbitrary values of $\mathcal{I}_k^t$. We have also considered different types of attacks on $\mathcal{I}_k^t$ in experiments in Appendix C.2. Empirical results show that attacks on coordinates have little effect on the convergence rate and final top-1 accuracy. We have added statements in Section 3.2 to avoid causing confusion to readers and greatly thank the reviewers for helping us improve our work.

The other concerns are addressed in the detailed response to each reviewer. We would greatly appreciate it if the reviewers could re-evaluate our work in light of our response.

---

### Decision · Program_Chairs · 2023-01-20

**Decision:**

Reject

**Justification For Why Not Higher Score:**

Low scores from the most informed reviewers.

**Justification For Why Not Lower Score:**

N/A

**Metareview: Summary, Strengths And Weaknesses:**

This paper proposes a technique for reducing communication in multi-silo FL. The technique is referred to as consensus sparsification and is a two-phase scheme that ensures that all clients agree on the same set of coordinates with additional randomization to ensure a level of differential privacy. The paper then combines it with a hierarchical scheme to enhance privacy and robust aggregation is applied on top for Byzantine robustness. The paper analyzes the convergence rate for the proposed scheme and empirically evaluates it against some other communication-efficient baselines.
Sparsification is a potentially useful way to reduce communication (although many more advanced techniques are known). The particular approach presented in this work is new and promising. However reviewers have found and detailed a significant number of issues with presentation and clarity of this work suggesting that it would benefit from a major revision.